# Dual Optimistic Ascent (PI Control) is the Augmented Lagrangian Method in Disguise

**Juan Ramirez**[*]     **Simon Lacoste-Julien**[‡]

Mila - Quebec AI Institute
DIRO, Université de Montréal
[‡] Canada CIFAR AI Chair

## Abstract

Constrained optimization is a powerful framework for enforcing requirements on neural networks. These constrained deep learning problems are typically solved using first-order methods on their min-max Lagrangian formulation, but such approaches often suffer from oscillations and can fail to find all local solutions. While the Augmented Lagrangian method (ALM) addresses these issues, practitioners often favor dual optimistic ascent schemes (PI control) on the standard Lagrangian, which perform well empirically but lack formal guarantees. In this paper, we establish a previously unknown equivalence between these approaches: dual optimistic ascent on the Lagrangian is equivalent to gradient descent-ascent on the Augmented Lagrangian. This finding allows us to transfer the robust theoretical guarantees of the ALM to the dual optimistic setting, proving it converges linearly to all local solutions. Furthermore, the equivalence provides principled guidance for tuning the optimism hyper-parameter. Our work closes a critical gap between the empirical success of dual optimistic methods and their theoretical foundation in the single-step, first-order regime commonly used in constrained deep learning.

## 1 Introduction

Machine learning problems frequently lead to constrained optimization formulations, where the objective function encodes the learning task, and the constraints guide optimization toward models with desirable properties such as sparsity (Gallego-Posada et al., 2022), fairness (Cotter et al., 2019a), or safety (Dai et al., 2024). Alternatively, learning can be incorporated directly into the constraints, as in SVMs and Feasible Learning (Ramirez et al., 2025b).

A widely adopted approach for solving constrained optimization problems involving deep neural networks is to formulate their associated Lagrangian and seek its min–max points (Elenter et al., 2022; Hashemizadeh et al., 2024; Hounie et al., 2023b; Kotary and Fioretto, 2024). This is typically done through gradient descent–ascent: descent on the problem (*primal*) variables—often with adaptive variants such as Adam (Kingma and Ba, 2015)—and ascent on the Lagrange multipliers (*dual* variables). This first-order Lagrangian approach is popular in constrained deep learning because it empirically handles non-convex problems and scales efficiently to models with billions of parameters.

However, gradient descent-ascent on the Lagrangian suffers from two major limitations: ① in the non-convex setting, it may fail to converge to certain local solutions of the problem—convergence is guaranteed only for solutions that correspond to local min–max points of the Lagrangian (Bertsekas, 2016, Prop. 5.4.2); and ② it often exhibits oscillations, where iterates repeatedly move in and out of the feasible set, slowing down convergence (Platt and Barr, 1987).

A seminal approach that addresses these limitations is the *Augmented Lagrangian* method (ALM), which incorporates a quadratic penalty term on the constraints (Hestenes, 1969; Powell, 1969; Rockafellar, 1973). With suitable hyperparameter choices, this method converges to all strict and regular local solutions of non-convex constrained problems, while mitigating oscillations.

Despite these advantages, the Augmented Lagrangian method is not the standard in constrained deep learning. Instead, the community largely relies on the original Lagrangian approach, mitigating its oscillatory behavior with alternative Lagrange multiplier updates such as the PI control strategy (Sohrabi

---

[*]Correspondence to: `juan.ramirez@mila.quebec`

et al., 2024; Stooke et al., 2020), also known as generalized optimistic gradient ascent (Mokhtari et al., 2020). These dual optimistic ascent methods have shown strong empirical performance in reducing oscillations, but their convergence properties remain largely underformalized.

This paper addresses this gap. We establish a simple yet previously unknown result: gradient descent–optimistic ascent on the Lagrangian is equivalent to gradient descent–ascent on the Augmented Lagrangian. For equality-constrained problems, the equivalence holds exactly at the level of matching primal iterates (Theorem 1). For general inequality constraints, we prove that both methods converge to the same set of points—namely, all local solutions of the constrained problem (Theorem 2).

In §4, we use this equivalence to formalize properties of the dual optimistic ascent (PI control) framework. We show that it strictly improves over standard GDA on the Lagrangian by recovering all local problem solutions (Theorem 3) and provide local convergence guarantees to these solutions (Corollaries 2 and 3). We further characterize the role of the optimism coefficient (proportional gain), formally showing that larger values reduce oscillations but may induce ill-conditioning (§4.3). Finally, in §5, we provide experimental results validating our theoretical findings.

Taken together, these results imply that dual optimistic ascent is principled *only* insofar as it mimics the Augmented Lagrangian method in the single-step, first-order regime. In more general settings—such as those employing multiple primal steps or second-order primal optimizers—this equivalence no longer holds, and dual optimistic ascent fails to inherit the robust convergence guarantees of the ALM; in such cases, we advocate for the explicit Augmented Lagrangian method instead.

**Scope.** All results are presented in the context of constrained optimization. Our claims about the optimistic gradient method are restricted to: ① its role in maximizing the dual player in a Lagrangian formulation (i.e., a linear player in a non-convex–linear min-max problem), and ② its application *only* to that player; for the non-convex primal player, we generally assume a different method is used.

## 2 BACKGROUND

### 2.1 CONSTRAINED OPTIMIZATION

Let $f : \mathbb{R}^d \to \mathbb{R}$, $\boldsymbol{g} : \mathbb{R}^d \to \mathbb{R}^m$, and $\boldsymbol{h} : \mathbb{R}^d \to \mathbb{R}^n$ be twice-differentiable functions. We consider the following family of constrained optimization problems:

$$\min_{\boldsymbol{x} \in \mathbb{R}^d} f(\boldsymbol{x}) \quad \text{subject to} \quad \boldsymbol{g}(\boldsymbol{x}) \preceq \boldsymbol{0}, \quad \boldsymbol{h}(\boldsymbol{x}) = \boldsymbol{0}, \tag{CMP}$$

where $\preceq$ denotes element-wise inequality. We refer to $f$ as the objective function, and to $\boldsymbol{g}$ and $\boldsymbol{h}$ as the inequality and equality constraints, respectively.

**Definitions**. A point $\boldsymbol{x}$ is *feasible* if it satisfies all constraints. A point $\boldsymbol{x}^*$ is a *local constrained minimizer* if it is feasible and there exists an $\epsilon > 0$ such that $f(\boldsymbol{x}^*) \leq f(\boldsymbol{x})$ for all feasible $\boldsymbol{x}$ with $\|\boldsymbol{x} - \boldsymbol{x}^*\| \leq \epsilon$. A *global constrained minimizer* is a feasible point that satisfies this inequality for all feasible $\boldsymbol{x}$. An inequality constraint $g_i(\boldsymbol{x}) \leq 0$ is *active* at $\boldsymbol{x}$ if it holds with equality, i.e., $g_i(\boldsymbol{x}) = 0$. We denote the set of active inequality constraint indices at $\boldsymbol{x}$ by $\mathcal{A}_{\boldsymbol{x}} = \{i \mid g_i(\boldsymbol{x}) = 0\}$. Equality constraints are always considered "active" when satisfied.

**Solving constrained optimization problems**. A general approach to solving differentiable constrained optimization problems is to find min-max points of their associated Lagrangian (Arrow et al., 1958), which correspond to solutions of the original problem (Bertsekas, 2016, Props. 4.3.1 & 4.3.2):

$$\boldsymbol{x}^*, \boldsymbol{\lambda}^*, \boldsymbol{\mu}^* \in \arg \min_{\boldsymbol{x} \in \mathbb{R}^d} \max_{\boldsymbol{\lambda} \succeq \boldsymbol{0}, \boldsymbol{\mu}} \mathcal{L}(\boldsymbol{x}, \boldsymbol{\lambda}, \boldsymbol{\mu}) \triangleq f(\boldsymbol{x}) + \boldsymbol{\lambda}^\top \boldsymbol{g}(\boldsymbol{x}) + \boldsymbol{\mu}^\top \boldsymbol{h}(\boldsymbol{x}), \tag{Lag}$$

where $\boldsymbol{\lambda} \succeq \boldsymbol{0}$ and $\boldsymbol{\mu}$ are the Lagrange multipliers associated with the inequality and equality constraints, respectively. We refer to $\boldsymbol{x}$ as the *primal* variables, and to $\boldsymbol{\lambda}$ and $\boldsymbol{\mu}$ as the *dual* variables.

A simple algorithm for finding min-max points of Eq. (Lag) is alternating[1] (projected) gradient descent-ascent (GDA): descent on $\boldsymbol{x}$ and (projected) ascent on $\boldsymbol{\lambda}$ and $\boldsymbol{\mu}$:

$$\boldsymbol{\mu}_{t+1} \leftarrow \boldsymbol{\mu}_t + \eta_{\text{dual}} \boldsymbol{h}(\boldsymbol{x}_t), \qquad \boldsymbol{\lambda}_{t+1} \leftarrow \left[ \boldsymbol{\lambda}_t + \eta_{\text{dual}} \boldsymbol{g}(\boldsymbol{x}_t) \right]_+,$$

$$\boldsymbol{x}_{t+1} \leftarrow \boldsymbol{x}_t - \eta_{\boldsymbol{x}} \left[ \nabla f(\boldsymbol{x}_t) + \boldsymbol{\lambda}_{t+1}^\top \nabla \boldsymbol{g}(\boldsymbol{x}_t) + \boldsymbol{\mu}_{t+1}^\top \nabla \boldsymbol{h}(\boldsymbol{x}_t) \right], \tag{Lag-GDA}$$

---

[1]Alternating updates are often preferred over simultaneous updates for constrained optimization because they provide stronger convergence guarantees when the Lagrangian is strongly convex (Zhang et al., 2022), without adding computational overhead (Sohrabi et al., 2024).

where $[\cdot]_+$ denotes projection onto the non-negative orthant to enforce $\boldsymbol{\lambda} \succeq \mathbf{0}$, and $\eta_{\boldsymbol{x}}, \eta_{\text{dual}} > 0$ are the primal and dual step-sizes. The primal update direction is a linear combination of the objective and constraint gradients, weighted by the current multipliers, which are typically initialized to zero. In practice, the descent step is often replaced with adaptive first-order schemes such as Adam. We refer to Eq. (Lag-GDA) as the *first-order Lagrangian approach*, or simply the Lagrangian approach.

**Constrained deep learning**. Applying constraints to deep neural networks—where the parameters $\boldsymbol{x}$ can number in the billions—is challenging due to the non-convexity and scale of the resulting optimization problems. Classical approaches that exploit problem structure or require costly operations, such as methods involving projections (Goldstein, 1964; Levitin and Polyak, 1966), feasible directions (Frank and Wolfe, 1956; Zoutendijk, 1960), or standard SQPs (Han, 1977; Powell, 1978; Wilson, 1963), are either inapplicable to non-convex deep learning or impractical at this scale.

By contrast, the Lagrangian approach in Eq. (Lag-GDA) has become a dominant method for constrained deep learning, with applications across a wide range of domains (Cotter et al., 2019a; Elenter et al., 2022; Hounie et al., 2023a;b; Ramirez et al., 2025b; Robey et al., 2021; Zhang et al., 2025), and integration into popular frameworks such as PyTorch (Paszke et al., 2019) and TensorFlow (Abadi et al., 2016) via libraries like Cooper (Gallego-Posada et al., 2025) and TFCO (Cotter et al., 2019b).

Its popularity stems from its convergence properties and computational scalability. It provides local convergence guarantees to min–max points of the Lagrangian, even in the non-convex setting (Bertsekas, 2016, Prop. 5.4.2), while remaining computationally efficient (Gallego-Posada, 2024; Ramirez et al., 2025a). Its overhead relative to unconstrained deep learning is negligible, limited to storing and updating the multipliers.[2]

**Limitations**. In the non-convex regime, the Lagrangian approach faces several drawbacks. Not all local constrained minimizers $\boldsymbol{x}^*$ of Eq. (CMP)—together with their corresponding optimal multipliers $\boldsymbol{\lambda}^* \succeq \mathbf{0}$ and $\boldsymbol{\mu}^*$—necessarily correspond to min-max points of the Lagrangian $\mathcal{L}$. In particular, $\boldsymbol{x}^*$ need not minimize $\mathcal{L}(\cdot, \boldsymbol{\lambda}^*, \boldsymbol{\mu}^*)$,[3] and the Hessian $\nabla_{\boldsymbol{x}}^2 \mathcal{L}(\boldsymbol{x}^*, \boldsymbol{\lambda}^*, \boldsymbol{\mu}^*)$ may fail to be positive definite. In fact, the Lagrangian can be *strictly concave* in some infeasible directions—whereas strict concavity precludes optimality in unconstrained problems, it can arise at constrained optima (Bertsekas, 2016, Prop. 4.3.2). Such non-convex stationary points cannot be reached via Eq. (Lag-GDA).

Despite this limitation, Eq. (Lag-GDA) has proven effective in practice for a wide range of non-convex constrained deep learning applications. However, another challenge limits its applicability: the optimization dynamics often induce oscillations in the multipliers and their associated constraints, causing the primal iterates to repeatedly move in and out of the feasible set (Platt and Barr, 1987). These oscillations slow convergence and can be critical in settings where infeasibility cannot be tolerated—for example, when violations disrupt physical systems or compromise safety.

These limitations have motivated two main strategies for improvement. The first is the *Augmented Lagrangian Method*, which makes the (Augmented) Lagrangian *strictly convex* at all strict and regular constrained minimizers of Eq. (CMP). The second replaces simple gradient ascent on the multipliers with *generalized optimistic gradient ascent*, which mitigates oscillations in the dynamics. These methods are discussed in §2.3 and §2.2, respectively. In §3, we prove that they are, in fact, equivalent.

## 2.2 DUAL OPTIMISTIC ASCENT

The oscillatory behavior inherent in the dynamics of Eq. (Lag-GDA) has motivated alternatives to plain gradient ascent for updating the dual variables. A prominent class of such methods is based on *optimism* (Rakhlin and Sridharan, 2013). In this work, we adopt the generalized optimistic gradient method of Mokhtari et al. (2020), applying it solely to $\boldsymbol{\lambda}$ and $\boldsymbol{\mu}$. Combining this with standard gradient descent on $\boldsymbol{x}$ and alternating updates—where the multipliers are updated first—yields:

$$\boldsymbol{\mu}_{t+1} \leftarrow \boldsymbol{\mu}_t + \eta_{\text{dual}}\, \boldsymbol{h}(\boldsymbol{x}_t) + \omega\big[\boldsymbol{h}(\boldsymbol{x}_t) - \boldsymbol{h}(\boldsymbol{x}_{t-1})\big],$$

$$\boldsymbol{\lambda}_{t+1} \leftarrow \Big[\boldsymbol{\lambda}_t + \eta_{\text{dual}}\, \boldsymbol{g}(\boldsymbol{x}_t) + \omega\big[\boldsymbol{g}(\boldsymbol{x}_t) - \boldsymbol{g}(\boldsymbol{x}_{t-1})\big]\Big]_+, \qquad \text{(Lag-GD-OA)}$$

$$\boldsymbol{x}_{t+1} \leftarrow \boldsymbol{x}_t - \eta_{\boldsymbol{x}} \big[\nabla f(\boldsymbol{x}_t) + \boldsymbol{\lambda}_{t+1}^\top \nabla \boldsymbol{g}(\boldsymbol{x}_t) + \boldsymbol{\mu}_{t+1}^\top \nabla \boldsymbol{h}(\boldsymbol{x}_t)\big]$$

---

[2] This overhead is negligible for two reasons: the number of constraints—and thus multipliers—is typically much smaller than the number of model parameters, adding minimal *memory* overhead; and the dual gradients correspond to constraint values, requiring no extra backward passes and thus negligible *computational* overhead.

[3] Even solutions that minimize $\mathcal{L}$—albeit not strictly—may not correspond to limit points of Eq. (Lag-GDA).

where $\omega > 0$ is the optimism coefficient. To avoid undefined evaluations at $t = 0$, we set the "previous" constraint violations equal to the current ones—i.e., $\boldsymbol{h}(\boldsymbol{x}_{-1}) = \boldsymbol{h}(\boldsymbol{x}_0)$ and $\boldsymbol{g}(\boldsymbol{x}_{-1}) = \boldsymbol{g}(\boldsymbol{x}_0)$. Under this convention, the first dual update reduces to a standard (projected) gradient ascent step with step-size $\eta_{\text{dual}}$, ensuring that Eq. (Lag-GD-OA) is well defined at $t = 0$.[4]

While optimism is often applied to both players in general min-max optimization (i.e., optimistic gradient descent–ascent; OGDA), we follow the more general approach of Sohrabi et al. (2024); Stooke et al. (2020)[5] for constrained problems by applying optimism only to the dual variables. This preserves flexibility in the choice of the primal update, allowing to retain task-specific training pipelines—such as standard Adam without optimism. Such flexibility is especially important in deep learning, where model training pipelines are often highly specialized, making it undesirable to modify them solely to accommodate alternative dual optimization algorithms.

Dual (generalized) optimistic updates have been applied to stabilize optimization dynamics of constrained deep learning problems across reinforcement learning (Moskovitz et al., 2023; Stooke et al., 2020), unsupervised learning (Shao et al., 2022), and supervised learning (Sohrabi et al., 2024).

Intuitively, the optimistic update in Eq. (Lag-GD-OA) dampens oscillations by adjusting dual ascent steps with information from past constraint violations; specifically, by adding the optimistic term $\omega\big(\boldsymbol{g}(\boldsymbol{x}_t) - \boldsymbol{g}(\boldsymbol{x}_{t-1})\big)$. When violations shrink across iterations, this correction term is negative, acting as a brake: the multiplier increases more conservatively, avoiding an "overshoot" beyond its optimum. This prevents over-penalization of the constraints in later primal steps, thereby stabilizing the dynamics and reducing oscillations. For further discussion, see Sohrabi et al. (2024, §4.3). In §3, we show that this stabilizing behavior is not accidental: in the single-step, first-order regime we consider, dual optimistic ascent is simply an Augmented Lagrangian method in disguise.

**Literature gaps**. While successful empirically, dual optimistic ascent on the Lagrangian remains largely underformalized. It lacks convergence guarantees and is supported exclusively by empirical demonstrations of its ability to mitigate oscillations.

Despite the vast literature on optimistic methods, most existing results do not directly apply to the dual optimistic ascent algorithm in Eq. (Lag-GD-OA). For general min-max games, OGDA is known to expand the set of stationary points to which it can converge relative to standard GDA (Daskalakis and Panageas, 2018). While this set can include the global solution in bilinear games, the nature of these additional points remains largely uncharacterized in general non-convex–non-concave games. Moreover, the established convergence guarantees for OGDA—such as global linear convergence under strong convexity–strong concavity (Gidel et al., 2019), sublinear convergence under convexity–concavity (Gorbunov et al., 2022), and sublinear convergence for non-convex–(strongly) concave settings (Mahdavinia et al., 2022)—are derived in frameworks incompatible with our own.

The inapplicability of these findings stems from two key issues. First, the cited results concern a different algorithm that applies optimism to *both* players, not just the dual player as in Eq. (Lag-GD-OA).[6] Second, the underlying assumptions are mismatched—they are either too restrictive, requiring (strong) convexity for the minimization player, or too general, assuming a non-convex–concave game that fails to exploit the specific non-convex–*linear* structure of the Lagrangian.

## 2.3 THE AUGMENTED LAGRANGIAN METHOD

The (HPR) Augmented Lagrangian function (Hestenes, 1969; Powell, 1969; Rockafellar, 1973) adds a quadratic penalty to the standard Lagrangian to penalize constraint violations:

$$\mathcal{L}_c(\boldsymbol{x}, \boldsymbol{\lambda}, \boldsymbol{\mu}) \triangleq f(\boldsymbol{x}) + \frac{1}{2c}\left[\|\boldsymbol{\mu} + c\,\boldsymbol{h}(\boldsymbol{x})\|_2^2 - \|\boldsymbol{\mu}\|_2^2 + \left\|[\boldsymbol{\lambda} + c\,\boldsymbol{g}(\boldsymbol{x})]_+\right\|_2^2 - \|\boldsymbol{\lambda}\|_2^2\right] \tag{AL}$$

where $c > 0$ is a penalty coefficient. Expanding the squared norms in Eq. (AL) reveals that constraint violations are penalized by both a linear term involving the multiplier and a quadratic term scaled by $c$.

---

[4]Alternatively, one can set $\boldsymbol{h}(\boldsymbol{x}_{-1}) = \boldsymbol{0}$ and $\boldsymbol{g}(\boldsymbol{x}_{-1}) = \boldsymbol{0}$, in which case the first dual step is a projected gradient ascent update with effective step size $\eta_{\text{dual}} + \omega$. Both conventions are equivalent for our purposes—the difference can be absorbed into the initialization of $(\boldsymbol{\lambda}_0, \boldsymbol{\mu}_0)$ without affecting our theoretical results.

[5]The generalized optimistic gradient method is a special case of the algorithms in both papers; specifically, it corresponds to a Proportional-Integral (PI) controller on the multipliers.

[6]Moreover, known convergence results for OGDA seldom consider the *generalized* case where $\omega \neq \eta_{\text{dual}}$, with the notable exception of Mokhtari et al. (2020) for bilinear games.

The projection $[\cdot]_+$ creates a *one-sided* penalty for inequalities: it penalizes violations ($g_i(\boldsymbol{x}) > 0$) but avoids pushing iterates deeper into the feasible region once a constraint is sufficiently satisfied ($g_i(\boldsymbol{x}) \leq -\lambda_i/c$). This ensures that optimization focuses only on violated or nearly-active constraints.

**The set of min-max points of $\mathcal{L}_c$.** The Augmented Lagrangian preserves and enhances the solution set of the standard Lagrangian. Since the two functions share the same set of stationary points (Rockafellar, 1973, Cor. 3.4), they have an identical pool of candidate solutions. Furthermore, every local min-max point of $\mathcal{L}$ remains a local min-max point of $\mathcal{L}_c$, ensuring that any solution attainable with the standard Lagrangian approach remains attainable within the Augmented Lagrangian framework.

More importantly, under regularity assumptions, *every* strict local constrained minimizer $\boldsymbol{x}^*$ admits a threshold $\bar{c} \geq 0$ such that, for all $c \geq \bar{c}$, the Augmented Lagrangian $\mathcal{L}_c$ is *strictly convex* at $\boldsymbol{x}^*$, even if the original Lagrangian $\mathcal{L}$ is not convex there (Bertsekas, 2016, §4.2.1). Consequently, all strict local constrained minimizers of Eq. (CMP) are upgraded to local min-max points of $\mathcal{L}_c$.

Two important algorithmic consequences follow: ① *all* strict local constrained minimizers become strict local min-max points of $\mathcal{L}_c$, for which GDA enjoys local linear convergence guarantees (Bertsekas, 2016, Prop. 5.4.2); and ② the sets of strict local constrained minimizers of Eq. (CMP) and strict local min-max points of $\mathcal{L}_c$ coincide, ensuring that GDA on the Augmented Lagrangian converges to—and *only* to—stationary points corresponding to solutions of Eq. (CMP). Further benefits of the Augmented Lagrangian approach—including its ability to mitigate oscillations—are discussed in §4.

**The Augmented Lagrangian Method** seeks min-max points of the Augmented Lagrangian function:

$$\boldsymbol{x}^*, \boldsymbol{\lambda}^*, \boldsymbol{\mu}^* \in \arg \min_{\boldsymbol{x} \in \mathbb{R}^d} \max_{\boldsymbol{\lambda} \succeq \boldsymbol{0}, \boldsymbol{\mu}} \mathcal{L}_c(\boldsymbol{x}, \boldsymbol{\lambda}, \boldsymbol{\mu}). \tag{1}$$

The classic algorithm for solving this problem—the Method of Multipliers—iteratively performs:

$$\boldsymbol{x}_{t+1} \in \arg \min_{\boldsymbol{x} \in \mathbb{R}^d} \mathcal{L}_{c_t}(\boldsymbol{x}, \boldsymbol{\lambda}_t, \boldsymbol{\mu}_t), \quad \boldsymbol{\mu}_{t+1} = \boldsymbol{\mu}_t + c_t\, \boldsymbol{h}(\boldsymbol{x}_{t+1}), \quad \boldsymbol{\lambda}_{t+1} = [\boldsymbol{\lambda}_t + c_t\, \boldsymbol{g}(\boldsymbol{x}_{t+1})]_+ . \tag{2}$$

In this method, the primal minimization is typically performed approximately, either for a fixed number of optimization steps or until a numerical tolerance is reached. The subsequent dual updates correspond to gradient ascent steps on the standard Lagrangian $\mathcal{L}$ with step-sizes $c_t$. The penalty coefficients are chosen as a non-decreasing sequence $c_{t+1} \geq c_t > 0$, often updated according to a schedule—for example, increasing $c_t$ when constraint violations are not reduced sufficiently across successive iterations (see, e.g., Birgin and Martínez (2014, Eq. 4.9 in Alg. 4.1)).

The Augmented Lagrangian method is a seminal approach for solving non-convex constrained optimization problems, with variants implemented in numerous optimization libraries across different programming languages (Conn et al., 2013; Gallego-Posada et al., 2025; Johnson; Pas et al., 2022). Its success spans a broad range of scientific problems, and it has become a key tool in constrained deep learning, applied in works ranging from seminal contributions to recent advances (Brouillard et al., 2020; Kotary and Fioretto, 2024; Lokhande et al., 2020; Platt and Barr, 1987; Shi et al., 2023).

In deep learning, where full (or even approximate) minimization of $\mathcal{L}_c$ is computationally infeasible, it is often replaced by one or more steps of a first-order optimizer, such as Adam. Accordingly, in this paper we consider the following family of Augmented Lagrangian algorithms: alternating gradient descent–(projected) gradient ascent on $\mathcal{L}_c$, with the primal update performed first.

$$\boldsymbol{x}_{t+1} \leftarrow \boldsymbol{x}_t - \eta_{\boldsymbol{x}} \left[ \nabla f(\boldsymbol{x}_t) + [\boldsymbol{\lambda}_t + c\, \boldsymbol{g}(\boldsymbol{x}_t)]_+^\top \nabla \boldsymbol{g}(\boldsymbol{x}_t) + (\boldsymbol{\mu}_t + c\, \boldsymbol{h}(\boldsymbol{x}_t))^\top \nabla \boldsymbol{h}(\boldsymbol{x}_t) \right]$$

$$\boldsymbol{\mu}_{t+1} \leftarrow \boldsymbol{\mu}_t + \eta_{\text{dual}}\, \boldsymbol{h}(\boldsymbol{x}_{t+1}) \qquad \boldsymbol{\lambda}_{t+1} \leftarrow \left(1 - \tfrac{\eta_{\text{dual}}}{c}\right) \boldsymbol{\lambda}_t + \tfrac{\eta_{\text{dual}}}{c} \left[\boldsymbol{\lambda}_t + c\, \boldsymbol{g}(\boldsymbol{x}_{t+1})\right]_+ \tag{AL-GDA}$$

where $0 < \eta_{\text{dual}} \leq c$ and $\eta_{\boldsymbol{x}} > 0$ are fixed step-sizes. The algorithm first takes a single primal gradient descent step on $\mathcal{L}_c$, then performs a dual gradient ascent step on the multipliers with step-size $\eta_{\text{dual}}$, which is independent of the penalty $c$. Notably, the $\boldsymbol{\lambda}$-update is a convex combination of the previous iterate $\boldsymbol{\lambda}_t$ and a new estimate $[\boldsymbol{\lambda}_t + c\, \boldsymbol{g}(\boldsymbol{x}_{t+1})]_+$. Setting $\eta_{\text{dual}} = c$ recovers the standard dual updates of the Method of Multipliers in Eq. (2). This method is therefore more specific than Eq. (2) by replacing the full primal minimization with a single gradient step and more general, as it decouples the dual step-size from $c$. For a full derivation of these updates, see Appendix B.

The primal update in Eq. (AL-GDA) can be viewed as a gradient descent step on the standard Lagrangian, using multiplier estimates that differ from their latest iterates. These estimates simulate a (projected) ascent step on $\mathcal{L}$, acting as a proactive "lookahead" to calibrate the primal updates. Recognizing this is the key insight behind the equivalence with dual optimistic ascent shown in §3.

# 3 DUAL OPTIMISTIC ASCENT (PI CONTROL) IS THE AUGMENTED LAGRANGIAN METHOD IN DISGUISE

In this section, we show that the Augmented Lagrangian method (Eq. (AL-GDA) in §2.3) and dual optimistic ascent (Eq. (Lag-GD-OA) in §2.2) are equivalent for equality-constrained problems when $c = \omega$, in the sense that their primal iterates coincide (Theorem 1 in §3.1). This equivalence does not extend to problems with inequality constraints. Nevertheless, Theorem 2 in §3.2 establishes that both algorithms share the same set of locally stable stationary points. Consequently, they achieve the same fundamental objective: ensuring that all strict and regular local constrained minimizers of Eq. (CMP) are limit points of their dynamics. We discuss further implications of these results in §4.

## 3.1 EQUALITY CONSTRAINTS

We first show that, for equality-constrained problems, the primal iterates of Eq. (Lag-GD-OA) and Eq. (AL-GDA) coincide when $\omega = c$ and the initialization is chosen appropriately.

**Theorem 1** (Equivalence for equality-constrained problems). *The primal iterates $\{\boldsymbol{x}_t\}_{t=0}^{\infty}$ generated by primal-first GDA on the Augmented Lagrangian (Eq. (AL-GDA)) and dual-first gradient descent–optimistic ascent on the Lagrangian (Eq. (Lag-GD-OA)) for an equality-constrained problem match, provided that: ① the penalty and optimism coefficients are constant and equal, $\omega = c > 0$; and ② their respective initializations are chosen as $(\boldsymbol{x}_0, \boldsymbol{\mu}_0)$ and $(\boldsymbol{x}_0, \boldsymbol{\mu}_0 + (c - \eta_{dual})\boldsymbol{h}(\boldsymbol{x}_0))$.*

*Proof.* See *Proof of Theorem 1* in Appendix D. □

Note that Theorem 1 applies to *any* first-order optimization algorithm used to minimize the (Augmented) Lagrangian, not only gradient descent. This holds because both algorithms generate the same sequence of primal gradients. In particular, the result extends to primal updates performed with first-order deep learning optimizers such as Adam.

## 3.2 GENERAL CONSTRAINED PROBLEMS WITH INEQUALITY CONSTRAINTS

Unlike the equality-constrained case, the iterates of Eq. (AL-GDA) and Eq. (Lag-GD-OA) do not coincide for problems with inequality constraints. This difference arises from the placement of projections on the inequality multipliers updates: in Eq. (Lag-GD-OA), they are applied only once during the update, whereas in Eq. (AL-GDA), they are applied twice—once to update the multiplier and again to compute the "lookahead" estimate for the primal gradient. Nevertheless, as shown in Theorem 2, the two algorithms share the same set of locally stable stationary points. In this sense, both are equally powerful—and strictly more so than Eq. (Lag-GDA).

First, we establish that both algorithms share the exact same set of stationary (fixed) points:

**Proposition 1** (Equivalence of Stationary Points). *The set of fixed points for algorithms Eqs. (AL-GDA) and (Lag-GD-OA) are the same, and correspond to the set of KKT points of Eq. (CMP).*

*Proof.* See *Proof of Proposition 1* in Appendix D. □

Having identified the fixed points, we now examine whether the algorithms actually converge to them. We rely on the standard notion of local stability:

**Definition 1** (Local stability). *Consider an algorithm*

$$\boldsymbol{x}_{t+1} \leftarrow F(\boldsymbol{x}_t, \boldsymbol{\lambda}_t, \boldsymbol{\mu}_t), \qquad \boldsymbol{\lambda}_{t+1} \leftarrow G(\boldsymbol{x}_t, \boldsymbol{\lambda}_t, \boldsymbol{\mu}_t), \qquad \boldsymbol{\mu}_{t+1} \leftarrow H(\boldsymbol{x}_t, \boldsymbol{\lambda}_t, \boldsymbol{\mu}_t). \tag{3}$$

*A fixed point $(\boldsymbol{x}^*, \boldsymbol{\lambda}^*, \boldsymbol{\mu}^*)$ of this algorithm is called a locally stable stationary point (LSSP) if the Jacobian $\mathcal{J}$ of the joint operator $[F, G, H]$, evaluated at $(\boldsymbol{x}^*, \boldsymbol{\lambda}^*, \boldsymbol{\mu}^*)$, has spectral radius strictly less than one, i.e., $\rho(\mathcal{J}) < 1$.*

A standard result states that an algorithm exhibits local linear convergence to all of its LSSPs (Bertsekas, 2016, Prop. 5.4.1). To analyze the stability of our two methods, we therefore study their respective operator Jacobians: $\mathcal{J}_{AL}$ for the Augmented Lagrangian method Eq. (AL-GDA) and $\mathcal{J}_{OG}$ for dual optimistic ascent Eq. (Lag-GD-OA). These are derived in Lemmas 2 and 4 of Appendix D.

**Theorem 2** (Equivalence of LSSPs - inequality constraints). *Let $(\boldsymbol{x}^*, \boldsymbol{\lambda}^*)$ be a stationary point of the Lagrangian for an inequality-constrained problem, satisfying strict complementary slackness (see Assumption 1 in Appendix A). Then Eq. (AL-GDA) with penalty coefficient $c$ converges locally to $(\boldsymbol{x}^*, \boldsymbol{\lambda}^*)$ if and only if Eq. (Lag-GD-OA) with optimism coefficient $\omega = c$ also converges locally to that point. Moreover, the spectral radii of the Jacobians of the two algorithms satisfy*

$$\rho(\mathcal{J}_{AL}) = \max\{\rho(\mathcal{J}_{OG}), 1 - \eta_{dual}/c\}. \tag{4}$$

*Proof.* See *Proof of Theorem 2* in Appendix D. □

By combining the results for equalities (§3.1) and inequalities (Theorem 2), we arrive at the following general equivalence covering mixed constraints:

**Corollary 1** (Equivalence of LSSPs). *The algorithms in Eqs. (AL-GDA) and (Lag-GD-OA) share the same set of locally stable stationary points satisfying Assumption 1.*

*Proof.* The result follows by applying the same logic as in the proof of Theorem 2, where equality constraints are treated as a special case of active inequality constraints. □

## 4 IMPLICATIONS AND DISCUSSION

This section builds on the equivalence results from §3 to establish formal properties of the dual optimistic ascent method (PI control), including a characterization of its locally stable stationary points in terms of solutions to the constrained problem (§4.1), convergence guarantees (§4.2), and practical guidelines for tuning the optimism hyperparameter (§4.3).

These results apply specifically to the single-step, first-order dynamics of Eq. (Lag-GD-OA). In §4.4, we explain how the equivalence breaks down once these assumptions are relaxed and argue that, in such settings, classical Augmented Lagrangian methods are the more principled approach.

### 4.1 CHARACTERIZING THE STABLE POINTS OF DUAL OPTIMISTIC ASCENT

We now show that dual optimistic ascent offers a principled approach to constrained optimization: it recovers all regular, strict local constrained minimizers of Eq. (CMP). Although the method has been applied to constrained problems before, this property has not been previously identified or formalized.

Subsequent results rely on three standard assumptions, which we state formally in Appendix A: strict complementary slackness (Assumption 1), the second-order sufficiency condition (Assumption 2), and the linear independence constraint qualification (Assumption 3). We now recall the following classical results for the Augmented Lagrangian method:

**Proposition 2.** *(Bertsekas, 2016, §4.2.1) Let $\boldsymbol{x}^*$ be a strict local constrained minimizer with corresponding Lagrange multipliers $\boldsymbol{\lambda}^* \succeq \mathbf{0}$ and $\boldsymbol{\mu}^*$, satisfying Assumptions 1 to 3. Then there exists $\bar{c} > 0$ such that, for every $c \geq \bar{c}$, the Augmented Lagrangian $\mathcal{L}_c$ is strictly convex in $\boldsymbol{x}$ at $(\boldsymbol{x}^*, \boldsymbol{\lambda}^*, \boldsymbol{\mu}^*)$.*

**Proposition 3.** *(ALM finds all constrained minimizers) For any strict local constrained minimizer $\boldsymbol{x}^*$ satisfying Assumptions 1 to 3, there exist $c > 0$ large enough and $\eta_{\boldsymbol{x}}, \eta_{dual} > 0$ small enough such that $\boldsymbol{x}^*$ is an LSSP of Eq. (AL-GDA).*

*Proof.* This is well-known for simultaneous GDA in equality-constrained problems (Bertsekas (2016, Prop. 5.4.2)). For a proof specific to Eq. (AL-GDA), see *Proof of Proposition 3* in Appendix D. □

The converse also holds: the LSSPs of ALM dynamics are precisely the local constrained minimizers of Eq. (CMP), indicating that Eq. (AL-GDA) does not converge to spurious stationary points of $\mathcal{L}_c$.

**Proposition 4.** *(ALM finds only constrained minimizers) Let $(\boldsymbol{x}^*, \boldsymbol{\lambda}^*, \boldsymbol{\mu}^*)$ be an LSSP of Eq. (AL-GDA) for some $\eta_{\boldsymbol{x}}, \eta_{dual}, c > 0$. Then $\boldsymbol{x}^*$ is a local constrained minimizer of Eq. (CMP).*

*Proof.* See *Proof of Proposition 4* in Appendix D. □

We now characterize the solution set of dual optimistic ascent.

**Theorem 3** (LSSPs of dual optimistic ascent). *Let $\boldsymbol{x}^*$ satisfy Assumptions 1 and 3. Then $\boldsymbol{x}^*$ is a strict local constrained minimizer of Eq. (CMP)—that is, it satisfies Assumption 2—if and only if there exists an optimism coefficient $\bar{\omega} \geq 0$ such that for all $\omega \geq \bar{\omega}$, there exist learning rates $\eta_{\boldsymbol{x}}, \eta_{dual} > 0$ small enough for which $\boldsymbol{x}^*$ is an LSSP of the dual optimistic ascent dynamics Eq. (Lag-GD-OA).*

*Proof.* This follows directly from the equivalence results in Corollary 1 together with the properties of Eq. (AL-GDA) given in Propositions 3 and 4 above. □

Theorem 3 establishes dual optimistic ascent as a principled framework for constrained optimization. While it is known that optimistic methods converge to a broader set of stable points than standard GDA (Daskalakis and Panageas, 2018), our result refines this for the constrained setting by precisely characterizing *which* additional stable points arise from (dual) optimism: it recovers all strict and regular local constrained minimizers of Eq. (CMP) and *only* such solutions.[7] This makes it strictly more powerful than simple Eq. (Lag-GDA). For an analysis of how large $\omega = c$ must be to guarantee convergence to a given $\boldsymbol{x}^*$, see Bertsekas (2014, Prop. 2.7).

### 4.2 Convergence Results for Dual Optimistic Ascent

**Non-convex problems**. Our equivalence results allow us to transfer the well-understood convergence guarantees of the Augmented Lagrangian method to the dual optimistic ascent framework.

**Corollary 2** (Local convergence rate of dual optimistic ascent). *For appropriate hyperparameter choices, Eq. (Lag-GD-OA) exhibits local linear convergence to all strict local constrained minimizers of Eq. (CMP) satisfying Assumptions 1 to 3. Whenever $\eta_{dual}$ is sufficiently close to $\omega = c$ (so that $1 - \eta_{dual}/c < \rho(\mathcal{J}_{AL})$), the convergence rate of Eq. (Lag-GD-OA) matches that of Eq. (AL-GDA).*

*Proof.* Local linear convergence follows directly from Theorem 3. The spectral norm relationship between $\rho(\mathcal{J}_{OG})$ and $\rho(\mathcal{J}_{AL})$ established in Theorem 2 links the convergence rates of the two algorithms for inequality-constrained problems. Note that equality constraints can be treated locally as active inequalities, and thus the same relationship between spectral norms holds in that case. □

This result also implies that for problems where a specific convergence rate is known for Eq. (Lag-GDA),[8] that rate transfers directly to Eq. (Lag-GD-OA), provided $\eta_{dual}$ is sufficiently close to $c$.

Global convergence, however, is more delicate. While the matching primal iterates for equality-constrained problems (Theorem 1) suggest that convergence guarantees may transfer directly across algorithms, this is not the case for inequality constraints. There, our equivalence is only local—ensuring the same set of LSSPs (Corollary 1) but not the same global behavior.

Even for equality constraints, a key obstacle prevents us from proving global convergence of dual optimistic ascent using known results for the Augmented Lagrangian method. Classic ALM results, such as the Q-linear rate in Bertsekas (2014, Prop. 2.7), apply to the Method of Multipliers (Eq. (2)) and rely on each primal step yielding a *sufficiently accurate (local) minimizer* of $\mathcal{L}_c$. In contrast, our equivalence holds only for *single-step first-order* updates, which do not meet this requirement.

One might then consider performing multiple primal minimization steps per dual optimistic ascent update to meet this condition. However, this breaks the equivalence to Eq. (AL-GDA) and instead amounts to a multi-step minimization of the standard Lagrangian $\mathcal{L}$. Since $\mathcal{L}$ may fail to be locally convex near a solution, such a procedure lacks convergence guarantees.

It may still be possible to establish global convergence of Eq. (Lag-GD-OA) for non-convex problems directly—following, for example, strategies developed for OGDA in nonconvex–concave games (Mahdavinia et al., 2022). However, this lies beyond the scope of our work.

**Convex problems.** In the convex setting, our equivalence theorems yield global convergence guarantees for dual optimistic ascent on equality-constrained problems. Extending these guarantees to inequality constraints remains an open direction for future work.

**Corollary 3** (Global linear convergence for convex equality-constrained problems). *Consider a problem with a convex, smooth objective $f(\boldsymbol{x})$ and affine equality constraints $\boldsymbol{h}(\boldsymbol{x}) = B\boldsymbol{x} + b$. Assume the problem has a unique solution $\boldsymbol{x}^*$ and that LICQ is satisfied (Assumption 3), i.e., $B$ has full row rank. Then, for any $\omega > 0$ and sufficiently small step-sizes $\eta_{dual}, \eta_{\boldsymbol{x}}$, Eq. (Lag-GD-OA) converges globally at a linear rate to the optimal pair $(\boldsymbol{x}^*, \boldsymbol{\mu}^*)$. The exact rate depends on the properties of $f$ and $B$; see Alghunaim and Sayed (2020, Theorem 1) for a precise characterization.*

---

[7]This highlights a key difference with general min-max games (e.g., GANs), where new equilibria stabilized by optimism can be spurious. In constrained optimization, these "spurious" solutions are a feature, not a bug.

[8]There is no general closed-form expression for the iteration complexity of the Augmented Lagrangian method in Eq. (AL-GDA); such rates can only be established under additional assumptions on $f$, $\boldsymbol{g}$, and $\boldsymbol{h}$.

*Proof.* This follows from the equivalence in Theorem 1, and Alghunaim and Sayed (2020, Theorem 1), which establishes global linear convergence for Eq. (AL-GDA). ☐

This global convergence guarantee hinges on the strong convexity of the Augmented Lagrangian $\mathcal{L}_c$ on $x$, which is induced even when the original objective $f$ is merely convex. In contrast, the standard Lagrangian remains only convex in $x$, and GDA is not guaranteed to converge on the resulting convex–concave problem (Gidel et al., 2019). Dual optimistic ascent avoids this failure mode by inheriting the favorable convergence properties of the Augmented Lagrangian method.

### 4.3 PRACTICAL IMPLICATIONS FOR HYPER-PARAMETER TUNING

Our equivalence result provides practical guidance on choosing the optimism hyper-parameter $\omega$. This parameter introduces a critical trade-off analogous to the coefficient $c$ in the Augmented Lagrangian method: larger values of $\omega$ are beneficial as they ① expand the set of attainable solutions and ② dampen oscillations in the optimization dynamics. However, ③ excessively large values can lead to an ill-conditioned problem, which may in turn slow convergence. We now formalize these points.

**Corollary 4** (Monotonic inclusion of solutions). *The set of strict local minimizers of Eq. (CMP) that are LSSPs of the dual optimistic ascent dynamics is monotonically non-decreasing in $\omega$. For sufficiently large $\omega$, this set coincides with all local minimizers satisfying Assumptions 1 to 3.*

*Proof.* This follows from the equivalence result in Corollary 1 and Prop. 3. ☐

In practice, however, it is generally unclear what value of $\omega$ is sufficiently large to turn a given solution $x^*$ into an LSSP; this would typically require knowing $x^*$ itself. This practical uncertainty motivates a simple strategy: to ensure all solutions are attainable, one may consider taking $\omega \to \infty$.

**Dampening Oscillations.** A key motivation for moving beyond standard GDA on the Lagrangian $\mathcal{L}$ is the presence of oscillations. Whenever the Hessian $\nabla_x^2 \mathcal{L}$ is indefinite, the dynamics are characterized by imaginary eigenvalues, an effect observed in practice (Sohrabi et al., 2024; Stooke et al., 2020) and expected in theory (Benzi and Simoncini, 2006, Prop 2.5 & Remark 2.8).

The Augmented Lagrangian method is a classical solution: as the penalty coefficient $c \to \infty$, the eigenvalues of its dynamics become purely real, eliminating oscillations (Benzi and Simoncini, 2006, Remark 2.9). Our equivalence result (Corollary 1) shows that Eq. (Lag-GD-OA) inherits this damping property, formalizing the insight that larger values of $\omega$ produce a more heavily dampened system.

**Proposition 5** (Dual optimistic ascent dampens oscillations). *Let $(x^*, \lambda^*)$ be a strict local constrained minimizer satisfying Assumptions 1 to 3. There exists a finite threshold $\bar{\omega} \geq 0$ such that for all optimism coefficients $\omega \geq \bar{\omega}$, the eigenvalues of the Jacobian $\mathcal{J}_{OG}$ are purely real. Furthermore, under non-degeneracy conditions, as $\omega$ approaches $\bar{\omega}$ from below, the maximum imaginary part of the eigenvalues decays as $\mathcal{O}(\sqrt{\bar{\omega} - \omega})$.*

*Proof.* See *Proof of Proposition 5* in Appendix D. ☐

As before, the exact threshold for $\omega$ to fully remove oscillations depends on problem-specific properties at $x^*$, making it unknown in practice. This again motivates selecting a generously large $\omega$.

**Conditioning trade-off**. However, while these observations may suggest that increasing the optimism coefficient is beneficial, we emphasize another aspect: Bertsekas (2014, Eq. 16 in §2.1) shows that as $c \to \infty$, $\mathcal{J}_{AL}$—and hence $\mathcal{J}_{OG}$—becomes increasingly ill-conditioned.

**Corollary 5** (Ill-conditioning of dual optimistic ascent). *As $\omega \to \infty$, the condition number of the operator Jacobian $\mathcal{J}_{OG}$ also tends to infinity.*

*Proof.* This follows from our equivalence result (Corollary 1) and the known ill-conditioning of the Augmented Lagrangian for a large penalty $c$ (Bertsekas, 2014, see Eq. 16 in §2.1). ☐

This reveals a critical trade-off: the optimism coefficient $\omega$ must be large enough to recover all solutions, yet not so large that it induces ill-conditioning. Since the optimal value depends on local properties at an unknown solution $x^*$ and may not generalize across the optimization landscape, $\omega$ must be tuned in practice. Our equivalence provides a principled path forward: practitioners can use well-established ALM penalty-scheduling techniques to dynamically tune the optimism coefficient.

### 4.4 WHEN THE DISGUISE FAILS: LIMITS OF DUAL OPTIMISM

While Eq. (Lag-GD-OA) and Eq. (AL-GDA) are equivalent, this equivalence relies critically on their single-step, first-order structure. The core mechanism is that dual optimistic ascent effectively performs descent steps on the Augmented Lagrangian (see §2.3), providing a principled basis for solving constrained optimization problems.

Once these assumptions are relaxed, this mechanism breaks down. Performing multiple primal steps between dual updates means the equivalence holds only for the *first* step; from the second step onward, the algorithm minimizes the standard Lagrangian with frozen multipliers. As discussed in §2.1, this is flawed for non-convex constrained optimization: the Lagrangian need not be locally convex in a neighborhood of a constrained minimizer, so this inner minimization may fail to converge to the desired solution.

The situation is similar for second-order primal methods. Our change of variables aligns the *gradients* of the two objectives, but not their *Hessians*: the Augmented Lagrangian includes extra positive curvature from the penalty term that the standard Lagrangian lacks. Consequently, Newton or quasi-Newton steps on the standard Lagrangian—even with optimistic multipliers—differ from those of ALM and do not inherit its local convexity or convergence guarantees.

In both cases, the algorithm effectively reverts to minimizing the standard Lagrangian, with all its associated stability issues and without the guarantees obtained via the ALM equivalence. In such settings, we therefore advocate for the explicit Augmented Lagrangian method, which enjoys Q-linear convergence when the subproblems are solved to sufficient accuracy (Bertsekas, 2014, Prop. 2.7).

## 5 EMPIRICAL VALIDATION

We provide empirical support for our theoretical findings in Appendix C. First, we numerically verify the exact equivalence of primal iterates for equality constraints (Theorem 1) on a non-convex 1D example. As illustrated in Figure 1 (Appendix C.1), the trajectories of dual optimistic ascent and the Augmented Lagrangian method coincide. Second, we demonstrate the practical utility of this equivalence in Appendix C.2 by showing that classical ALM penalty scheduling strategies can be directly applied to the optimism coefficient $\omega$ to stabilize convergence (Figure 2).

Beyond these synthetic examples, our theory rationalizes the empirical success of dual optimistic ascent reported in the deep learning literature. Specifically, the recovery of all strict local solutions (Theorem 3) explains success where standard GDA fails (Ramirez et al., 2025a, Table 1), and the spectral analysis (Prop. 5) provides a rigorous basis for the oscillation dampening observed in reinforcement learning (Stooke et al., 2020) and constrained classification (Sohrabi et al., 2024).

## 6 CONCLUSION

In this paper, we establish that dual optimistic ascent (PI control) on the Lagrangian is, in fact, the Augmented Lagrangian method in disguise. This equivalence bridges a gap between theory and practice: it allows us to transfer the formal guarantees of the ALM to the widely used dual optimistic ascent method. Furthermore, our work provides a principled interpretation of the optimism hyperparameter $\omega$, reframing it from a heuristic knob into a formal regulator of the trade-off between solution accessibility and numerical conditioning.

Crucially, our analysis also delineates the boundaries of dual optimistic ascent. We show that it is theoretically well-founded *only* in the single-step, first-order regime common in deep learning. Once one departs from this regime—for instance, by employing multi-step primal updates or second-order optimizers—the equivalence vanishes. In such cases, there is no guarantee that dual optimism will recover all local constrained solutions, and the explicit Augmented Lagrangian method remains the superior, principled framework.

A natural direction for future work is to explore whether similar principles apply to optimistic methods in general min–max games, potentially offering new insights into the stabilization of GANs and other adversarial formulations. Additionally, the connection established here opens the door to transferring hyperparameter tuning techniques from control theory (PI gains) to constrained optimization (penalty schedules), promising more robust automated tuning strategies.

ACKNOWLEDGEMENTS AND DISCLOSURE OF FUNDING

This work was supported by the Canada CIFAR AI Chair program (Mila), the NSERC Discovery Grant RGPIN-2025-05123, by an unrestricted gift from Google, and by Samsung Electronics Co., Ltd. Simon Lacoste-Julien is a CIFAR Associate Fellow in the Learning in Machines & Brains program.

We thank Jose Gallego-Posada and Ioannis Mitliagkas for insightful discussions that ultimately led to this work; their observation that dual optimistic ascent and the Augmented Lagrangian method exhibit similar stabilizing effects was a key motivation for this investigation.

## STATEMENT ON THE USE OF LARGE LANGUAGE MODELS

Large Language Models (LLMs) were used to assist in the preparation of this manuscript. Their role was that of research assistants: aiding in the literature review, cross-checking proofs, and evaluating the soundness of technical arguments. They were also employed to improve the clarity of the text.

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

# Appendix

# A  ASSUMPTIONS AND FURTHER RESULTS

This section provides the formal definitions for the key assumptions used in the main text (Appendix A.1). We also present an additional result showing that applying dual optimism directly to the Augmented Lagrangian function produces a compounding o-optimistic/penalty effect, rather than giving rise to a completely new method (Appendix A.2).

## A.1  ASSUMPTIONS

**Assumption 1** (Strict complementary slackness). A stationary point $(\boldsymbol{x}^*, \boldsymbol{\lambda}^*, \boldsymbol{\mu}^*)$ of the Lagrangian $\mathcal{L}$ is said to satisfy *strict complementary slackness* if:

$$\lambda_i^* > 0 \text{ and } g_i(\boldsymbol{x}^*) = 0 \quad \text{or} \quad \lambda_i^* = 0 \text{ and } g_i(\boldsymbol{x}^*) < 0, \qquad \forall i = 1, \dots, m. \tag{5}$$

**Assumption 2** (Second-order sufficiency condition). A stationary point $(\boldsymbol{x}^*, \boldsymbol{\lambda}^*, \boldsymbol{\mu}^*)$ of $\mathcal{L}$ satisfies the *second-order sufficiency condition* if the Hessian of the Lagrangian is positive definite on the tangent space of the active constraints. That is,

$$\boldsymbol{d}^\top \nabla_{\boldsymbol{x}}^2 \mathcal{L}(\boldsymbol{x}^*, \boldsymbol{\lambda}^*, \boldsymbol{\mu}^*) \boldsymbol{d} > 0, \quad \forall \boldsymbol{d} \in \mathbb{R}^d \setminus \{\boldsymbol{0}\} \text{ such that } \nabla \boldsymbol{g}_{\mathcal{A}_{\boldsymbol{x}}}(\boldsymbol{x}^*) \boldsymbol{d} = 0 \text{ and } \nabla \boldsymbol{h}(\boldsymbol{x}^*) \boldsymbol{d} = 0. \tag{6}$$

Note that any KKT point $(\boldsymbol{x}^*, \boldsymbol{\lambda}^*, \boldsymbol{\mu}^*)$ satisfying Assumption 2 is a local constrained minimizer of Eq. (CMP) (Bertsekas, 2016, Prop. 4.3.2).

**Assumption 3** (Linear independence constraint qualification). A point $\boldsymbol{x} \in \mathbb{R}^d$ satisfies the *linear independence constraint qualification* if the matrix

$$\left[ \nabla \boldsymbol{g}_{\mathcal{A}_{\boldsymbol{x}}}(\boldsymbol{x})^\top, \nabla \boldsymbol{h}(\boldsymbol{x})^\top \right]^\top \quad \text{has full row rank.} \tag{7}$$

## A.2  DUAL OPTIMISM ON THE AUGMENTED LAGRANGIAN

We further show that, for equality constraints, applying dual optimistic ascent to the Augmented Lagrangian produces a compounding optimistic effect in the dual updates, without introducing any new phenomena.

**Proposition 6** (Dual optimism on the Augmented Lagrangian - equalities). *Consider applying dual optimism to the Augmented Lagrangian $\mathcal{L}_c$ via primal-first alternating gradient descent–optimistic ascent with optimism coefficient $\omega$. The resulting primal iterates are equivalent to those of Eq. (AL-GDA) on $\mathcal{L}_{c+\omega}$ and to those of Eq. (Lag-GD-OA) with an optimism coefficient of $c + \omega$.*

*Proof.* See *Proof of Proposition 6* in Appendix D. □

This result demonstrates that the two frameworks are fully interchangeable for equality-constrained problems; one can reproduce the dynamics of the other by adjusting the hyperparameters accordingly.

While this equivalence is formally established only for equality constraints, we hypothesize that a similar relationship holds for problems with inequality constraints. A formal analysis of this more general case is left for future work.

## B  THE AUGMENTED LAGRANGIAN METHOD

For the reader's convenience, in this section we provide computations of the gradient, Hessian, and stationary points of the Augmented Lagrangian function (Eq. (AL)).

Recall the Augmented Lagrangian function:

$$\mathcal{L}_c(\boldsymbol{x}, \boldsymbol{\lambda}, \boldsymbol{\mu}) = f(\boldsymbol{x}) + \frac{1}{2c}\left[\|\boldsymbol{\mu} + c\,\boldsymbol{h}(\boldsymbol{x})\|_2^2 - \|\boldsymbol{\mu}\|_2^2 + \left\|[\boldsymbol{\lambda} + c\,\boldsymbol{g}(\boldsymbol{x})]_+\right\|_2^2 - \|\boldsymbol{\lambda}\|_2^2\right] \tag{8}$$

$$= f(\boldsymbol{x}) + \boldsymbol{\mu}^\top \boldsymbol{h}(\boldsymbol{x}) + \frac{c}{2}\|\boldsymbol{h}(\boldsymbol{x})\|_2^2 + \sum_{i=1}^m \begin{cases} \lambda_i g_i(\boldsymbol{x}) + \frac{c}{2} g_i^2(\boldsymbol{x}), & \text{if } \lambda_i + c\,g_i(\boldsymbol{x}) \geq 0 \\ -\lambda_i^2/2c, & \text{otherwise} \end{cases}. \tag{9}$$

### B.1  GRADIENTS OF $\mathcal{L}_c$

The primal gradient corresponds to:

$$\nabla_{\boldsymbol{x}}\mathcal{L}_c(\boldsymbol{x}, \boldsymbol{\lambda}, \boldsymbol{\mu}) = \nabla f(\boldsymbol{x}) + \left(\boldsymbol{\mu}_t + c\,\boldsymbol{h}(\boldsymbol{x})\right)^\top \nabla \boldsymbol{h}(\boldsymbol{x}) + \sum_{i=1}^m \begin{cases} [\lambda_i + c\,g_i(\boldsymbol{x})]\nabla g_i(\boldsymbol{x}), & \text{if } \lambda_i + c\,g_i(\boldsymbol{x}) \geq 0 \\ 0, & \text{otherwise} \end{cases} \tag{10}$$

$$= \nabla f(\boldsymbol{x}) + \left(\boldsymbol{\mu}_t + c\,\boldsymbol{h}(\boldsymbol{x})\right)^\top \nabla \boldsymbol{h}(\boldsymbol{x}) + \left[\boldsymbol{\lambda}_t + c\,\boldsymbol{g}(\boldsymbol{x})\right]_+^\top \nabla \boldsymbol{g}(\boldsymbol{x}). \tag{11}$$

The dual gradients are:

$$\nabla_{\boldsymbol{\mu}}\mathcal{L}_c(\boldsymbol{x}, \boldsymbol{\lambda}, \boldsymbol{\mu}) = \boldsymbol{h}(\boldsymbol{x}), \tag{12}$$

and

$$\nabla_{\boldsymbol{\lambda}}\mathcal{L}_c(\boldsymbol{x}, \boldsymbol{\lambda}, \boldsymbol{\mu}) = \sum_{i=1}^m \begin{cases} g_i(\boldsymbol{x}), & \text{if } \lambda_i + c\,g_i(\boldsymbol{x}) \geq 0 \\ -\lambda_i/c, & \text{otherwise} \end{cases} \tag{13}$$

$$= \frac{1}{c}\sum_{i=1}^m \begin{cases} c\,g_i(\boldsymbol{x}), & \text{if } \lambda_i + c\,g_i(\boldsymbol{x}) \geq 0 \\ -\lambda_i, & \text{otherwise} \end{cases} \tag{14}$$

$$= -\frac{\boldsymbol{\lambda}}{c} + \frac{1}{c}\sum_{i=1}^m \begin{cases} \lambda_i + c\,g_i(\boldsymbol{x}), & \text{if } \lambda_i + c\,g_i(\boldsymbol{x}) \geq 0 \\ 0, & \text{otherwise} \end{cases} \tag{15}$$

$$= -\frac{\boldsymbol{\lambda}}{c} + \frac{1}{c}[\boldsymbol{\lambda} + c\,\boldsymbol{g}(\boldsymbol{x})]_+. \tag{16}$$

Gathering these together, we get:

$$\begin{aligned} \nabla_{\boldsymbol{x}}\mathcal{L}_c(\boldsymbol{x}, \boldsymbol{\lambda}, \boldsymbol{\mu}) &= \nabla f(\boldsymbol{x}) + \left(\boldsymbol{\mu}_t + c\,\boldsymbol{h}(\boldsymbol{x})\right)^\top \nabla \boldsymbol{h}(\boldsymbol{x}) + \left[\boldsymbol{\lambda}_t + c\,\boldsymbol{g}(\boldsymbol{x})\right]_+^\top \nabla \boldsymbol{g}(\boldsymbol{x}) \\ \nabla_{\boldsymbol{\mu}}\mathcal{L}_c(\boldsymbol{x}, \boldsymbol{\lambda}, \boldsymbol{\mu}) &= \boldsymbol{h}(\boldsymbol{x}) \\ \nabla_{\boldsymbol{\lambda}}\mathcal{L}_c(\boldsymbol{x}, \boldsymbol{\lambda}, \boldsymbol{\mu}) &= \sum_{i=1}^m \begin{cases} g_i(\boldsymbol{x}), & \text{if } \lambda_i + c\,g_i(\boldsymbol{x}) \geq 0 \\ -\lambda_i/c, & \text{otherwise} \end{cases} \\ &= -\frac{\boldsymbol{\lambda}}{c} + \frac{1}{c}[\boldsymbol{\lambda} + c\,\boldsymbol{g}(\boldsymbol{x})]_+. \end{aligned} \tag{17}$$

### B.2  THE HESSIAN OF $\mathcal{L}_c$

As can be observed in Eq. (17) in Appendix B.1, the gradients of $\mathcal{L}_c$ with respect to $\boldsymbol{x}$ and $\boldsymbol{\lambda}$ are not differentiable whenever some $\lambda_i + c\,g_i(\boldsymbol{x})$ is exactly 0. Therefore, we will compute the following Hessian derivations exclusively for $(\boldsymbol{x}, \boldsymbol{\lambda}, \boldsymbol{\mu})$ tuples with $\lambda_i + c\,g_i(\boldsymbol{x}) \neq 0$, for every $i = 1, \ldots, m$.

Note that at any stationary (KKT) tuple $(\boldsymbol{x}^*, \boldsymbol{\lambda}^*, \boldsymbol{\mu}^*)$ of $\mathcal{L}_c$ satisfying *strict* complimentary slackness (Assumption 1) falls in the set of points with a well defined Hessian, for all $c > 0$.

The Hessian of $\mathcal{L}_c$ can be represented as a $3 \times 3$ block matrix:

$$\nabla^2 \mathcal{L}_c = \begin{pmatrix} \nabla^2_{\boldsymbol{x}} \mathcal{L}_c & \nabla^2_{\boldsymbol{x}\boldsymbol{\lambda}} \mathcal{L}_c & \nabla^2_{\boldsymbol{x}\boldsymbol{\mu}} \mathcal{L}_c \\ \nabla^2_{\boldsymbol{\lambda}\boldsymbol{x}} \mathcal{L}_c & \nabla^2_{\boldsymbol{\lambda}\boldsymbol{\lambda}} \mathcal{L}_c & \nabla^2_{\boldsymbol{\lambda}\boldsymbol{\mu}} \mathcal{L}_c \\ \nabla^2_{\boldsymbol{\mu}\boldsymbol{x}} \mathcal{L}_c & \nabla^2_{\boldsymbol{\mu}\boldsymbol{\lambda}} \mathcal{L}_c & \nabla^2_{\boldsymbol{\mu}\boldsymbol{\mu}} \mathcal{L}_c \end{pmatrix}. \tag{18}$$

Let $I_{\mathcal{A}_{\boldsymbol{x}}}$ be the set of indices $i$ for which $\lambda_i + c\, g_i(\boldsymbol{x}) \geq 0$, and let $\mathbf{I}_{\mathcal{A}_{\boldsymbol{x}}} = \mathrm{diag}(I_{\mathcal{A}_{\boldsymbol{x}}})$ be the diagonal matrix containing these indicators.

**The Primal-Primal Block ($\nabla^2_{\boldsymbol{x}} \mathcal{L}_c$).** $\nabla^2_{\boldsymbol{x}} \mathcal{L}_c$ yields the Hessian of the standard Lagrangian evaluated at extrapolated multipliers, plus quadratic terms:

$$\nabla^2_{\boldsymbol{x}} \mathcal{L}_c = \nabla^2_{\boldsymbol{x}} \mathcal{L} \big|_{\boldsymbol{x}, [\boldsymbol{\lambda} + c\, \boldsymbol{g}(\boldsymbol{x})]_+, \boldsymbol{\mu} + c\, \boldsymbol{h}(\boldsymbol{x})} + c \left[ \nabla \boldsymbol{h}(\boldsymbol{x})^\top \nabla \boldsymbol{h}(\boldsymbol{x}) + \nabla \boldsymbol{g}(\boldsymbol{x})^\top \mathbf{I}_{\mathcal{A}_{\boldsymbol{x}}} \nabla \boldsymbol{g}(\boldsymbol{x}) \right], \tag{19}$$

where $\nabla^2_{\boldsymbol{x}} \mathcal{L}$ is the Hessian of the standard Lagrangian $\mathcal{L}$:

$$\nabla^2_{\boldsymbol{x}} \mathcal{L} \big|_{\boldsymbol{x}, [\boldsymbol{\lambda} + c\, \boldsymbol{g}(\boldsymbol{x})]_+, \boldsymbol{\mu} + c\, \boldsymbol{h}(\boldsymbol{x})} = \nabla^2 f(\boldsymbol{x}) + [\boldsymbol{\lambda} + c\, \boldsymbol{g}(\boldsymbol{x})]_+^\top \nabla^2 \boldsymbol{g}(\boldsymbol{x}) + (\boldsymbol{\mu} + c\, \boldsymbol{h}(\boldsymbol{x}))^\top \nabla^2 \boldsymbol{h}(\boldsymbol{x}). \tag{20}$$

**The Primal-Dual Inequality Blocks ($\nabla^2_{\boldsymbol{x}\boldsymbol{\lambda}} \mathcal{L}_c$ and $\nabla^2_{\boldsymbol{\lambda}\boldsymbol{x}} \mathcal{L}_c$).** Differentiating $\nabla_{\boldsymbol{x}} \mathcal{L}_c$ with respect to $\boldsymbol{\lambda}$ yields the Jacobian of inequality constraints for those satisying the inequality $\lambda_i + c\, g_i(\boldsymbol{x}) \geq 0$, and $0$ for the rest:

$$\nabla^2_{\boldsymbol{x}\boldsymbol{\lambda}} \mathcal{L}_c = \nabla \boldsymbol{g}(\boldsymbol{x})^\top \mathbf{I}_{\mathcal{A}_{\boldsymbol{x}}}. \tag{21}$$

It also follows that

$$\nabla^2_{\boldsymbol{\lambda}\boldsymbol{x}} \mathcal{L}_c = \mathbf{I}_{\mathcal{A}_{\boldsymbol{x}}} \nabla \boldsymbol{g}(\boldsymbol{x}) = \left[ \nabla^2_{\boldsymbol{x}\boldsymbol{\lambda}} \mathcal{L}_c \right]^\top. \tag{22}$$

**The Primal-Dual Equality Block ($\nabla^2_{\boldsymbol{x}\boldsymbol{\mu}} \mathcal{L}_c$ and $\nabla^2_{\boldsymbol{\mu}\boldsymbol{x}} \mathcal{L}_c$).** Yields the Jacobian of the equality constraints:

$$\nabla^2_{\boldsymbol{x}\boldsymbol{\mu}} \mathcal{L}_c = \nabla \boldsymbol{h}(\boldsymbol{x})^\top, \tag{23}$$

with

$$\nabla^2_{\boldsymbol{\mu}\boldsymbol{x}} \mathcal{L}_c = \nabla \boldsymbol{h}(\boldsymbol{x}) = \left[ \nabla^2_{\boldsymbol{x}\boldsymbol{\mu}} \mathcal{L}_c \right]^\top. \tag{24}$$

**The Dual-Dual Inequality Block ($\nabla^2_{\boldsymbol{\lambda}\boldsymbol{\lambda}} \mathcal{L}_c$).** Differentiating $\nabla_{\boldsymbol{\lambda}} \mathcal{L}_c$ with respect to $\boldsymbol{\lambda}$ gives:

$$\nabla^2_{\boldsymbol{\lambda}\boldsymbol{\lambda}} \mathcal{L}_c = \frac{1}{c} (\mathbf{I}_{\mathcal{A}_{\boldsymbol{x}}} - I) = -\frac{1}{c} (I - \mathbf{I}_{\mathcal{A}_{\boldsymbol{x}}}). \tag{25}$$

**All other blocks are zero** as the corresponding gradients do not depend on the other dual variables:

$$\nabla^2_{\boldsymbol{\lambda}\boldsymbol{\mu}} \mathcal{L}_c = \nabla^2_{\boldsymbol{\mu}\boldsymbol{\lambda}} \mathcal{L}_c = \nabla^2_{\boldsymbol{\mu}\boldsymbol{\mu}} \mathcal{L}_c = \boldsymbol{0}. \tag{26}$$

Combining all the blocks gives the full Hessian of the Augmented Lagrangian function:

$$\nabla^2 \mathcal{L}_c(\boldsymbol{x}, \boldsymbol{\lambda}, \boldsymbol{\mu}) = \begin{pmatrix} \nabla^2_{\boldsymbol{x}} \mathcal{L}_c & \nabla \boldsymbol{g}(\boldsymbol{x})^\top \mathbf{I}_{\mathcal{A}_{\boldsymbol{x}}} & \nabla \boldsymbol{h}(\boldsymbol{x})^\top \\ \mathbf{I}_{\mathcal{A}_{\boldsymbol{x}}} \nabla \boldsymbol{g}(\boldsymbol{x}) & -\frac{1}{c}(I - \mathbf{I}_{\mathcal{A}_{\boldsymbol{x}}}) & \boldsymbol{0} \\ \nabla \boldsymbol{h}(\boldsymbol{x}) & \boldsymbol{0} & \boldsymbol{0} \end{pmatrix}. \tag{27}$$

### B.3 PRIMAL-FIRST ALTERNATING GRADIENT DESCENT-ASCENT ON $\mathcal{L}_c$

Primal-first alternating gradient descent-projected gradient ascent on the Augmented Lagrangian $\mathcal{L}_c$ has the following structure:

$$\begin{aligned} \boldsymbol{x}_{t+1} &\leftarrow \boldsymbol{x}_t - \eta_{\boldsymbol{x}} \nabla_{\boldsymbol{x}} \mathcal{L}_c(\boldsymbol{x}_t, \boldsymbol{\lambda}_t, \boldsymbol{\mu}_t) \\ \boldsymbol{\mu}_{t+1} &\leftarrow \boldsymbol{\mu}_t + \eta_{\mathrm{dual}} \nabla_{\boldsymbol{\mu}} \mathcal{L}_c(\boldsymbol{x}_{t+1}, \boldsymbol{\lambda}_t, \boldsymbol{\mu}_t) \\ \boldsymbol{\lambda}_{t+1} &\leftarrow \left[ \boldsymbol{\lambda}_t + \eta_{\mathrm{dual}} \nabla_{\boldsymbol{\lambda}} \mathcal{L}_c(\boldsymbol{x}_{t+1}, \boldsymbol{\lambda}_t, \boldsymbol{\mu}_t) \right]_+. \end{aligned} \tag{28}$$

Using the gradients computed in Appendix B.1, we can produce the primal gradient descent update:

$$\boldsymbol{x}_{t+1} \leftarrow \boldsymbol{x}_t - \eta_{\boldsymbol{x}} \left[ \nabla f(\boldsymbol{x}_t) + \left( \boldsymbol{\mu}_t + c\,\boldsymbol{h}(\boldsymbol{x}_t) \right)^\top \nabla \boldsymbol{h}(\boldsymbol{x}_t) + \left[ \boldsymbol{\lambda}_t + c\,\boldsymbol{g}(\boldsymbol{x}_t) \right]_+^\top \nabla \boldsymbol{g}(\boldsymbol{x}_t) \right]. \tag{29}$$

Moreover, the dual ascent updates correspond to:

$$\boldsymbol{\mu}_{t+1} \leftarrow \boldsymbol{\mu}_t + \eta_{\text{dual}} \boldsymbol{h}(\boldsymbol{x}_{t+1}), \tag{30}$$

and to

$$\boldsymbol{\lambda}_{t+1} \leftarrow \left[ \boldsymbol{\lambda}_t + \eta_{\text{dual}} \left[ -\frac{\boldsymbol{\lambda}_t}{c} + \frac{1}{c}[\boldsymbol{\lambda}_t + c\,\boldsymbol{g}(\boldsymbol{x}_{t+1})]_+ \right] \right]_+ \tag{31}$$

$$\leftarrow \left[ \left( 1 - \frac{\eta_{\text{dual}}}{c} \right) \boldsymbol{\lambda}_t + \frac{\eta_{\text{dual}}}{c} \left[ \boldsymbol{\lambda}_t + c\,\boldsymbol{g}(\boldsymbol{x}_{t+1}) \right]_+ \right]_+ \tag{32}$$

$$\leftarrow \left( 1 - \frac{\eta_{\text{dual}}}{c} \right) \boldsymbol{\lambda}_t + \frac{\eta_{\text{dual}}}{c} \left[ \boldsymbol{\lambda}_t + c\,\boldsymbol{g}(\boldsymbol{x}_{t+1}) \right]_+. \tag{33}$$

The last step follows from the fact that the update before the projection consists of a convex combination of two non-negative terms whenever $\eta_{\text{dual}} \leq c$, and thus it itself remains non-negative.

Note that the ascent update with respect to $\boldsymbol{\lambda}$ reduces to the usual Augmented Lagrangian Method ascent update whenever $\eta_{\text{dual}} = c$:

$$\boldsymbol{\lambda}_{t+1} \leftarrow \left[ \boldsymbol{\lambda}_t + c\,\boldsymbol{g}(\boldsymbol{x}_{t+1}) \right]_+. \tag{34}$$

Gathering these together, we get the gradient descent-ascent updates on $\mathcal{L}_c$ from Eq. (AL-GDA),:

$$\boldsymbol{x}_{t+1} \leftarrow \boldsymbol{x}_t - \eta_{\boldsymbol{x}} \left[ \nabla f(\boldsymbol{x}_t) + \left( \boldsymbol{\mu}_t + c\,\boldsymbol{h}(\boldsymbol{x}_t) \right)^\top \nabla \boldsymbol{h}(\boldsymbol{x}_t) + \left[ \boldsymbol{\lambda}_t + c\,\boldsymbol{g}(\boldsymbol{x}_t) \right]_+^\top \nabla \boldsymbol{g}(\boldsymbol{x}_t) \right] \tag{35}$$

$$\boldsymbol{\mu}_{t+1} \leftarrow \boldsymbol{\mu}_t + \eta_{\text{dual}} \boldsymbol{h}(\boldsymbol{x}_{t+1}) \tag{36}$$

$$\boldsymbol{\lambda}_{t+1} \leftarrow \left( 1 - \frac{\eta_{\text{dual}}}{c} \right) \boldsymbol{\lambda}_t + \frac{\eta_{\text{dual}}}{c} \left[ \boldsymbol{\lambda}_t + c\,\boldsymbol{g}(\boldsymbol{x}_{t+1}) \right]_+. \tag{37}$$

## C  EXPERIMENTS

We consider a simple 1D equality-constrained problem to illustrate our theoretical results. In Appendix C.1, we numerically corroborate the equality-constrained equivalence in Theorem 1; in Appendix C.2, we support the hyperparameter-tuning discussion in §4.3 by applying a classical Augmented Lagrangian penalty schedule to the optimism coefficient in dual optimistic ascent.

We consider the following constrained optimization problem:

$$\min_x \frac{1}{2} x^2, \quad \text{subject to } e^x = e. \tag{38}$$

We formulate the constraint as $h(x) = e^x - e$ rather than the linear form $x - 1 = 0$. This ensures that the constraint gradient $\nabla h(x) = e^x$ depends on the iterate, creating a nonlinear optimization landscape where the equivalence between methods is non-trivial. The solution is $x^* = 1$.

All experiments use the hyperparameters in Table 1 and were implemented in PyTorch (Paszke et al., 2019) with Cooper (Gallego-Posada et al., 2025). Our code is available at https://github.com/juan43ramirez/pi-control-is-alm.

Table 1: Hyperparameters used for all experiments.

| Primal Optimizer | $\eta_x$ | Momentum | $\eta_{\text{dual}}$ | $\omega/c$ | $x_0$ | ALM multipliers |
|---|---|---|---|---|---|---|
| GD + Polyak Momentum | 0.01 | 0.5 | 0.1 | 1.0 | 2.0 | 0.0 |

We use gradient descent with Polyak momentum (rather than vanilla GD) for the primal optimizer to emphasize that our findings extend beyond the simple GD setup: any first-order primal optimizer preserves the equivalence (including Adam).

We initialize the multipliers for dual optimistic ascent and the Augmented Lagrangian method according to the change-of-variables rule in Theorem 1 to ensure matching primal iterates:

$$\mu_0^{\text{OGA}} = \mu_0^{\text{ALM}} + (c - \eta_{\text{dual}}) \, h(x_0). \tag{39}$$

### C.1  PRIMAL ITERATES MATCH FOR EQUALITY CONSTRAINTS

Figure 1 numerically illustrates Theorem 1, comparing dual optimistic ascent in Eq. (Lag-GD-OA) with the gradient-descent–ascent Augmented Lagrangian method in Eq. (AL-GDA), using the hyperparameters from Table 1. As predicted, although the internal multipliers of the two methods differ, they produce identical primal iterates.

To provide intuition, we plot the "*effective multiplier*" used by each method—that is, the value that enters the primal gradient when viewed from the perspective of minimizing the standard Lagrangian. For dual optimistic ascent, this is simply $\mu_t^{\text{OGA}}$; for the Augmented Lagrangian method, it is the extrapolated multiplier $\mu_t^{\text{ALM}} + c \, h(x_t)$.

The plots show that these effective multipliers coincide, visually reinforcing the equivalence. This equality of effective multipliers is the key insight used in the proof of Theorem 1.

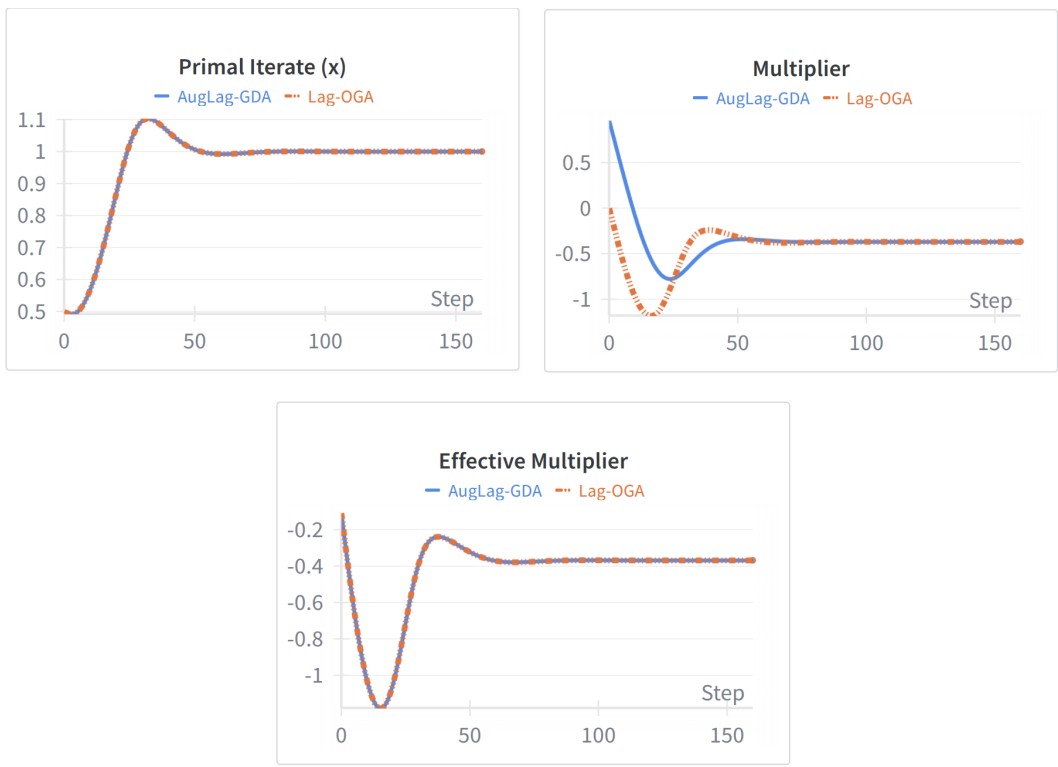

Figure 1: Comparison of iterates for the Augmented Lagrangian method and dual optimistic ascent on the equality-constrained problem ($e^x = e$). As predicted by Theorem 1, the primal iterates $x_t$ are identical.

## C.2 SCHEDULING OF $\omega$ IN DUAL OPTIMISTIC ASCENT

A standard way to schedule the Augmented Lagrangian penalty coefficient $c$ (cf. §2.3) is to use a multiplicative heuristic (see Birgin and Martínez (2014, Eq. 4.9 in Alg. 4.1)):

$$c_{t+1} = \begin{cases} \gamma c_t, & \text{if } \|h(x_t)\| > \beta\|h(x_{t-1})\|, \\ c_t, & \text{otherwise}, \end{cases} \tag{40}$$

where $\gamma > 1$ is the growth factor and $\beta \in (0, 1)$ is the required improvement rate. It is also common to impose an absolute tolerance—not increasing $c$ if the violation $|h(x_t)|$ is already below this threshold—to help avoid numerical instability.

To demonstrate that our theoretical connection is useful for hyperparameter tuning, we adopt this same strategy to schedule the optimism coefficient $\omega$ in dual optimistic ascent. Figure 2 compares dual optimistic ascent with a fixed $\omega$ against a version with an adaptive schedule, using $\gamma = 2$, $\beta = 0.99$, and an absolute tolerance of $10^{-2}$. We choose such a high $\beta$ because the scheduler is evaluated at every optimization step—despite only one primal update occurring between checks—so the expected reduction in violation per step is small, necessitating a lenient improvement threshold.

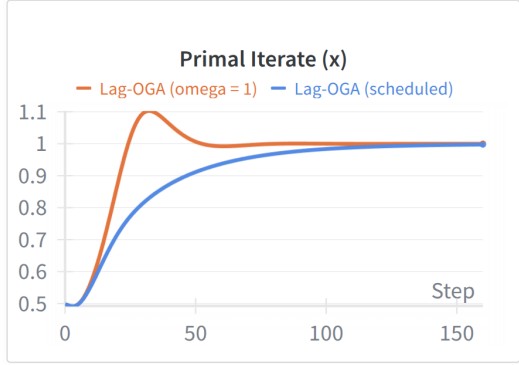
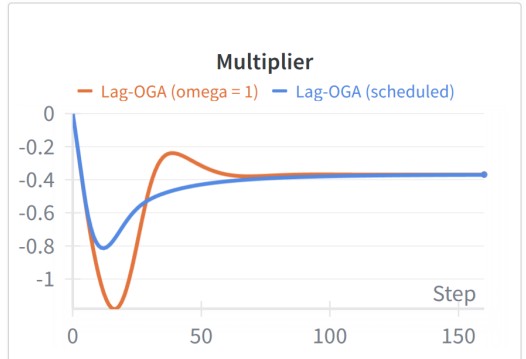

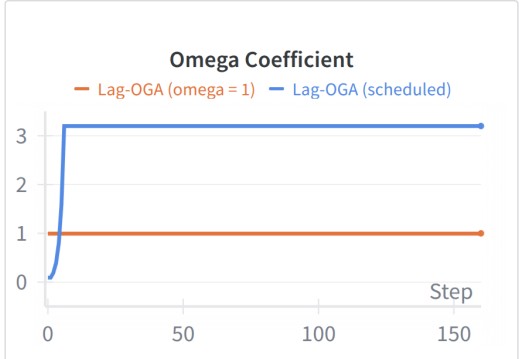

Figure 2: Comparison of iterates for dual optimistic ascent with and without an ALM-inspired $\omega$ scheduler on the equality-constrained problem. The scheduled version reduces multiplier overshoot and allows the primal iterate to converge without overshooting the feasible solution.

## D  PROOFS

**Lemma 1.** *Let $k > 0$ be a constant. A tuple $(\boldsymbol{x}^*, \boldsymbol{\lambda}^*)$ with $\boldsymbol{\lambda}^* \succeq \boldsymbol{0}$ satisfies:*

$$\boldsymbol{\lambda}^* = \big[\boldsymbol{\lambda}^* + k\,\boldsymbol{g}(\boldsymbol{x}^*)\big]_+ \tag{41}$$

*if and only if $(\boldsymbol{x}^*, \boldsymbol{\lambda}^*)$ satisfies primal feasibility and complimentary slackness. That is,*

$$\boldsymbol{g}(\boldsymbol{x}^*) \preceq \boldsymbol{0} \quad \text{and} \quad \boldsymbol{\lambda}^* \odot \boldsymbol{g}(\boldsymbol{x}^*) = \boldsymbol{0}. \tag{42}$$

*Proof.* "$\Rightarrow$." Let $(\boldsymbol{x}^*, \boldsymbol{\lambda}^*)$ satisfy $\boldsymbol{\lambda}^* = \big[\boldsymbol{\lambda}^* + k\,\boldsymbol{g}(\boldsymbol{x}^*)\big]_+$. We can analyze this relationship component-wise, $\lambda_i^* = \max\{0, \lambda_i^* + k\,g_i(\boldsymbol{x}^*)\}$:

- If $\lambda_i^* > 0$, then the equation requires $\lambda_i^* = \lambda_i^* + k\,g_i(\boldsymbol{x}^*)$, which means $g_i(\boldsymbol{x}^*) = 0$.

- If $\lambda_i^* = 0$, then the equation becomes $0 = \max\{0, k\,g_i(\boldsymbol{x}^*)\}$, which requires $k\,g_i(\boldsymbol{x}^*) \leq 0$, and thus $g_i(\boldsymbol{x}^*) \leq 0$.

Combining these two cases, we have that $g_i(\boldsymbol{x}^*) \leq 0$ for all $i$ (primal feasibility) and that for each $i$, either $\lambda_i^* = 0$ or $g_i(\boldsymbol{x}^*) = 0$ or both. This is precisely the definition of complementary slackness.

"$\Leftarrow$." Let $(\boldsymbol{x}^*, \boldsymbol{\lambda}^*)$ satisfy primal feasibility and complementary slackness. We analyze the relationship component-wise for each $i$:

- If $\lambda_i^* > 0$, then complementary slackness ($\lambda_i^* g_i(\boldsymbol{x}^*) = 0$) implies that $g_i(\boldsymbol{x}^*) = 0$. In this case, the right-hand side of the target equation becomes $\max\{0, \lambda_i^* + k \cdot 0\} = \max\{0, \lambda_i^*\} = \lambda_i^*$. The equality holds.

- If $\lambda_i^* = 0$, then primal feasibility implies that $g_i(\boldsymbol{x}^*) \leq 0$. Since $k > 0$, we have $k\, g_i(\boldsymbol{x}^*) \leq 0$. In this case, the right-hand side becomes $\max\{0, 0 + k\, g_i(\boldsymbol{x}^*)\} = \max\{0, k\, g_i(\boldsymbol{x}^*)\} = 0 = \lambda_i^*$. The equality also holds.

Since the equality $\lambda_i^* = \max\{0, \lambda_i^* + k\, g_i(\boldsymbol{x}^*)\}$ holds for all components $i$, the vector equation $\boldsymbol{\lambda}^* = \left[\boldsymbol{\lambda}^* + k\, \boldsymbol{g}(\boldsymbol{x}^*)\right]_+$ is satisfied. $\qquad\square$

***Proof of Proposition*** *1.* We will now show that stationary (fixed) points of algorithms Eq. (AL-GDA) and Eq. (Lag-GD-OA) match, regardless of their hyper-parameter choices.

"$\Rightarrow$." Let $(\boldsymbol{x}^*, \boldsymbol{\lambda}^*, \boldsymbol{\mu}^*)$ be a fixed point of the Augmented Lagrangian dynamics of Eq. (AL-GDA). It follows that:

$$\boldsymbol{\mu}^* = \boldsymbol{\mu}^* + \eta_{\text{dual}}\boldsymbol{h}(\boldsymbol{x}^*) \tag{43}$$

$$\boldsymbol{\lambda}^* = \left(1 - \frac{\eta_{\text{dual}}}{c}\right)\boldsymbol{\lambda}^* + \frac{\eta_{\text{dual}}}{c}\left[\boldsymbol{\lambda}^* + c\, \boldsymbol{g}(\boldsymbol{x}^*)\right]_+. \tag{44}$$

These equations imply primal feasibility and complimentary slackness. In fact, from the first equation, $\boldsymbol{\mu}^* = \boldsymbol{\mu}^* + \eta_{\text{dual}}\boldsymbol{h}(\boldsymbol{x}^*)$ implies that $\eta_{\text{dual}}\boldsymbol{h}(\boldsymbol{x}^*) = 0$, and since $\eta_{\text{dual}} > 0$, we must have $\boldsymbol{h}(\boldsymbol{x}^*) = 0$. This is primal feasibility for the equality constraints.

For the second equation, we can simplify:

$$\boldsymbol{\lambda}^* = \left(1 - \frac{\eta_{\text{dual}}}{c}\right)\boldsymbol{\lambda}^* + \frac{\eta_{\text{dual}}}{c}\left[\boldsymbol{\lambda}^* + c\, \boldsymbol{g}(\boldsymbol{x}^*)\right]_+ \tag{45}$$

$$\frac{\eta_{\text{dual}}}{c}\boldsymbol{\lambda}^* = \frac{\eta_{\text{dual}}}{c}\left[\boldsymbol{\lambda}^* + c\, \boldsymbol{g}(\boldsymbol{x}^*)\right]_+ \tag{46}$$

$$\boldsymbol{\lambda}^* = \left[\boldsymbol{\lambda}^* + c\, \boldsymbol{g}(\boldsymbol{x}^*)\right]_+ \tag{47}$$

As a consequence of Lemma 1, this condition implies both primal feasibility for the inequality constraints ($\boldsymbol{g}(\boldsymbol{x}^*) \preceq \boldsymbol{0}$) and complementary slackness ($\lambda_i^* g_i(\boldsymbol{x}^*) = 0$ for all $i$).

Primal feasibility can be used to argue:

$$\boldsymbol{\mu}^* = \boldsymbol{\mu}^* + c\, \boldsymbol{h}(\boldsymbol{x}^*), \tag{48}$$

since $c > 0$. Together with the fact that $\boldsymbol{\lambda}^* = \left[\boldsymbol{\lambda}^* + c\, \boldsymbol{g}(\boldsymbol{x}^*)\right]_+$, it follows that:

$$\boldsymbol{x}^* = \boldsymbol{x}^* - \eta_{\boldsymbol{x}}\left[\nabla f(\boldsymbol{x}^*) + \left[\boldsymbol{\lambda}^* + c\, \boldsymbol{g}(\boldsymbol{x}^*)\right]_+^\top \nabla\boldsymbol{g}(\boldsymbol{x}^*) + \left(\boldsymbol{\mu}^* + c\, \boldsymbol{h}(\boldsymbol{x}^*)\right)^\top \nabla\boldsymbol{h}(\boldsymbol{x}^*)\right] \tag{49}$$

$$= \boldsymbol{x}^* - \eta_{\boldsymbol{x}}\left[\nabla f(\boldsymbol{x}^*) + (\boldsymbol{\lambda}^*)^\top \nabla\boldsymbol{g}(\boldsymbol{x}^*) + (\boldsymbol{\mu}^*)^\top \nabla\boldsymbol{h}(\boldsymbol{x}^*)\right], \tag{50}$$

which corresponds to the primal stationarity condition of dual optimistic ascent in Eq. (Lag-GD-OA); and also simply means:

$$\nabla_{\boldsymbol{x}}\mathcal{L}(\boldsymbol{x}^*, \boldsymbol{\lambda}^*, \boldsymbol{\mu}^*) = 0. \tag{51}$$

We have thus established that $(\boldsymbol{x}^*, \boldsymbol{\lambda}^*, \boldsymbol{\mu}^*)$ is a KKT point of Eq. (CMP).

Finally, we show that the dual variables are also stationary under the optimistic ascent dynamics. At the fixed point $(\boldsymbol{x}^*, \boldsymbol{\lambda}^*, \boldsymbol{\mu}^*)$, we have $\boldsymbol{x}_t = \boldsymbol{x}_{t-1} = \boldsymbol{x}^*$. The dual updates from Eq. (Lag-GD-OA) therefore become:

$$\boldsymbol{\mu}^* = \boldsymbol{\mu}^* + \eta_{\text{dual}}\boldsymbol{h}(\boldsymbol{x}^*) + \omega\left(\boldsymbol{h}(\boldsymbol{x}^*) - \boldsymbol{h}(\boldsymbol{x}^*)\right) \tag{52}$$

$$\boldsymbol{\lambda}^* = \left[\boldsymbol{\lambda}^* + \eta_{\text{dual}}\boldsymbol{g}(\boldsymbol{x}^*) + \omega\left(\boldsymbol{g}(\boldsymbol{x}^*) - \boldsymbol{g}(\boldsymbol{x}^*)\right)\right]_+ \tag{53}$$

The update for $\boldsymbol{\mu}^*$ simplifies to $\boldsymbol{\mu}^* = \boldsymbol{\mu}^* + \eta_{\text{dual}}\boldsymbol{h}(\boldsymbol{x}^*)$. As primal feasibility requires $\boldsymbol{h}(\boldsymbol{x}^*) = 0$, this becomes $\boldsymbol{\mu}^* = \boldsymbol{\mu}^*$, which is trivially satisfied.

The update for a stationary $\boldsymbol{\lambda}^*$ simplifies to $\boldsymbol{\lambda}^* = \left[\boldsymbol{\lambda}^* + \eta_{\text{dual}}\boldsymbol{g}(\boldsymbol{x}^*)\right]_+$. This condition holds due to Lemma 1 and the combination of primal feasibility and complementary slackness proved above.

Since the stationarity conditions for the primal variable $\boldsymbol{x}$ and the dual variables $\boldsymbol{\lambda}$ and $\boldsymbol{\mu}$ are all met, the tuple $(\boldsymbol{x}^*, \boldsymbol{\lambda}^*, \boldsymbol{\mu}^*)$ is a fixed point of the dual-first optimistic ascent dynamics in Eq. (Lag-GD-OA).

"$\Leftarrow$." Let $(\boldsymbol{x}^*, \boldsymbol{\lambda}^*, \boldsymbol{\mu}^*)$ be a fixed point of the dual-first optimistic ascent dynamics of Eq. (Lag-GD-OA). At a stationary point, the optimistic term $\boldsymbol{g}(\boldsymbol{x}_t) - \boldsymbol{g}(\boldsymbol{x}_{t-1})$ vanishes. Therefore, the fixed-point conditions are:

$$\boldsymbol{\mu}^* = \boldsymbol{\mu}^* + \eta_{\text{dual}} \boldsymbol{h}(\boldsymbol{x}^*) \tag{54}$$

$$\boldsymbol{\lambda}^* = \left[\boldsymbol{\lambda}^* + \eta_{\text{dual}} \, \boldsymbol{g}(\boldsymbol{x}^*)\right]_+ \tag{55}$$

$$\boldsymbol{x}^* = \boldsymbol{x}^* - \eta_{\boldsymbol{x}} \left[\nabla f(\boldsymbol{x}^*) + (\boldsymbol{\lambda}^*)^\top \nabla \boldsymbol{g}(\boldsymbol{x}^*) + (\boldsymbol{\mu}^*)^\top \nabla \boldsymbol{h}(\boldsymbol{x}^*)\right] \tag{56}$$

From these three conditions, we can deduce that the tuple $(\boldsymbol{x}^*, \boldsymbol{\lambda}^*, \boldsymbol{\mu}^*)$ must satisfy the KKT conditions:

1. From the $\boldsymbol{\mu}$ condition, we get $\boldsymbol{h}(\boldsymbol{x}^*) = 0$.

2. The $\boldsymbol{\lambda}$ update condition, $\boldsymbol{\lambda}^* = \left[\boldsymbol{\lambda}^* + \eta_{\text{dual}} \, \boldsymbol{g}(\boldsymbol{x}^*)\right]_+$, together with Lemma 1, implies both primal feasibility for inequalities and complementary slackness.

3. The $\boldsymbol{x}$ update implies that the gradient of the Lagrangian is zero: $\nabla_{\boldsymbol{x}} \mathcal{L}(\boldsymbol{x}^*, \boldsymbol{\lambda}^*, \boldsymbol{\mu}^*) = 0$.

Now we will show that this KKT point is also a fixed point of the Augmented Lagrangian dynamics in Eq. (AL-GDA).

- **For $\boldsymbol{\mu}$:** Primal feasibility implies $\boldsymbol{\mu}^* = \boldsymbol{\mu}^* + \eta_{\text{dual}} \boldsymbol{h}(\boldsymbol{x}^*)$.

- **For $\boldsymbol{\lambda}$:** Primal feasibility and complimentary slackness imply $\boldsymbol{\lambda}^* = \left[\boldsymbol{\lambda}^* + c \, \boldsymbol{g}(\boldsymbol{x}^*)\right]_+$ (Lemma 1). This in turn implies:

$$\boldsymbol{\lambda}^* = \left(1 - \frac{\eta_{\text{dual}}}{c}\right) \boldsymbol{\lambda}^* + \frac{\eta_{\text{dual}}}{c} \boldsymbol{\lambda}^* = \left(1 - \frac{\eta_{\text{dual}}}{c}\right) \boldsymbol{\lambda}^* + \frac{\eta_{\text{dual}}}{c} \left[\boldsymbol{\lambda}^* + c \, \boldsymbol{g}(\boldsymbol{x}^*)\right]_+. \tag{57}$$

The primal update for the Augmented Lagrangian method is stationary if the gradient term is zero:

$$\nabla f(\boldsymbol{x}^*) + \left[\boldsymbol{\lambda}^* + c \, \boldsymbol{g}(\boldsymbol{x}^*)\right]_+^\top \nabla \boldsymbol{g}(\boldsymbol{x}^*) + \left(\boldsymbol{\mu}^* + c \, \boldsymbol{h}(\boldsymbol{x}^*)\right)^\top \nabla \boldsymbol{h}(\boldsymbol{x}^*) = 0 \tag{58}$$

But as argued before, $\boldsymbol{\lambda}^* = \left[\boldsymbol{\lambda}^* + c \, \boldsymbol{g}(\boldsymbol{x}^*)\right]_+$ and $\boldsymbol{\mu}^* = \boldsymbol{\mu}^* + c \, \boldsymbol{h}(\boldsymbol{x}^*)$. Therefore:

$$\nabla f(\boldsymbol{x}^*) + (\boldsymbol{\lambda}^*)^\top \nabla \boldsymbol{g}(\boldsymbol{x}^*) + (\boldsymbol{\mu}^*)^\top \nabla \boldsymbol{h}(\boldsymbol{x}^*) = \nabla_{\boldsymbol{x}} \mathcal{L}(\boldsymbol{x}^*, \boldsymbol{\lambda}^*, \boldsymbol{\mu}^*) \tag{59}$$

From the Eq. (Lag-GD-OA) fixed-point conditions, we know this term is zero. Thus, the primal update is stationary.

Since the stationarity conditions for the primal and dual variables are all met, the point $(\boldsymbol{x}^*, \boldsymbol{\lambda}^*, \boldsymbol{\mu}^*)$ is also a fixed point of the Augmented Lagrangian dynamics. This completes the proof.

$\square$

## D.1 Equivalence Proof for Equality-constrained Problems

***Proof of Theorem 1.*** Consider the following primal-first alternating gradient descent–ascent updates on the Augmented Lagrangian:

$$\boldsymbol{x}_{t+1} \leftarrow \texttt{MinStep}\left(\boldsymbol{x}_t, \nabla f(\boldsymbol{x}_t) + \left(\boldsymbol{\mu}_t^{(\text{ALM})} + c \, \boldsymbol{h}(\boldsymbol{x}_t)\right)^\top \nabla \boldsymbol{h}(\boldsymbol{x}_t)\right),$$
$$\boldsymbol{\mu}_{t+1}^{(\text{ALM})} \leftarrow \boldsymbol{\mu}_t^{(\text{ALM})} + \eta_{\text{dual}} \, \boldsymbol{h}(\boldsymbol{x}_{t+1}). \tag{60}$$

These are analogous to Eq. (AL-GDA), but stated for a generic first-order primal minimization step `MinStep`. This step could, for instance, be replaced with Adam (Kingma and Ba, 2015), and the remainder of the proof would still hold.

Now consider the following dual-first alternating gradient descent–optimistic ascent updates on the Lagrangian, similar to Eq. (Lag-GD-OA):

$$\boldsymbol{\mu}_{t+1}^{(\text{OGA})} \leftarrow \boldsymbol{\mu}_t^{(\text{OGA})} + \eta_{\text{dual}}\, \boldsymbol{h}(\boldsymbol{x}_t) + \omega\big(\boldsymbol{h}(\boldsymbol{x}_t) - \boldsymbol{h}(\boldsymbol{x}_{t-1})\big),$$
$$\boldsymbol{x}_{t+1} \leftarrow \texttt{MinStep}\Big(\boldsymbol{x}_t, \nabla f(\boldsymbol{x}_t) + \big(\boldsymbol{\mu}_{t+1}^{(\text{OGA})}\big)^\top \nabla \boldsymbol{h}(\boldsymbol{x}_t)\Big). \tag{61}$$

Suppose $\omega = c$. We show by induction that the primal iterates $\{\boldsymbol{x}_t\}_{t=0}^\infty$ generated by the primal-first ALM algorithm and the dual-first OGA algorithm are identical.

**Inductive Step**. Define:

$$\boldsymbol{\mu}_{t+1}^{(\text{OGA})} = \boldsymbol{\mu}_t^{(\text{ALM})} + c\, \boldsymbol{h}(\boldsymbol{x}_t). \tag{62}$$

With this substitution, the primal update in Eq. (60) becomes

$$\boldsymbol{x}_{t+1} = \texttt{MinStep}\Big(\boldsymbol{x}_t, \nabla f(\boldsymbol{x}_t) + \big(\boldsymbol{\mu}_{t+1}^{(\text{OGA})}\big)^\top \nabla \boldsymbol{h}(\boldsymbol{x}_t)\Big) \tag{63}$$

$$= \texttt{MinStep}\Big(\boldsymbol{x}_t, \nabla_{\boldsymbol{x}} \mathcal{L}\big(\boldsymbol{x}_t, \boldsymbol{\mu}_{t+1}^{(\text{OGA})}\big)\Big), \tag{64}$$

i.e., a descent step on the Lagrangian with multipliers $\boldsymbol{\mu}_{t+1}^{(\text{OGA})}$.

For the dual update, we have

$$\boldsymbol{\mu}_{t+1}^{(\text{OGA})} = \boldsymbol{\mu}_t^{(\text{ALM})} + c\, \boldsymbol{h}(\boldsymbol{x}_t) \tag{65}$$

$$= \boldsymbol{\mu}_{t-1}^{(\text{ALM})} + \eta_{\text{dual}}\, \boldsymbol{h}(\boldsymbol{x}_t) + c\, \boldsymbol{h}(\boldsymbol{x}_t) \tag{66}$$

$$= \boldsymbol{\mu}_t^{(\text{OGA})} - c\, \boldsymbol{h}(\boldsymbol{x}_{t-1}) + \eta_{\text{dual}}\, \boldsymbol{h}(\boldsymbol{x}_t) + c\, \boldsymbol{h}(\boldsymbol{x}_t) \tag{67}$$

$$= \boldsymbol{\mu}_t^{(\text{OGA})} + \eta_{\text{dual}}\, \boldsymbol{h}(\boldsymbol{x}_t) + c\big(\boldsymbol{h}(\boldsymbol{x}_t) - \boldsymbol{h}(\boldsymbol{x}_{t-1})\big), \tag{68}$$

which matches the optimistic ascent update in Eq. (126) with $\omega = c$. Combining this dual update with Eq. (63) yields exactly the algorithm in Eq. (126). Thus, if the primal iterates generated by both algorithms match up until step $t$, they will match at step $t+1$ under the change of variable $\boldsymbol{\mu}_{t+1}^{(\text{OGA})} = \boldsymbol{\mu}_t^{(\text{ALM})} + c\, \boldsymbol{h}(\boldsymbol{x}_t)$.

**Base Case**. We establish the base case for the induction by showing that the first primal iterates, $\boldsymbol{x}_1$, are identical under the specified initializations.

The primal descent update for Eq. (125) corresponds to a descent step on the Lagrangian with an effective multiplier of $\boldsymbol{\mu}_0^{(\text{ALM})} + c\, \boldsymbol{h}(\boldsymbol{x}_0)$. The primal descent update for Eq. (126) corresponds to a descent step on the Lagrangian with an effective multiplier of $\boldsymbol{\mu}_1^{(\text{OGA})}$. For the first step, the difference term in Eq. (126) is typically set to zero (as $\boldsymbol{h}(\boldsymbol{x}_{-1})$ is undefined), so that

$$\boldsymbol{\mu}_1^{(\text{OGA})} = \boldsymbol{\mu}_0^{(\text{OGA})} + \eta_{\text{dual}}\, \boldsymbol{h}(\boldsymbol{x}_0). \tag{69}$$

For the primal iterates to be identical, their effective multipliers must satisfy

$$\boldsymbol{\mu}_0^{(\text{ALM})} + c\, \boldsymbol{h}(\boldsymbol{x}_0) = \boldsymbol{\mu}_0^{(\text{OGA})} + \eta_{\text{dual}}\, \boldsymbol{h}(\boldsymbol{x}_0). \tag{70}$$

This equality is ensured by initializing the OGA method relative to the ALM method as

$$\boldsymbol{\mu}_0^{(\text{OGA})} = \boldsymbol{\mu}_0^{(\text{ALM})} + (c - \eta_{\text{dual}})\, \boldsymbol{h}(\boldsymbol{x}_0). \tag{71}$$

With this initialization, both effective multipliers for the first primal update are identical, yielding the same first iterate for both algorithms:

$$\boldsymbol{x}_1 = \texttt{MinStep}\Big(\boldsymbol{x}_0, \nabla f(\boldsymbol{x}_0) + \big(\boldsymbol{\mu}_0^{(\text{ALM})} + c\, \boldsymbol{h}(\boldsymbol{x}_0)\big)^\top \nabla \boldsymbol{h}(\boldsymbol{x}_0)\Big). \tag{72}$$

Combined with the recurrence in Eq. (63), this initial condition implies by induction that all subsequent primal iterates $\boldsymbol{x}_t$ also coincide. Therefore, the two algorithms are equivalent.

$$\square$$

### D.2 EQUIVALENCE PROOF FOR INEQUALITY-CONSTRAINED PROBLEMS

This subsection provides the full proof of Theorem 2, which establishes the equivalence of the locally stable stationary points (LSSPs) for the two algorithms in inequality-constrained problems. Our proof strategy is as follows: in Lemmas 2 and 4, we first derive the Jacobians of the update operators defined by Eq. (AL-GDA) and Eq. (Lag-GD-OA), respectively. We then derive their corresponding characteristic polynomials in Lemmas 3 and 5. Finally, in the main proof of Theorem 2, we establish a relationship between the roots of these polynomials, which govern local convergence when $\omega = c$.

**Lemma 2.** *Consider primal-first alternating gradient descent–projected gradient ascent on the Augmented Lagrangian from Eq. (AL-GDA), applied to a problem with inequality constraints only:*

$$\boldsymbol{x}_{t+1} \leftarrow \boldsymbol{x}_t - \eta_{\boldsymbol{x}} \left[ \nabla f(\boldsymbol{x}_t) + \left[ \boldsymbol{\lambda}_t + c\, \boldsymbol{g}(\boldsymbol{x}_t) \right]_+^\top \nabla \boldsymbol{g}(\boldsymbol{x}_t) \right], \tag{73}$$

$$\boldsymbol{\lambda}_{t+1} \leftarrow \left( 1 - \frac{\eta_{dual}}{c} \right) \boldsymbol{\lambda}_t + \frac{\eta_{dual}}{c} \left[ \boldsymbol{\lambda}_t + c\, \boldsymbol{g}(\boldsymbol{x}_{t+1}) \right]_+, \tag{74}$$

*with $\eta_{dual} < c$.*

*Let $(\boldsymbol{x}^*, \boldsymbol{\lambda}^*)$ be a stationary (KKT) point of Eq. (CMP) satisfying Assumption 1. The Jacobian of the update operator in Eq. (73), evaluated at $(\boldsymbol{x}^*, \boldsymbol{\lambda}^*)$ and acting on the partitioned state*

$$(\boldsymbol{x}_t, \boldsymbol{\lambda}_{t,\mathcal{A}_{\boldsymbol{x}}}, \boldsymbol{\lambda}_{t,I}) \mapsto (\boldsymbol{x}_{t+1}, \boldsymbol{\lambda}_{t+1,A}, \boldsymbol{\lambda}_{t+1,I}),$$

*is*

$$\mathcal{J}_{AL} = \begin{pmatrix} \mathbb{I} - \eta_{\boldsymbol{x}}(A + c\,B^\top B) & -\eta_{\boldsymbol{x}} B^\top & 0 \\ \eta_{dual} B \left( \mathbb{I} - \eta_{\boldsymbol{x}}(A + c\,B^\top B) \right) & \mathbb{I} - \eta_{\boldsymbol{x}} \eta_{dual} B B^\top & 0 \\ 0 & 0 & (1 - \eta_{dual}/c)\mathbb{I} \end{pmatrix}. \tag{75}$$

*Here, $\boldsymbol{\lambda}_t = (\boldsymbol{\lambda}_{t,\mathcal{A}_{\boldsymbol{x}}}, \boldsymbol{\lambda}_{t,I})$ partitions the multipliers into active components*

$$\boldsymbol{\lambda}_{t,\mathcal{A}_{\boldsymbol{x}}} = \{ \lambda_i \mid g_i(\boldsymbol{x}^*) = 0, \ \lambda_i^* > 0 \},$$

*and inactive ones*

$$\boldsymbol{\lambda}_{t,I} = \{ \lambda_i \mid g_i(\boldsymbol{x}^*) < 0, \ \lambda_i^* = 0 \}.$$

*Under the same partition, applied to the constraints $\boldsymbol{g}(\boldsymbol{x}) = [\boldsymbol{g}_{\mathcal{A}_{\boldsymbol{x}}}(\boldsymbol{x})^\top, \boldsymbol{g}_I(\boldsymbol{x})^\top]^\top$, the matrices are*

$$A = \nabla_{\boldsymbol{x}}^2 \mathcal{L}(\boldsymbol{x}^*, \boldsymbol{\lambda}^*) = \nabla^2 f(\boldsymbol{x}^*) + \sum_{i \in \mathcal{A}_{\boldsymbol{x}}} \lambda_i^* \nabla^2 g_i(\boldsymbol{x}^*), \tag{76}$$

$$B = \nabla \boldsymbol{g}_{\mathcal{A}_{\boldsymbol{x}}}(\boldsymbol{x}^*). \tag{77}$$

*Proof.* The Augmented Lagrangian method in Eq. (73) alternates between primal and dual updates, beginning with the primal variables. It can therefore be written as the composition of two operators, one for each step. The Jacobian of the full update is

$$\mathcal{J}_{AL} = \mathcal{J}_{\boldsymbol{\lambda}} \mathcal{J}_{\boldsymbol{x}}, \tag{78}$$

where $\mathcal{J}_{\boldsymbol{x}}$ and $\mathcal{J}_{\boldsymbol{\lambda}}$ denote the Jacobians of the primal and dual steps, respectively.

The Jacobian of the primal step $(\boldsymbol{x}_t, \boldsymbol{\lambda}_t) \mapsto (\boldsymbol{x}_{t+1}, \boldsymbol{\lambda}_t)$ is

$$\mathcal{J}_{\boldsymbol{x}} = \begin{pmatrix} \mathbb{I} - \eta_{\boldsymbol{x}} \nabla_{\boldsymbol{x}}^2 \mathcal{L}_c & -\eta_{\boldsymbol{x}} \nabla_{\boldsymbol{x}\boldsymbol{\lambda}}^2 \mathcal{L}_c \\ 0 & \mathbb{I} \end{pmatrix}, \tag{79}$$

and the Jacobian of the dual step $(\boldsymbol{x}_{t+1}, \boldsymbol{\lambda}_t) \mapsto (\boldsymbol{x}_{t+1}, \boldsymbol{\lambda}_{t+1})$ is

$$\mathcal{J}_{\boldsymbol{\lambda}} = \begin{pmatrix} \mathbb{I} & 0 \\ \eta_{dual} \nabla_{\boldsymbol{\lambda}\boldsymbol{x}}^2 \mathcal{L}_c & \mathbb{I} + \eta_{dual} \nabla_{\boldsymbol{\lambda}}^2 \mathcal{L}_c \end{pmatrix}. \tag{80}$$

The primal Jacobian is well defined for any $(\boldsymbol{x}_t, \boldsymbol{\lambda}_t)$ with $\boldsymbol{\lambda}_t \succeq \boldsymbol{0}$ whenever $\lambda_{t,i} + c\, g_i(\boldsymbol{x}_t) \neq 0$ for all constraints $i = 1, \ldots, m$. Similarly, the dual jacobian is well defined whenever $\lambda_{t,i} + c\, g_i(\boldsymbol{x}_{t+1}) \neq 0$. In particular, both Jacobians are well defined at $(\boldsymbol{x}^*, \boldsymbol{\lambda}^*)$ due to strict complimentary slackness.

Multiplying gives

$$
\mathcal{J}_{\mathrm{AL}} = \begin{pmatrix} \mathbb{I} - \eta_{\boldsymbol{x}} \nabla_{\boldsymbol{x}}^2 \mathcal{L}_c & -\eta_{\boldsymbol{x}} \nabla_{\boldsymbol{x}\boldsymbol{\lambda}}^2 \mathcal{L}_c \\ \eta_{\mathrm{dual}} \nabla_{\boldsymbol{\lambda}\boldsymbol{x}}^2 \mathcal{L}_c \left( \mathbb{I} - \eta_{\boldsymbol{x}} \nabla_{\boldsymbol{x}}^2 \mathcal{L}_c \right) & \mathbb{I} + \eta_{\mathrm{dual}} \nabla_{\boldsymbol{\lambda}}^2 \mathcal{L}_c - \eta_{\boldsymbol{x}} \eta_{\mathrm{dual}} (\nabla_{\boldsymbol{\lambda}\boldsymbol{x}}^2 \mathcal{L}_c)(\nabla_{\boldsymbol{x}\boldsymbol{\lambda}}^2 \mathcal{L}_c) \end{pmatrix}. \tag{81}
$$

To recover the structure of $\mathcal{J}_{\mathrm{AL}}$, we evaluate $\nabla_{\boldsymbol{x}}^2 \mathcal{L}_c$, $\nabla_{\boldsymbol{x}\boldsymbol{\lambda}}^2 \mathcal{L}_c$, and $\nabla_{\boldsymbol{\lambda}}^2 \mathcal{L}_c$ at $(\boldsymbol{x}^*, \boldsymbol{\lambda}^*)$. Explicit forms of these terms at generic $(\boldsymbol{x}, \boldsymbol{\lambda})$ were already derived in Appendix B.2.

We now partition the state $(\boldsymbol{x}_t, \boldsymbol{\lambda}_t)$ into $(\boldsymbol{x}_t, \boldsymbol{\lambda}_{t,\mathcal{A}_{\boldsymbol{x}}}, \boldsymbol{\lambda}_{t,I})$, where $\boldsymbol{\lambda}_{t,\mathcal{A}_{\boldsymbol{x}}}$ corresponds to active constraints at $\boldsymbol{x}^*$ and $\boldsymbol{\lambda}_{t,I}$ to inactive ones. This partition is well defined under strict complementarity.

**The primal Hessian** ($\nabla_{\boldsymbol{x}}^2 \mathcal{L}_c$). Complementary slackness implies $\boldsymbol{\lambda}^* = [\boldsymbol{\lambda}^* + c\,\boldsymbol{g}(\boldsymbol{x}^*)]_+$, both for active and inactive constraints (Lemma 1).

Substituting into the primal Hessian (see Eq. (19)) yields

$$
\nabla_{\boldsymbol{x}}^2 \mathcal{L}_c \big|_{\boldsymbol{x}^*, \boldsymbol{\lambda}^*} = \nabla^2 f(\boldsymbol{x}^*) + (\boldsymbol{\lambda}^*)^\top \nabla^2 \boldsymbol{g}(\boldsymbol{x}^*) + c\, \nabla \boldsymbol{g}_{\mathcal{A}_{\boldsymbol{x}}}(\boldsymbol{x}^*)^\top \nabla \boldsymbol{g}_{\mathcal{A}_{\boldsymbol{x}}}(\boldsymbol{x}^*) \tag{82}
$$

$$
= A + c\, B^\top B. \tag{83}
$$

**The primal–dual Hessians for _active_ constraints** are

$$
\nabla_{\boldsymbol{x}\boldsymbol{\lambda}_{\mathcal{A}_{\boldsymbol{x}}}}^2 \mathcal{L}_c \big|_{\boldsymbol{x}^*, \boldsymbol{\lambda}^*} = \nabla \boldsymbol{g}_{\mathcal{A}_{\boldsymbol{x}}}(\boldsymbol{x}^*)^\top = B^\top, \qquad \nabla_{\boldsymbol{\lambda}_{\mathcal{A}_{\boldsymbol{x}}}\boldsymbol{x}}^2 \mathcal{L}_c \big|_{\boldsymbol{x}^*, \boldsymbol{\lambda}^*} = \nabla \boldsymbol{g}_{\mathcal{A}_{\boldsymbol{x}}}(\boldsymbol{x}^*) = B. \tag{84}
$$

**The primal–dual Hessians for _inactive_ constraints** vanish:

$$
\nabla_{\boldsymbol{x}\boldsymbol{\lambda}_I}^2 \mathcal{L}_c \big|_{\boldsymbol{x}^*, \boldsymbol{\lambda}^*} = 0, \quad \nabla_{\boldsymbol{\lambda}_I \boldsymbol{x}}^2 \mathcal{L}_c \big|_{\boldsymbol{x}^*, \boldsymbol{\lambda}^*} = 0. \tag{85}
$$

**The dual–dual Hessian for active constraints** vanishes:

$$
\nabla_{\boldsymbol{\lambda}_{\mathcal{A}_{\boldsymbol{x}}}}^2 \mathcal{L}_c \big|_{\boldsymbol{x}^*, \boldsymbol{\lambda}^*} = 0. \tag{86}
$$

**The dual–dual Hessian for inactive constraints** is

$$
\nabla_{\boldsymbol{\lambda}_I}^2 \mathcal{L}_c \big|_{\boldsymbol{x}^*, \boldsymbol{\lambda}^*} = -\frac{1}{c}\,\mathbb{I}. \tag{87}
$$

Substituting all these expressions, the Jacobian $\mathcal{J}_{\mathrm{AL}}$, evaluated at $(\boldsymbol{x}^*, \boldsymbol{\lambda}^*) = (\boldsymbol{x}^*, \boldsymbol{\lambda}_{\mathcal{A}_{\boldsymbol{x}}}^*, \mathbf{0})$ and acting on the partitioned state $(\boldsymbol{x}_t, \boldsymbol{\lambda}_{t,\mathcal{A}_{\boldsymbol{x}}}, \boldsymbol{\lambda}_{t,I})$, is

$$
\mathcal{J}_{\mathrm{AL}} = \begin{pmatrix} \mathbb{I} - \eta_{\boldsymbol{x}}(A + c\, B^\top B) & -\eta_{\boldsymbol{x}} B^\top & 0 \\ \eta_{\mathrm{dual}} B (\mathbb{I} - \eta_{\boldsymbol{x}}(A + c\, B^\top B)) & \mathbb{I} - \eta_{\boldsymbol{x}} \eta_{\mathrm{dual}} B B^\top & 0 \\ 0 & 0 & (1 - \eta_{\mathrm{dual}}/c)\mathbb{I} \end{pmatrix}. \tag{88}
$$

$\square$

**Lemma 3.** *The characteristic polynomial of the Jacobian $\mathcal{J}_{AL}$ at $(\boldsymbol{x}^*, \boldsymbol{\lambda}^*)$ from Lemma 2 is*

$$
\chi_{\mathcal{J}_{AL}}(\sigma) = \left(1 - \frac{\eta_{dual}}{c} - \sigma\right)^{m - |\mathcal{A}_{\boldsymbol{x}}|} \det \left( (1-\sigma)^2 \mathbb{I} - \eta_{\boldsymbol{x}}(1-\sigma)A - \eta_{\boldsymbol{x}}(c(1-\sigma) - \eta_{dual}\sigma) B^\top B \right). \tag{89}
$$

*Proof.* The Jacobian $\mathcal{J}_{\mathrm{AL}}$ is block lower-triangular with respect to the partition between the active subsystem $(\boldsymbol{x}, \boldsymbol{\lambda}_{\mathcal{A}_{\boldsymbol{x}}})$ and the inactive subsystem $\boldsymbol{\lambda}_I$. Its characteristic polynomial is therefore the product of the characteristic polynomials of these two diagonal blocks.

**Inactive subsystem.** The bottom-right block corresponding to the dual variables of inactive inequality constraints is

$$
\mathcal{J}_{\mathrm{AL},I} = \left(1 - \frac{\eta_{\mathrm{dual}}}{c}\right) \mathbb{I},
$$

whose characteristic polynomial is

$$\chi_{\mathcal{J}_{\text{AL},I}}(\sigma) = \left(1 - \frac{\eta_{\text{dual}}}{c} - \sigma\right)^{|I_I|}. \tag{90}$$

**Active subsystem.** The Jacobian of the active subsystem is

$$\mathcal{J}_{\text{AL},\mathcal{A}_{\boldsymbol{x}}} = \begin{pmatrix} \mathbb{I} - \eta_{\boldsymbol{x}}(A + c\, B^\top B) & -\eta_{\boldsymbol{x}} B^\top \\ \eta_{\text{dual}} B(\mathbb{I} - \eta_{\boldsymbol{x}}(A + c\, B^\top B)) & \mathbb{I} - \eta_{\boldsymbol{x}}\eta_{\text{dual}} BB^\top \end{pmatrix}. \tag{91}$$

This matrix is similar to

$$\mathcal{J}'_{\text{AL},\mathcal{A}_{\boldsymbol{x}}} = \begin{pmatrix} \mathbb{I} - \eta_{\boldsymbol{x}} A - \eta_{\boldsymbol{x}}(c + \eta_{\text{dual}}) B^\top B & -\eta_{\boldsymbol{x}} B^\top \\ \eta_{\text{dual}} B & \mathbb{I} \end{pmatrix} \tag{92}$$

via the transformation matrix

$$P = \begin{pmatrix} \mathbb{I} & 0 \\ -\eta_{\text{dual}} B & I \end{pmatrix}. \tag{93}$$

The characteristic polynomial of the active subsystem is found by computing the determinant of $\mathcal{J}'_{\text{AL},\mathcal{A}_{\boldsymbol{x}}} - \sigma\mathbb{I}$:

$$\chi_{\mathcal{J}_{\text{AL},\mathcal{A}_{\boldsymbol{x}}}}(\sigma) = \det\begin{pmatrix} \mathbb{I} - \eta_{\boldsymbol{x}} A - \eta_{\boldsymbol{x}}(c + \eta_{\text{dual}}) B^\top B - \sigma\mathbb{I} & -\eta_{\boldsymbol{x}} B^\top \\ \eta_{\text{dual}} B & (1 - \sigma)\mathbb{I} \end{pmatrix}. \tag{94}$$

Let the blocks of the matrix be denoted $M_{11}, M_{12}, M_{21}, M_{22}$. The bottom two blocks, $M_{21} = \eta_{\text{dual}} B$ and $M_{22} = (1 - \sigma)\mathbb{I}$, commute since the identity matrix commutes with any matrix. We can therefore use the block determinant identity $\det(M) = \det(M_{11}M_{22} - M_{12}M_{21})$.

We first compute the product $M_{11}M_{22}$:

$$M_{11}M_{22} = \left((1 - \sigma)\mathbb{I} - \eta_{\boldsymbol{x}} A - \eta_{\boldsymbol{x}}(c + \eta_{\text{dual}}) B^\top B\right)(1 - \sigma)\mathbb{I} \tag{95}$$

$$= (1 - \sigma)^2 \mathbb{I} - \eta_{\boldsymbol{x}}(1 - \sigma)A - \eta_{\boldsymbol{x}}(c + \eta_{\text{dual}})(1 - \sigma) B^\top B. \tag{96}$$

Next, we compute the product $M_{12}M_{21}$:

$$M_{12}M_{21} = (-\eta_{\boldsymbol{x}} B^\top)(\eta_{\text{dual}} B) = -\eta_{\boldsymbol{x}}\eta_{\text{dual}} B^\top B. \tag{97}$$

Subtracting the second product from the first yields the matrix $M_{11}M_{22} - M_{12}M_{21}$:

$$(1 - \sigma)^2 \mathbb{I} - \eta_{\boldsymbol{x}}(1 - \sigma)A - \left[\eta_{\boldsymbol{x}}(c + \eta_{\text{dual}})(1 - \sigma) - \eta_{\boldsymbol{x}}\eta_{\text{dual}}\right] B^\top B. \tag{98}$$

We simplify the coefficient of the $B^\top B$ term:

$$\eta_{\boldsymbol{x}}(c + \eta_{\text{dual}})(1 - \sigma) - \eta_{\boldsymbol{x}}\eta_{\text{dual}} = \eta_{\boldsymbol{x}}\left[c(1 - \sigma) + \eta_{\text{dual}}(1 - \sigma) - \eta_{\text{dual}}\right] = \eta_{\boldsymbol{x}}(c(1 - \sigma) - \eta_{\text{dual}}\sigma). \tag{99}$$

Thus, the characteristic polynomial of the active subsystem is

$$\chi_{\mathcal{J}_{\text{AL},\mathcal{A}_{\boldsymbol{x}}}}(\sigma) = \det\left((1 - \sigma)^2 \mathbb{I} - \eta_{\boldsymbol{x}}(1 - \sigma)A - \eta_{\boldsymbol{x}}(c(1 - \sigma) - \eta_{\text{dual}}\sigma) B^\top B\right). \tag{100}$$

Therefore, the characteristic polynomial of the full Jacobian $\mathcal{J}_{\text{AL}}$ is the product of the polynomials for the active and inactive subsystems, $\chi_{\mathcal{J}_{\text{AL}}}(\sigma) = \chi_{\mathcal{J}_{\text{AL},I}}(\sigma) \cdot \chi_{\mathcal{J}_{\text{AL},\mathcal{A}_{\boldsymbol{x}}}}(\sigma)$, which gives the final result.

$\square$

**Lemma 4.** *Consider dual-first alternating gradient descent–projected optimistic ascent on the Lagrangian, analogous to Eq.* (Lag-GD-OA)*, applied to a problem with inequality constraints only:*

$$\boldsymbol{\lambda}_{t+1} \leftarrow \left[\boldsymbol{\lambda}_t + \eta_{\text{dual}}\, \boldsymbol{g}(\boldsymbol{x}_t) + \omega\left(\boldsymbol{g}(\boldsymbol{x}_t) - \boldsymbol{g}(\boldsymbol{x}_{t-1})\right)\right]_+ \tag{101}$$

$$\boldsymbol{x}_{t+1} \leftarrow \boldsymbol{x}_t - \eta_{\boldsymbol{x}}\left[\nabla f(\boldsymbol{x}_t) + \boldsymbol{\lambda}_{t+1}^\top \nabla \boldsymbol{g}(\boldsymbol{x}_t)\right]. \tag{102}$$

*Let $(\boldsymbol{x}^*, \boldsymbol{\lambda}^*)$ be a stationary (KKT) point of Eq. (CMP) satisfying Assumption 1. The Jacobian of the update operator in Eq. (101), evaluated at $(\boldsymbol{x}^*, \boldsymbol{\lambda}^*)$ and acting on the state*

$$(\boldsymbol{x}_t, \boldsymbol{x}_{t-1}, \boldsymbol{\lambda}_{t,\mathcal{A}_{\boldsymbol{x}}}, \boldsymbol{\lambda}_{t,I}) \mapsto (\boldsymbol{x}_{t+1}, \boldsymbol{x}_t, \boldsymbol{\lambda}_{t+1,A}, \boldsymbol{\lambda}_{t+1,I}),$$

*is*

$$\mathcal{J}_{OG} = \begin{pmatrix} \mathbb{I} - \eta_{\boldsymbol{x}} A - \eta_{\boldsymbol{x}}(\eta_{dual} + \omega) B^\top B & \eta_{\boldsymbol{x}} \omega \, B^\top B & -\eta_{\boldsymbol{x}} B^\top & 0 \\ \mathbb{I} & 0 & 0 & 0 \\ (\eta_{dual} + \omega) B & -\omega \, B & \mathbb{I} & 0 \\ 0 & 0 & 0 & 0 \end{pmatrix}. \tag{103}$$

*Here, as in Lemmas 2 and 3, the multipliers are partitioned as $\boldsymbol{\lambda}_t = (\boldsymbol{\lambda}_{t,\mathcal{A}_{\boldsymbol{x}}}, \boldsymbol{\lambda}_{t,I})$, where the active components are*

$$\boldsymbol{\lambda}_{t,\mathcal{A}_{\boldsymbol{x}}} = \{\, \lambda_i \mid g_i(\boldsymbol{x}^*) = 0, \ \lambda_i^* > 0 \,\},$$

*and the inactive components are*

$$\boldsymbol{\lambda}_{t,I} = \{\, \lambda_i \mid g_i(\boldsymbol{x}^*) < 0, \ \lambda_i^* = 0 \,\}.$$

*Under this same partition of the constraints $\boldsymbol{g}(\boldsymbol{x}) = [\boldsymbol{g}_{\mathcal{A}_{\boldsymbol{x}}}(\boldsymbol{x})^\top, \boldsymbol{g}_I(\boldsymbol{x})^\top]^\top$, the matrices are once more defined as*

$$A = \nabla_{\boldsymbol{x}}^2 \mathcal{L}(\boldsymbol{x}^*, \boldsymbol{\lambda}^*) = \nabla^2 f(\boldsymbol{x}^*) + \sum_{i \in \mathcal{A}_{\boldsymbol{x}}} \lambda_i^* \nabla^2 g_i(\boldsymbol{x}^*), \tag{104}$$

$$B = \nabla \boldsymbol{g}_{\mathcal{A}_{\boldsymbol{x}}}(\boldsymbol{x}^*). \tag{105}$$

*Proof.* The dual–first optimistic ascent method in Eq. (101) alternates between updates of the dual and primal variables, beginning with the dual step. Accordingly, the overall update operator can be written as the composition of the two corresponding operators. Its Jacobian is thus

$$\mathcal{J}_{\mathrm{OG}} = \mathcal{J}_{\boldsymbol{x}} \mathcal{J}_{\boldsymbol{\lambda}}, \tag{106}$$

where $\mathcal{J}_{\boldsymbol{x}}$ and $\mathcal{J}_{\boldsymbol{\lambda}}$ denote the Jacobians of the primal and dual steps, respectively. Since the dual update depends on both $\boldsymbol{x}_t$ and $\boldsymbol{x}_{t-1}$, we evaluate these Jacobians on the augmented state

$$(\boldsymbol{x}_t, \boldsymbol{x}_{t-1}, \boldsymbol{\lambda}_t) \mapsto (\boldsymbol{x}_{t+1}, \boldsymbol{x}_t, \boldsymbol{\lambda}_{t+1}). \tag{107}$$

The primal Jacobian is well defined for any $(\boldsymbol{x}_t, \boldsymbol{x}_{t-1}, \boldsymbol{\lambda}_{t+1})$ with $\boldsymbol{\lambda}_{t+1} \succeq \mathbf{0}$. In contrast, the dual Jacobian is well defined only when

$$\lambda_{t,i} + \eta_{\mathrm{dual}} \, g_i(\boldsymbol{x}_t) + \omega \left( g_i(\boldsymbol{x}_t) - g_i(\boldsymbol{x}_{t-1}) \right) \neq 0. \tag{108}$$

At $(\boldsymbol{x}^*, \boldsymbol{x}^*, \boldsymbol{\lambda}^*)$ this condition holds due to strict complementarity. Indeed,

$$\lambda_i^* + \eta_{\mathrm{dual}} \, g_i(\boldsymbol{x}^*) + \omega \left( g_i(\boldsymbol{x}^*) - g_i(\boldsymbol{x}^*) \right) = \lambda_i^* + \eta_{\mathrm{dual}} \, g_i(\boldsymbol{x}^*) \neq 0. \tag{109}$$

At this point, two cases arise:

- **Active constraints** ($\lambda_i^* > 0$ and $g_i(\boldsymbol{x}^*) = 0$). Then

$$[\lambda_i^* + \eta_{\mathrm{dual}} \, g_i(\boldsymbol{x}^*)]_+ = [\lambda_i^*]_+ = \lambda_i^*. \tag{110}$$

- **Inactive constraints** ($\lambda_i^* = 0$ and $g_i(\boldsymbol{x}^*) < 0$). Then

$$[\lambda_i^* + \eta_{\mathrm{dual}} \, g_i(\boldsymbol{x}^*)]_+ = [\omega \, g_i(\boldsymbol{x}^*)]_+ = 0 = \lambda_i^*. \tag{111}$$

In both cases, the projection condition reduces to whether $\lambda_i^* > 0$, i.e., whether the constraint is active at $\boldsymbol{x}^*$. Thus, we adopt the same partition $\boldsymbol{\lambda}_t = (\boldsymbol{\lambda}_{t,\mathcal{A}_{\boldsymbol{x}}}, \boldsymbol{\lambda}_{t,I})$ as in Lemma 2, and reuse the definitions of $A$ and $B$.

It follows that the Jacobian of the dual update, acting on the augmented and partitioned state $(\boldsymbol{x}_t, \boldsymbol{x}_{t-1}, \boldsymbol{\lambda}_{t,\mathcal{A}_{\boldsymbol{x}}}, \boldsymbol{\lambda}_{t,I}) \mapsto (\boldsymbol{x}_t, \boldsymbol{x}_{t-1}, \boldsymbol{\lambda}_{t+1,A}, \boldsymbol{\lambda}_{t+1,I})$, and evaluated at $(\boldsymbol{x}^*, \boldsymbol{x}^*, \boldsymbol{\lambda}^*_{\mathcal{A}_{\boldsymbol{x}}}, \boldsymbol{\lambda}^*_I)$, is

$$
\mathcal{J}_{\boldsymbol{\lambda}} = \begin{pmatrix} \mathbb{I} & 0 & 0 & 0 \\ 0 & \mathbb{I} & 0 & 0 \\ (\eta_{\text{dual}} + \omega)B & -\omega B & \mathbb{I} & 0 \\ 0 & 0 & 0 & 0 \end{pmatrix}, \tag{112}
$$

while the Jacobian of the primal update,

$$
(\boldsymbol{x}_t, \boldsymbol{x}_{t-1}, \boldsymbol{\lambda}_{t+1,A}, \boldsymbol{\lambda}_{t+1,I}) \mapsto (\boldsymbol{x}_{t+1}, \boldsymbol{x}_t, \boldsymbol{\lambda}_{t+1,A}, \boldsymbol{\lambda}_{t+1,I}), \tag{113}
$$

evaluated at $(\boldsymbol{x}^*, \boldsymbol{x}^*, \boldsymbol{\lambda}^*_{\mathcal{A}_{\boldsymbol{x}}}, \boldsymbol{\lambda}^*_I)$, is

$$
\mathcal{J}_{\boldsymbol{x}} = \begin{pmatrix} \mathbb{I} - \eta_{\boldsymbol{x}}A & 0 & -\eta_{\boldsymbol{x}}B^\top & 0 \\ \mathbb{I} & 0 & 0 & 0 \\ 0 & 0 & \mathbb{I} & 0 \\ 0 & 0 & 0 & \mathbb{I} \end{pmatrix}. \tag{114}
$$

Multiplying the two blocks yields

$$
\mathcal{J}_{\text{OG}} = \begin{pmatrix} \mathbb{I} - \eta_{\boldsymbol{x}}A - \eta_{\boldsymbol{x}}(\eta_{\text{dual}} + \omega)B^\top B & \eta_{\boldsymbol{x}}\omega B^\top B & -\eta_{\boldsymbol{x}}B^\top & 0 \\ \mathbb{I} & 0 & 0 & 0 \\ (\eta_{\text{dual}} + \omega)B & -\omega B & \mathbb{I} & 0 \\ 0 & 0 & 0 & 0 \end{pmatrix}. \tag{115}
$$

$\square$

**Lemma 5.** *The characteristic polynomial of the Jacobian $\mathcal{J}_{OG}$ at $(\boldsymbol{x}^*, \boldsymbol{\lambda}^*)$ from Lemma 4 is*

$$
\chi_{\mathcal{J}_{OG}}(\sigma) = (-\sigma)^{m-|\mathcal{A}_{\boldsymbol{x}}|+d} \det\left((1-\sigma)^2\mathbb{I} - \eta_{\boldsymbol{x}}(1-\sigma)A - \eta_{\boldsymbol{x}}(\omega(1-\sigma) - \eta_{dual}\sigma)B^\top B\right). \tag{116}
$$

*Proof.* The Jacobian $\mathcal{J}_{\text{OG}}$ is block lower-triangular with respect to the partition between the active subsystem $(\boldsymbol{x}_t, \boldsymbol{x}_{t-1}, \boldsymbol{\lambda}_{t,\mathcal{A}_{\boldsymbol{x}}})$ and the inactive subsystem $\boldsymbol{\lambda}_{t,I}$. Its characteristic polynomial is therefore the product of the characteristic polynomials of these two diagonal blocks.

**Inactive subsystem.** The bottom-right block corresponding to the dual variables of inactive inequality constraints is $\mathcal{J}_{\text{OG},I} = 0$. It's characteristic polynomial is

$$
\chi_{\mathcal{J}_{\text{OG},I}}(\sigma) = \det(0 - \sigma\mathbb{I}) = \det(-\sigma\mathbb{I}_{|I_I|}) = (-\sigma)^{|I_I|}. \tag{117}
$$

**Active Subsystem**

The Jacobian of the active subsystem, which maps $(\boldsymbol{x}_t, \boldsymbol{x}_{t-1}, \boldsymbol{\lambda}_{t,\mathcal{A}_{\boldsymbol{x}}}) \mapsto (\boldsymbol{x}_{t+1}, \boldsymbol{x}_t, \boldsymbol{\lambda}_{t+1,A})$, is

$$
\mathcal{J}_{\text{OG},\mathcal{A}_{\boldsymbol{x}}} = \begin{pmatrix} \mathbb{I} - \eta_{\boldsymbol{x}}A - \eta_{\boldsymbol{x}}(\eta_{\text{dual}} + \omega)B^\top B & \eta_{\boldsymbol{x}}\omega B^\top B & -\eta_{\boldsymbol{x}}B^\top \\ \mathbb{I} & 0 & 0 \\ (\eta_{\text{dual}} + \omega)B & -\omega B & \mathbb{I} \end{pmatrix}. \tag{118}
$$

We seek its characteristic polynomial, $\chi_{\mathcal{J}_{\text{OG},\mathcal{A}_{\boldsymbol{x}}}}(\sigma) = \det(\mathcal{J}_{\text{OG},\mathcal{A}_{\boldsymbol{x}}} - \sigma\mathbb{I})$. Let $M = \mathcal{J}_{\text{OG},\mathcal{A}_{\boldsymbol{x}}} - \sigma\mathbb{I}$, with blocks $M_{ij}$:

$$
M = \begin{pmatrix} \mathbb{I} - \eta_{\boldsymbol{x}}A - \eta_{\boldsymbol{x}}(\eta_{\text{dual}} + \omega)B^\top B - \sigma\mathbb{I} & \eta_{\boldsymbol{x}}\omega B^\top B & -\eta_{\boldsymbol{x}}B^\top \\ \mathbb{I} & -\sigma\mathbb{I} & 0 \\ (\eta_{\text{dual}} + \omega)B & -\omega B & (1-\sigma)\mathbb{I} \end{pmatrix}. \tag{119}
$$

To simplify the determinant calculation, we apply a determinant-preserving column operation, $C_2 \leftarrow C_2 + \sigma C_1$, which zeros out the $(2,2)$ block:

$$
\det(M) = \det\begin{pmatrix} M_{11} & M_{12} + \sigma M_{11} & M_{13} \\ \mathbb{I} & -\sigma\mathbb{I} + \sigma\mathbb{I} & 0 \\ M_{31} & M_{32} + \sigma M_{31} & M_{33} \end{pmatrix} = \det\begin{pmatrix} M_{11} & M'_{12} & M_{13} \\ \mathbb{I} & 0 & 0 \\ M_{31} & M'_{32} & M_{33} \end{pmatrix}, \tag{120}
$$

where $M'_{12} = M_{12} + \sigma M_{11}$ and $M'_{32} = M_{32} + \sigma M_{31}$. Swapping the first two block rows (corresponding to $\boldsymbol{x}_{t+1}$ and $\boldsymbol{x}_t$) multiplies the determinant by $(-1)^d$, yielding a block upper-triangular matrix:

$$\chi_{\mathcal{J}_{\text{OG}},\mathcal{A}_{\boldsymbol{x}}}(\sigma) = (-1)^d \det \begin{pmatrix} \mathbb{I} & 0 & 0 \\ M_{11} & M'_{12} & M_{13} \\ M_{31} & M'_{32} & M_{33} \end{pmatrix} = (-1)^d \det(\mathbb{I}) \det \begin{pmatrix} M'_{12} & M_{13} \\ M'_{32} & M_{33} \end{pmatrix}. \tag{121}$$

Since $M_{33} = (1-\sigma)\mathbb{I}$ is a multiple of the identity matrix, it commutes with all other blocks. We can thus compute the determinant of the remaining $2 \times 2$ block matrix as $\det(M'_{12}M_{33} - M_{13}M'_{32})$. Let's compute this product:

$$\begin{aligned}
M'_{12}&M_{33} - M_{13}M'_{32} \\
&= (M_{12} + \sigma M_{11})M_{33} - M_{13}(M_{32} + \sigma M_{31}) \\
&= (1-\sigma)(M_{12} + \sigma M_{11}) - M_{13}(M_{32} + \sigma M_{31}) \\
&= (1-\sigma)\big(\eta_{\boldsymbol{x}}\omega B^\top B + \sigma(\mathbb{I} - \eta_{\boldsymbol{x}}A - \eta_{\boldsymbol{x}}(\eta_{\text{dual}} + \omega)B^\top B - \sigma \mathbb{I})\big) \\
&\quad - (-\eta_{\boldsymbol{x}}B^\top)\big(-\omega B + \sigma(\eta_{\text{dual}} + \omega)B\big) \\
&= \sigma(1-\sigma)(1-\sigma)\mathbb{I} - \eta_{\boldsymbol{x}}\sigma(1-\sigma)A \\
&\quad + \big[(1-\sigma)\eta_{\boldsymbol{x}}\omega - \sigma(1-\sigma)\eta_{\boldsymbol{x}}(\eta_{\text{dual}} + \omega) - \eta_{\boldsymbol{x}}\omega + \sigma\eta_{\boldsymbol{x}}(\eta_{\text{dual}} + \omega)\big]B^\top B \\
&= \sigma(1-\sigma)^2\mathbb{I} - \eta_{\boldsymbol{x}}\sigma(1-\sigma)A \\
&\quad + \eta_{\boldsymbol{x}}\big[\omega - \sigma\omega - \sigma\eta_{\text{dual}} - \sigma\omega + \sigma^2\eta_{\text{dual}} + \sigma^2\omega - \omega + \sigma\eta_{\text{dual}} + \sigma\omega\big]B^\top B \\
&= \sigma(1-\sigma)^2\mathbb{I} - \eta_{\boldsymbol{x}}\sigma(1-\sigma)A + \eta_{\boldsymbol{x}}\big[\sigma^2\eta_{\text{dual}} - \sigma\omega(1-\sigma)\big]B^\top B \\
&= \sigma(1-\sigma)^2\mathbb{I} - \eta_{\boldsymbol{x}}\sigma(1-\sigma)A - \eta_{\boldsymbol{x}}\sigma\big(\omega(1-\sigma) - \eta_{\text{dual}}\sigma\big)B^\top B.
\end{aligned}$$

The determinant of this $d \times d$ matrix is:

$$\det(M'_{12}M_{33} - M_{13}M'_{32}) = \sigma^d \det\big((1-\sigma)^2\mathbb{I} - \eta_{\boldsymbol{x}}(1-\sigma)A - \eta_{\boldsymbol{x}}\big(\omega(1-\sigma) - \eta_{\text{dual}}\sigma\big)B^\top B\big). \tag{122}$$

Substituting this back into the expression for $\chi_{\mathcal{J}_{\text{OG}},\mathcal{A}_{\boldsymbol{x}}}(\sigma)$:

$$\chi_{\mathcal{J}_{\text{OG}},\mathcal{A}_{\boldsymbol{x}}}(\sigma) = (-1)^d \cdot \sigma^d \det(\dots) \tag{123}$$

$$= (-\sigma)^d \det\big((1-\sigma)^2\mathbb{I} - \eta_{\boldsymbol{x}}(1-\sigma)A - \eta_{\boldsymbol{x}}\big(\omega(1-\sigma) - \eta_{\text{dual}}\sigma\big)B^\top B\big). \tag{124}$$

The characteristic polynomial of the full Jacobian $\mathcal{J}_{\text{OG}}$ is the product of the polynomials for the active and inactive subsystems, $\chi_{\mathcal{J}_{\text{OG}}}(\sigma) = \chi_{\mathcal{J}_{\text{OG}},I}(\sigma) \cdot \chi_{\mathcal{J}_{\text{OG}},\mathcal{A}_{\boldsymbol{x}}}(\sigma)$, which gives the final result. $\square$

***Proof of Theorem** 2.* Consider the following algorithms for solving inequality-constrained problems:

① Eq. (AL-GDA): Primal-first alternating gradient descent–projected gradient ascent on the Augmented Lagrangian,

$$\begin{aligned}
\boldsymbol{x}_{t+1} &\leftarrow \boldsymbol{x}_t - \eta_{\boldsymbol{x}}\left[\nabla f(\boldsymbol{x}_t) + \big[\boldsymbol{\lambda}_t + c\,\boldsymbol{g}(\boldsymbol{x}_t)\big]_+^\top \nabla \boldsymbol{g}(\boldsymbol{x}_t)\right] \\
\boldsymbol{\lambda}_{t+1} &\leftarrow \left(1 - \frac{\eta_{\text{dual}}}{c}\right)\boldsymbol{\lambda}_t + \frac{\eta_{\text{dual}}}{c}\big[\boldsymbol{\lambda}_t + c\,\boldsymbol{g}(\boldsymbol{x}_{t+1})\big]_+,
\end{aligned} \tag{125}$$

where $0 < \eta_{\text{dual}} \leq c$.

② Eq. (Lag-GD-OA): dual-first alternating gradient descent–projected optimistic ascent on the Lagrangian,

$$\begin{aligned}
\boldsymbol{\lambda}_{t+1} &\leftarrow \big[\boldsymbol{\lambda}_t + \eta_{\text{dual}}\,\boldsymbol{g}(\boldsymbol{x}_t) + \omega\big(\boldsymbol{g}(\boldsymbol{x}_t) - \boldsymbol{g}(\boldsymbol{x}_{t-1})\big)\big]_+ \\
\boldsymbol{x}_{t+1} &\leftarrow \boldsymbol{x}_t - \eta_{\boldsymbol{x}}\big[\nabla f(\boldsymbol{x}_t) + \boldsymbol{\lambda}_{t+1}^\top \nabla \boldsymbol{g}(\boldsymbol{x}_t)\big],
\end{aligned} \tag{126}$$

where $\omega > 0$.

We will analyze the local convergence properties of both algorithms at a stationary (KKT) point $(\boldsymbol{x}^*, \boldsymbol{\lambda}^*)$, satisfying Assumption 1, and operating on a partition of the $\boldsymbol{\lambda} = (\boldsymbol{\lambda}_A, \boldsymbol{\lambda}_I)$ state into those corresponding to active and inactive constraints at $\boldsymbol{x}^*$:

$$\boldsymbol{\lambda}_{t,\mathcal{A}_{\boldsymbol{x}}} = \{ \lambda_i \mid g_i(\boldsymbol{x}^*) = 0, \ \lambda_i^* > 0 \},$$

and inactive ones

$$\boldsymbol{\lambda}_{t,I} = \{ \lambda_i \mid g_i(\boldsymbol{x}^*) < 0, \ \lambda_i^* = 0 \}.$$

Let the matrices $A$ and $B$ be defined at the stationary point $(\boldsymbol{x}^*, \boldsymbol{\lambda}^*)$ as

$$A = \nabla_{\boldsymbol{x}}^2 \mathcal{L}(\boldsymbol{x}^*, \boldsymbol{\lambda}^*) \quad \text{and} \quad B = \nabla \boldsymbol{g}_{\mathcal{A}_{\boldsymbol{x}}}(\boldsymbol{x}^*), \tag{127}$$

where $A$ is the **Hessian of the Lagrangian** with respect to $\boldsymbol{x}$ and $B$ is the **Jacobian of the active constraints**.

From Lemmas 2 and 3, we know that the Jacobian $\mathcal{J}_{\mathrm{AL}}$ of the Augmented Lagrangian updates in Eq. (125), acting on the partitioned state $(\boldsymbol{x}_t, \boldsymbol{\lambda}_{t,\mathcal{A}_{\boldsymbol{x}}}, \boldsymbol{\lambda}_{t,I}) \mapsto (\boldsymbol{x}_{t+1}, \boldsymbol{\lambda}_{t+1,A}, \boldsymbol{\lambda}_{t+1,I})$ and evaluated at $(\boldsymbol{x}^*, \boldsymbol{\lambda}^*) = (\boldsymbol{x}^*, \boldsymbol{\lambda}_{\mathcal{A}_{\boldsymbol{x}}}^*, \boldsymbol{0})$, is

$$\mathcal{J}_{\mathrm{AL}} = \begin{pmatrix} \mathbb{I} - \eta_{\boldsymbol{x}}(A + c\, B^\top B) & -\eta_{\boldsymbol{x}} B^\top & 0 \\ \eta_{\mathrm{dual}} B(\mathbb{I} - \eta_{\boldsymbol{x}}(A + c\, B^\top B)) & \mathbb{I} - \eta_{\boldsymbol{x}} \eta_{\mathrm{dual}} B B^\top & 0 \\ 0 & 0 & (1 - \eta_{\mathrm{dual}}/c)\mathbb{I} \end{pmatrix}; \tag{128}$$

and that the characteristic polynomial for this Jacobian is:

$$\chi_{\mathcal{J}_{\mathrm{AL}}}(\sigma) = \left(1 - \frac{\eta_{\mathrm{dual}}}{c} - \sigma\right)^{m-|\mathcal{A}_{\boldsymbol{x}}|} \det\left((1-\sigma)^2 \mathbb{I} - \eta_{\boldsymbol{x}}(1-\sigma)A - \eta_{\boldsymbol{x}}(c(1-\sigma) - \eta_{\mathrm{dual}}\sigma)B^\top B\right). \tag{129}$$

Moreover, from Lemmas 4 and 5, we know that for the same choices of $A$ and $B$ matrices, it follows that the Jacobian $\mathcal{J}_{\mathrm{OG}}$ of the dual optimistic ascent updates in Eq. (126), acting on the partitioned state $(\boldsymbol{x}_t, \boldsymbol{x}_{t-1}, \boldsymbol{\lambda}_{t,\mathcal{A}_{\boldsymbol{x}}}, \boldsymbol{\lambda}_{t,I}) \mapsto (\boldsymbol{x}_{t+1}, \boldsymbol{x}_t, \boldsymbol{\lambda}_{t+1,A}, \boldsymbol{\lambda}_{t+1,I})$ and evaluated at $(\boldsymbol{x}^*, \boldsymbol{\lambda}^*) = (\boldsymbol{x}^*, \boldsymbol{x}^*, \boldsymbol{\lambda}_{\mathcal{A}_{\boldsymbol{x}}}^*, \boldsymbol{0})$, is

$$\mathcal{J}_{\mathrm{OG}} = \begin{pmatrix} \mathbb{I} - \eta_{\boldsymbol{x}} A - \eta_{\boldsymbol{x}}(\eta_{\mathrm{dual}} + \omega)B^\top B & \eta_{\boldsymbol{x}} \omega\, B^\top B & -\eta_{\boldsymbol{x}} B^\top & 0 \\ \mathbb{I} & 0 & 0 & 0 \\ (\eta_{\mathrm{dual}} + \omega)B & -\omega\, B & \mathbb{I} & 0 \\ 0 & 0 & 0 & 0 \end{pmatrix}, \tag{130}$$

and that the characteristic polynomial for this Jacobian is:

$$\chi_{\mathcal{J}_{\mathrm{OG}}}(\sigma) = (-\sigma)^{|I_I|+d} \det\left((1-\sigma)^2 \mathbb{I} - \eta_{\boldsymbol{x}}(1-\sigma)A - \eta_{\boldsymbol{x}}(\omega(1-\sigma) - \eta_{\mathrm{dual}}\sigma)B^\top B\right), \tag{131}$$

**Remark on the $\sigma = 1$ case** A subtle but important point is that, although the structure of these characteristic polynomials might suggest $\sigma = 1$ could be a root if $\det(B^\top B) = 0$, it can be shown that $\sigma = 1$ is in fact not an eigenvalue of either $\mathcal{J}_{\mathrm{AL}}$ or $\mathcal{J}_{\mathrm{OG}}$ under Assumptions 1 to 3—conditions that are generally necessary, though not sufficient, for convergence.

We establish the equivalence of the local convergence properties of the two algorithms by setting the optimistic parameter $\omega = c$ and comparing the characteristic polynomials of their Jacobians.

Upon setting $\omega = c$ in $\chi_{\mathcal{J}_{\mathrm{OG}}}(\sigma)$, the non-trivial determinant term becomes identical to that in $\chi_{\mathcal{J}_{\mathrm{AL}}}(\sigma)$:

$$\det\left((1-\sigma)^2 \mathbb{I} - \eta_{\boldsymbol{x}}(1-\sigma)A - \eta_{\boldsymbol{x}}(c(1-\sigma) - \eta_{\mathrm{dual}}\sigma)B^\top B\right). \tag{132}$$

The roots of this determinant define the set of non-trivial eigenvalues, which are shared between both methods. Local convergence of either algorithm depends on whether all these shared eigenvalues lie within the unit circle.

The remaining eigenvalues are given by the roots of the other factors in each polynomial:

- For the Augmented Lagrangian method, the factor $\left(1 - \frac{\eta_{\text{dual}}}{c} - \sigma\right)^{|I_I|}$ yields $|I_I|$ eigenvalues at $\sigma = 1 - \eta_{\text{dual}}/c$. Given the condition $0 < \eta_{\text{dual}} \leq c$, these eigenvalues lie in the interval $(0, 1)$, and are thus stable.

- For the optimistic ascent method, the factor $(-\sigma)^{|I_I|+d}$ yields $|I_I| + d$ eigenvalues at $\sigma = 0$, which are also stable.

**Conclusion**. Since the non-trivial eigenvalues governing convergence are identical for both algorithms, and the remaining trivial eigenvalues for both are stable, the spectral radius $\rho(\mathcal{J}_{\text{AL}}) < 1$ if and only if $\rho(\mathcal{J}_{\text{OG}}) < 1$. Therefore, with the choice of $\omega = c$, one algorithm converges locally if and only if the other one does.

Moreover, it holds that:

$$\rho(\mathcal{J}_{\text{AL}}) = \max\{\rho(\mathcal{J}_{\text{OG}}), 1 - \eta_{\text{dual}}/c\}. \tag{133}$$

This implies that if the shared eigenvalues dictate the convergence rate (i.e., when $1 > \rho(\mathcal{J}_{\text{OG}}) \geq 1 - \eta_{\text{dual}}/c$), both algorithms converge at the same local rate. $\qquad\square$

### D.3 FURTHER PROOFS

This subsection includes the proofs for Prop. 3 (ALM finds all constrained minimizers) and Prop. 4 (ALM finds only constrained minimizers).

***Proof of Proposition 3***. Let $\boldsymbol{\lambda}^* \succeq \mathbf{0}$ and $\boldsymbol{\mu}^*$ be the optimal Lagrange multipliers for $\boldsymbol{x}^*$ such that $(\boldsymbol{x}^*, \boldsymbol{\lambda}^*, \boldsymbol{\mu}^*)$ form a KKT point of Eq. (CMP). Since $\boldsymbol{x}^*$ satisfies Assumption 3, these multipliers exist and are unique.

By Lemma 2, the Jacobian of algorithm Eq. (AL-GDA), acting on the partitioned state $(\boldsymbol{x}_t, \boldsymbol{\lambda}_{t,\mathcal{A}_{\boldsymbol{x}}}, \boldsymbol{\lambda}_{t,I}) \mapsto (\boldsymbol{x}_{t+1}, \boldsymbol{\lambda}_{t+1,A}, \boldsymbol{\lambda}_{t+1,I})$ and evaluated at $(\boldsymbol{x}^*, \boldsymbol{\lambda}^*) = (\boldsymbol{x}^*, \boldsymbol{\lambda}^*_{\mathcal{A}_{\boldsymbol{x}}}, \mathbf{0})$, is

$$\mathcal{J}_{\text{AL}} = \begin{pmatrix} \mathbb{I} - \eta_{\boldsymbol{x}}(A + c\,B^\top B) & -\eta_{\boldsymbol{x}}B^\top & 0 \\ \eta_{\text{dual}}B(\mathbb{I} - \eta_{\boldsymbol{x}}(A + c\,B^\top B)) & \mathbb{I} - \eta_{\boldsymbol{x}}\eta_{\text{dual}}BB^\top & 0 \\ 0 & 0 & (1 - \eta_{\text{dual}}/c)\mathbb{I} \end{pmatrix}, \tag{134}$$

where

$$A = \nabla^2_{\boldsymbol{x}}\mathcal{L}(\boldsymbol{x}^*, \boldsymbol{\lambda}^*), \qquad B = \nabla \boldsymbol{g}_{\mathcal{A}_{\boldsymbol{x}}}(\boldsymbol{x}^*). \tag{135}$$

This Jacobian is well-defined thanks to Assumption 1.

We now proceed by grouping equality constraints with active inequality constraints, considering the Jacobian on the state partitioned as $(\boldsymbol{x}_t, (\boldsymbol{\lambda}_{t,\mathcal{A}_{\boldsymbol{x}}}, \boldsymbol{\mu}_t), \boldsymbol{\lambda}_{t,I})$. This yields the same Jacobian $\mathcal{J}_{\text{AL}}$, except $B$ is redefined as

$$A = \nabla^2_{\boldsymbol{x}}\mathcal{L}(\boldsymbol{x}^*, \boldsymbol{\lambda}^*, \boldsymbol{\mu}^*), \qquad B = \begin{pmatrix} \nabla \boldsymbol{g}_{\mathcal{A}_{\boldsymbol{x}}}(\boldsymbol{x}^*) \\ \nabla \boldsymbol{h}(\boldsymbol{x}^*) \end{pmatrix}. \tag{136}$$

We will now prove that under Assumptions 2 and 3, there exist $c > 0$ large enough and $\eta_{\boldsymbol{x}}, \eta_{\text{dual}} > 0$ small enough such that all eigenvalues of $\mathcal{J}_{\text{AL}}$ lie within the unit circle.

$\mathcal{J}_{\text{AL}}$ is block lower-triangular, so its eigenvalues are the union of the eigenvalues of its blocks. The eigenvalues of the inactive inequality block are $\sigma = 1 - \eta_{\text{dual}}/c$. Under the assumption of Eq. (AL-GDA) that $\eta_{\text{dual}} \leq c$, these satisfy $|\sigma| < 1$.

Setting $\eta = \eta_{\boldsymbol{x}} = \eta_{\text{dual}}$ for simplicity, the upper-left block is

$$\mathcal{J}_{\text{AL},\mathcal{A}_{\boldsymbol{x}}} = \begin{pmatrix} \mathbb{I} - \eta(A + c\,B^\top B) & -\eta B^\top \\ \eta B(\mathbb{I} - \eta(A + c\,B^\top B)) & \mathbb{I} - \eta^2 BB^\top \end{pmatrix}. \tag{137}$$

As in Lemma 3, $\mathcal{J}_{\text{AL},\mathcal{A}_{\boldsymbol{x}}}$ is similar to

$$\mathcal{J}'_{\text{AL},\mathcal{A}_{\boldsymbol{x}}} = \begin{pmatrix} \mathbb{I} - \eta A - \eta(c+\eta)B^\top B & -\eta B^\top \\ \eta B & \mathbb{I} \end{pmatrix} \tag{138}$$

$$= \mathbb{I} - \eta \begin{pmatrix} A + (c+\eta)B^\top B & B^\top \\ -B & 0 \end{pmatrix} \tag{139}$$

$$= \mathbb{I} - \eta M. \tag{140}$$

By Prop. 2, under Assumptions 2 and 3, there exists $c > 0$ such that $A + cB^\top B$ is positive definite. Hence $A + (c+\eta)B^\top B$ is also positive definite. By Bertsekas (2016, Prop. 5.4.2), $M$ has eigenvalues with positive real parts, so for sufficiently small $\eta > 0$, all eigenvalues of $\mathcal{J}'_{\text{AL},\mathcal{A}_{\boldsymbol{x}}}$ satisfy $|\sigma| < 1$.

Combining this with the inactive block, we conclude $\rho(\mathcal{J}_{\text{AL}}) < 1$, completing the proof. $\qquad\square$

***Proof of Proposition 4.*** Let $(\boldsymbol{x}^*, \boldsymbol{\lambda}^*, \boldsymbol{\mu}^*)$ be an LSSP of the algorithm in Eq. (AL-GDA). By Prop. 1, $(\boldsymbol{x}^*, \boldsymbol{\lambda}^*, \boldsymbol{\mu}^*)$ is a KKT point of Eq. (CMP). Moreover, we know that $\rho(\mathcal{J}_{\text{AL}}) < 1$, where

$$\mathcal{J}_{\text{AL}} = \begin{pmatrix} \mathbb{I} - \eta_{\boldsymbol{x}}(A + c B^\top B) & -\eta_{\boldsymbol{x}}B^\top & 0 \\ \eta_{\text{dual}}B(\mathbb{I} - \eta_{\boldsymbol{x}}(A + c B^\top B)) & \mathbb{I} - \eta_{\boldsymbol{x}}\eta_{\text{dual}}BB^\top & 0 \\ 0 & 0 & (1 - \eta_{\text{dual}}/c)\mathbb{I} \end{pmatrix}, \tag{141}$$

and

$$A = \nabla_{\boldsymbol{x}}^2 \mathcal{L}(\boldsymbol{x}^*, \boldsymbol{\lambda}^*, \boldsymbol{\mu}^*), \qquad B = \begin{pmatrix} \nabla \boldsymbol{g}_{\mathcal{A}_{\boldsymbol{x}}}(\boldsymbol{x}^*) \\ \nabla \boldsymbol{h}(\boldsymbol{x}^*) \end{pmatrix}, \tag{142}$$

as in the *Proof of Proposition 3*. The assumption $\rho(\mathcal{J}_{\text{AL}}) < 1$ implies that $\rho(\mathcal{J}'_{\text{AL},\mathcal{A}_{\boldsymbol{x}}}) < 1$, where

$$\mathcal{J}'_{\text{AL},\mathcal{A}_{\boldsymbol{x}}} = \begin{pmatrix} \mathbb{I} - \eta A - \eta(c+\eta)B^\top B & -\eta B^\top \\ \eta B & \mathbb{I} \end{pmatrix} \tag{143}$$

is similar to the upper-left block of $\mathcal{J}_{\text{AL}}$. Here we set $\eta = \eta_{\boldsymbol{x}} = \eta_{\text{dual}}$ for simplicity.

We proceed by contradiction. Suppose that $\boldsymbol{x}^*$ is not a local constrained minimizer of Eq. (CMP). Then $(\boldsymbol{x}^*, \boldsymbol{\lambda}^*, \boldsymbol{\mu}^*)$ must violate the second-order sufficiency conditions of Assumption 2 (Bertsekas, 2016, Prop. 4.3.2).

This implies the existence of a nonzero vector $\boldsymbol{d} \in \mathbb{R}^d$ in the null space of $B$ such that

$$\boldsymbol{d}^\top A \boldsymbol{d} \le 0. \tag{144}$$

By the Rayleigh-Ritz theorem, choose $\boldsymbol{d}$ as a normalized eigenvector ($\|\boldsymbol{d}\| = 1$) corresponding to the smallest eigenvalue, $\alpha$, of $A$ restricted to the subspace $\{\boldsymbol{d} \mid B\boldsymbol{d} = 0\}$. By the assumption, $\alpha \le 0$.

Consider the action of $\mathcal{J}'_{\text{AL},\mathcal{A}_{\boldsymbol{x}}}$ on the vector $[\boldsymbol{d}^\top, 0]^\top$:

$$\mathcal{J}'_{\text{AL},\mathcal{A}_{\boldsymbol{x}}} \begin{pmatrix} \boldsymbol{d} \\ 0 \end{pmatrix} = \begin{pmatrix} (\mathbb{I} - \eta A - \eta(c+\eta)B^\top B)\boldsymbol{d} \\ \eta B\boldsymbol{d} \end{pmatrix} = \begin{pmatrix} \boldsymbol{d} - \eta A\boldsymbol{d} \\ 0 \end{pmatrix}, \tag{145}$$

using $B\boldsymbol{d} = 0$.

Then

$$\left\| \mathcal{J}'_{\text{AL},\mathcal{A}_{\boldsymbol{x}}} \begin{pmatrix} \boldsymbol{d} \\ 0 \end{pmatrix} \right\|^2 = \|\boldsymbol{d} - \eta A\boldsymbol{d}\|^2 = \|\boldsymbol{d}\|^2 - 2\eta(\boldsymbol{d}^\top A\boldsymbol{d}) + \eta^2\|A\boldsymbol{d}\|^2 = 1 - 2\eta\alpha + \eta^2\|A\boldsymbol{d}\|^2 \ge 1, \tag{146}$$

since $\alpha \le 0$ and $\|\boldsymbol{d}\|^2 = 1$.

Thus, we have found a vector $[\boldsymbol{d}^\top, 0]^\top$ with unit norm for which $\mathcal{J}'_{\text{AL},\mathcal{A}_{\boldsymbol{x}}}$ is non-contractive, contradicting the assumption $\rho(\mathcal{J}_{\text{AL}}) < 1$.

Therefore, our initial assumption that Assumption 2 is violated must be false.

$\qquad\square$

***Proof of Proposition*** 5. This result adapts the spectral analysis of Benzi and Simoncini (2006), which assumes simultaneous updates, to our alternating-update setting. Moreover, we explicitly derives the decay rate of the imaginary parts.

By Corollary 1, the eigenvalues of $\mathcal{J}_{\mathrm{OG}}$ are either trivial zeros—which are always real—or coincide with the non-trivial eigenvalues of the Augmented Lagrangian Jacobian $\mathcal{J}_{\mathrm{AL}}$. These non-trivial eigenvalues lie in the active subsystem (which we consider as including the equality constraints), denoted $\mathcal{J}_{\mathrm{AL},\mathcal{A}_{\boldsymbol{x}}}$. Setting a common learning rate $\eta = \eta_{\boldsymbol{x}} = \eta_{\mathrm{dual}}$ for simplicity, we note that $\mathcal{J}_{\mathrm{AL},\mathcal{A}_{\boldsymbol{x}}}$ is similar to $\mathcal{J}'_{\mathrm{AL},\mathcal{A}_{\boldsymbol{x}}}$ (as in the proof of Lemma 2), where

$$\mathcal{J}'_{\mathrm{AL},\mathcal{A}_{\boldsymbol{x}}}(\omega) = I - \eta\,M(\omega), \tag{147}$$

with

$$M(\omega) = \begin{pmatrix} A + (\omega + \eta)B^{\top}B & B^{\top} \\ -B & 0 \end{pmatrix}, \tag{148}$$

where $A$ and $B$ are defined as in the *Proof of Proposition 3*:

$$A = \nabla_{\boldsymbol{x}}^2 \mathcal{L}(\boldsymbol{x}^*, \boldsymbol{\lambda}^*, \boldsymbol{\mu}^*), \qquad B = \begin{pmatrix} \nabla\boldsymbol{g}_{\mathcal{A}_{\boldsymbol{x}}}(\boldsymbol{x}^*) \\ \nabla\boldsymbol{h}(\boldsymbol{x}^*) \end{pmatrix}. \tag{149}$$

Consequently, the eigenvalues of $\mathcal{J}_{\mathrm{AL}}(\omega)$ are real if and only if the eigenvalues of $M(\omega)$ are real.

Let $\sigma$ be an eigenvalue of $M(\omega)$ and $z = [u^{\top}, v^{\top}]^{\top} \neq 0$ be its eigenvector. First, consider the case $u = 0$. The eigenvalue equations $Mz = \sigma z$ become:

$$B^{\top}v = 0 \quad \text{and} \quad 0 = \sigma v. \tag{150}$$

If $\sigma \neq 0$, the second equation forces $v = 0$—so $z = 0$, a contradiction. Hence any eigenpair with $u = 0$ must have $\sigma = 0$, which is trivially real, and we therefore disregard this case and assume $u \neq 0$ from now on.

We normalize such that $u^*u = 1$. As shown in Benzi and Simoncini (2006, Prop. 2.5), $\sigma$ satisfies:

$$\sigma^2 - \sigma\,u^*\big(A + (\omega + \eta)B^{\top}B\big)u + u^*B^{\top}Bu = 0. \tag{151}$$

These eigenvalues are real if and only if the discriminant $\Delta_u(\omega) \geq 0$, where:

$$\Delta_u(\omega) = \big[u^*(A + (\omega + \eta)B^{\top}B)u\big]^2 - 4\,u^*B^{\top}Bu. \tag{152}$$

As a function of $\omega$, $\Delta_u(\omega)$ is a quadratic with a positive leading coefficient $(u^*B^{\top}Bu)^2$ whenever $Bu \neq 0$—hence, it is convex in $\omega$. Note that if $Bu = 0$, then $B^{\top}Bu = 0$ and (151) reduces to

$$\sigma^2 - \sigma\,u^*Au = 0, \tag{153}$$

whose roots are real for all $\omega$. We thus focus on directions where $Bu \neq 0$.

We now demonstrate the existence of a single finite threshold $\bar{\omega}$ such that for every eigenvector direction with $Bu \neq 0$ and every $\omega \geq \bar{\omega}$, the discriminant satisfies $\Delta_u(\omega) \geq 0$.

Let

$$a(u) = u^*(A + \eta B^{\top}B)u, \qquad b(u) = u^*B^{\top}Bu. \tag{154}$$

Then for all $\omega$,

$$u^*(A + (\omega + \eta)B^{\top}B)u = a(u) + \omega b(u), \tag{155}$$

and the discriminant simplifies to:

$$\Delta_u(\omega) = \big(a(u) + \omega b(u)\big)^2 - 4b(u). \tag{156}$$

Since $Bu \neq 0$, we have $b(u) > 0$.

Let $\gamma_{\min}$ and $\gamma_{\max}$ denote the smallest and largest nonzero eigenvalues of $B^{\top}B$, and let $\alpha$ be the largest eigenvalue of $A + \eta B^{\top}B$ (restricted to the relevant subspace). By Assumption 3, $\gamma_{\min} > 0$.

Then, for any unit $u$ with $Bu \neq 0$,

$$\gamma_{\min} \leq b(u) \leq \gamma_{\max} \quad \text{and} \quad |a(u)| \leq \alpha. \tag{157}$$

Define

$$\bar{\omega} := \frac{\alpha + 2\sqrt{\gamma_{\max}}}{\gamma_{\min}}. \tag{158}$$

For any $\omega \geq \bar{\omega}$ and any unit $u$ with $Bu \neq 0$, we have

$$\omega b(u) - |a(u)| \ \geq \ \omega\gamma_{\min} - \alpha \ \geq \ \bar{\omega}\gamma_{\min} - \alpha = 2\sqrt{\gamma_{\max}} \ \geq \ 2\sqrt{b(u)}, \tag{159}$$

where we used $b(u) \leq \gamma_{\max}$. Hence,

$$|a(u) + \omega b(u)| \ \geq \ \omega b(u) - |a(u)| \ \geq \ 2\sqrt{b(u)}. \tag{160}$$

Substituting this bound into the discriminant gives

$$\Delta_u(\omega) = \big(a(u) + \omega b(u)\big)^2 - 4b(u) \ \geq \ (2\sqrt{b(u)})^2 - 4b(u) = 0. \tag{161}$$

Thus, for all $\omega \geq \bar{\omega}$ and all unit vectors $u$ with $Bu \neq 0$, the discriminant is nonnegative and the corresponding eigenvalues $\sigma$ are real. Hence all eigenvalues of $M(\omega)$ are real for every $\omega \geq \bar{\omega}$, and therefore all non-trivial eigenvalues of $\mathcal{J}_{\mathrm{AL}}$ and $\mathcal{J}_{\mathrm{OG}}$ are real as well.

To study the decay rate of the imaginary parts as $\omega \uparrow \bar{\omega}$, consider a complex conjugate pair of eigenvalues $\sigma(\omega), \overline{\sigma(\omega)}$ of $M(\omega)$ that becomes real at some $\omega^\star \leq \bar{\omega}$. At such a point, we must have

$$\Delta_{u(\omega^\star)}(\omega^\star) = 0. \tag{162}$$

We assume a generic non-degeneracy condition (ND): the corresponding eigenvalue is simple and

$$\frac{\partial}{\partial\omega}\Delta_{u(\omega)}(\omega)\Big|_{\omega=\omega^\star} \neq 0. \tag{163}$$

Under (ND), $\Delta_{u(\omega)}(\omega)$ has a simple root at $\omega^\star$, and a first-order Taylor expansion around $\omega^\star$ yields

$$\Delta_{u(\omega)}(\omega) = K(\omega - \omega^\star) + \mathcal{O}\big((\omega - \omega^\star)^2\big), \qquad K \neq 0. \tag{164}$$

For $\omega < \omega^\star$ sufficiently close to $\omega^\star$, we have $\Delta_{u(\omega)}(\omega) < 0$. Substituting this into (151) yields a pair of complex conjugate roots whose imaginary parts satisfy

$$|\Im\sigma(\omega)| = \tfrac{1}{2}\sqrt{-\Delta_{u(\omega)}(\omega)} = \mathcal{O}\big(\sqrt{\omega^\star - \omega}\big) \quad \text{as } \omega \uparrow \omega^\star. \tag{165}$$

Since there are finitely many eigenvalue branches—and by construction, $\bar{\omega}$ is the minimal threshold above which all eigenvalues are real—this bound is uniform over all complex eigenvalues of $M(\omega)$ for $\omega$ sufficiently close to $\bar{\omega}$:

$$\max_{\sigma(\omega)\in\mathrm{spec}(M(\omega))} |\Im\sigma(\omega)| = \mathcal{O}\big(\sqrt{\bar{\omega} - \omega}\big) \quad \text{as } \omega \uparrow \bar{\omega}. \tag{166}$$

Finally, the eigenvalues of $\mathcal{J}_{\mathrm{AL}}(\omega)$ are of the form $1 - \eta\sigma(\omega)$. Since the non-trivial eigenvalues of $\mathcal{J}_{\mathrm{OG}}(\omega)$ coincide with those of $\mathcal{J}_{\mathrm{AL}}(\omega)$, we conclude:

$$\max_{\lambda(\omega)\in\mathrm{spec}(\mathcal{J}_{\mathrm{OG}}(\omega))} |\Im\lambda(\omega)| = |\eta| \max_{\sigma(\omega)\in\mathrm{spec}(M(\omega))} |\Im\sigma(\omega)| = \mathcal{O}\big(\sqrt{\bar{\omega} - \omega}\big). \tag{167}$$

$\square$

