# OpenReview forum: "Dual Optimistic Ascent (PI Control) is the Augmented Lagrangian Method in Disguise"
_ICLR.cc/2026/Conference — ICLR 2026 Poster_

### Official Review · Reviewer_2hch · 2025-10-30

**Soundness:** 3
**Presentation:** 3
**Contribution:** 2
**Rating:** 6
**Confidence:** 2

**Summary:**

The paper studies dual-only optimistic ascent applied to the standard Lagrangian. The central result establishes a precise equivalence between this method and gradient descent–ascent on the Augmented Lagrangian (ALM) when the optimism and penalty parameters are aligned. This equivalence holds at the iterate level for equality-constrained problems (under specific initialization) and, for inequality constraints, extends to locally stable stationary points, transferring local linear convergence and damping properties. The work thus generalizes the theoretical foundation of ALM-based gradient methods to the dual-only optimistic ascent framework, explains the empirically observed stabilization effect of dual optimism, and offers practical guidance on parameter tuning (e.g., matching optimism and penalty parameters) through the ALM perspective.

**Strengths:**

This paper reduces dual-only optimistic ascent to the Augmented Lagrangian Method (ALM), establishing a theoretical foundation for its practical use in constrained deep-learning pipelines. The theoretical derivation seems sound.

**Weaknesses:**

The generality of the results appears limited, as the analysis focuses on the specific setting of dual-only optimism with single-step first-order primal updates. Since the reviewer is not deeply familiar with the literature on constrained optimization via dual optimistic ascent, it is somewhat difficult to assess the broader significance of the contribution, especially given that the problem formulation seems rather specialized.

**Questions:**

Could the authors provide further justification or point the reviewer to references explaining why existing theoretical analyses of general min–max optimization are insufficient for this setting? It would also be helpful to clarify what specific advantages the proposed analysis offers over prior frameworks.

---

> ### Author Response · Authors · 2025-11-20
>
> We thank the reviewer for their comments, and for highlighting that our paper establishes a solid theoretical foundation for the popular dual optimistic ascent (PI control) scheme.
>
> ### Weaknesses
>
>
> > The generality of the results appears limited, as the analysis focuses on the specific setting of dual-only optimism with single-step first-order primal updates [...] the problem formulation seems rather specialized.
>
>
> We understand the reviewer’s concern. The formulation we study is indeed specialized, but it is precisely the one used in practice in constrained deep learning. Our goal is to rigorously analyze and contextualize the approach the community has already converged on. In this sense, our paper provides the first formal justification for why this specific setup is so successful.
>
> The community’s preference for this setup is driven by two pragmatic choices:
> * **Dual-only optimism** preserves full flexibility on the primal side: in practice, the primal update is handled by the model’s optimizer of choice (Adam, SGD with momentum, etc.), and dual-only optimism integrates seamlessly with this.
> * **Single-step, first-order primal updates** are the standard in deep learning, reflecting computational efficiency requirements rather than any theoretical consideration.
>
> As we show in the paper, it is exactly in this common-practice regime that dual optimistic ascent becomes algebraically equivalent to an Augmented Lagrangian step and thus inherits the ALM’s desirable properties. **This is the broader significance the reviewer is asking about: the method’s empirical success is not accidental—it works well in practice because it is secretly performing an Augmented Lagrangian step.** Outside this regime (e.g., with multi-step or second-order primal updates, as discussed in our response to **Reviewer GKqh**), this equivalence breaks, and one should generally prefer the explicit ALM formulation.
>
> Finally, although the algorithms may seem specialized, our equivalence results are fairly general, holding for any first-order primal optimizer—including Adam or SGD with momentum—not just vanilla gradient descent. In short, while the formulation is tailored to how constrained deep learning is actually performed, the results remain broadly applicable within that setting and clarify why a widely used method behaves as well as it does.
>
>
> ### Questions
>
> > Could the authors provide further justification or point the reviewer to references explaining why existing theoretical analyses of general min–max optimization are insufficient for this setting?
>
> Most theoretical analyses of general min–max optimization—especially those studying optimistic methods like Optimistic Gradient Descent/Ascent (OGDA)—assume **symmetric algorithms**, where the *same* update rule (e.g., optimism) is applied to both the primal (min) and dual (max) players.
>
> This symmetry is a powerful analytical simplification: it allows the joint update to be viewed as a single operator over the joint variable space, making convergence analyses more tractable.
>
> However, this core assumption **does not hold** in the setting used in constrained deep learning, which is the focus of our paper. There, the setup is fundamentally **asymmetric**: the primal player uses a standard first-order optimizer without optimism (such as SGD or Adam), while the dual player uses optimistic ascent.
>
> Because of this mismatch in assumptions, the theoretical guarantees from symmetric analyses (such as those for OGDA) are insufficient and do not transfer to the “dual-only optimism” setting. In short, existing analyses study a different algorithm—one that is not actually employed in this practical context.
>
> >  It would also be helpful to clarify what specific advantages the proposed analysis offers over prior frameworks.
>
>
> If the reviewer is referring to convergence guarantees, we emphasize that, to the best of our knowledge, **there are no previously established convergence guarantees for dual optimistic ascent (PI control) in constrained optimization**. Prior work on this method has been largely empirical, demonstrating its practical effectiveness but without a theoretical foundation.
>
> In contrast, our analysis:
> - **Provides the first rigorous convergence guarantees for dual optimistic ascent in the constrained setting**, obtained by leveraging its algebraic equivalence to the Augmented Lagrangian Method (ALM).
> - **Clarifies when the method is principled**—specifically, in the single-step, first-order regime where it matches an ALM update; and when it isn’t, in which case one should **prefer the explicit ALM framework** (e.g., for multi-step or second-order variants, where the equivalence breaks).
>
> In this sense, our contribution is not about choosing among existing convergence frameworks, but about giving dual optimistic ascent a solid theoretical footing by connecting it to the well-understood ALM theory.

---

### Official Review · Reviewer_GKqh · 2025-11-01

**Soundness:** 2
**Presentation:** 3
**Contribution:** 2
**Rating:** 4
**Confidence:** 4

**Summary:**

This paper establishes a theoretical equivalence between dual optimistic ascent (PI control) and the Augmented Lagrangian Method (ALM). Specifically, it shows that for equality-constrained problems, dual optimistic ascent on the Lagrangian is identical (in primal iterates) to gradient descent–ascent on the Augmented Lagrangian when the optimism coefficient ω equals the penalty parameter c. For general inequality constraints, the two methods share the same locally stable stationary points (LSSPs), implying the same convergence targets. This equivalence allows transferring ALM’s robust theoretical guarantees—especially linear convergence—to the dual optimistic framework, previously known mainly for empirical success. The paper further analyzes implications for convergence, stability, oscillation damping, and hyperparameter tuning.

**Strengths:**

This submission has novel and elegant theoretical contribution. The main equivalence result is new and insightful: it unifies two popular but seemingly different optimization paradigms. The insight that “dual optimism ≈ augmented regularization” is elegant and of clear theoretical and practical significance. This bridge justifies the strong empirical performance of PI-control methods used in deep RL and constrained deep learning.

This submission is mathematically sound and grounded in classical references. The local stability analysis via Jacobians is carefully argued and well contextualized. Theorems systematically extend the equivalence to convergence, oscillation damping, and conditioning, providing a unified picture.

The practical implications for tuning the optimism coefficient ω using ALM’s penalty intuition are particularly valuable. The discussion connects to recent empirical work in reinforcement learning and fairness-constrained training. This paper has some practical use in deep learning and LLM.

**Weaknesses:**

The most significant weakness of this submission is that this paper lacks empirical verification from numerical experiments. Although the theory neatly explains prior empirical results, no new experiments are provided. The reviewer suggest adding a simple numerical example (e.g., a 2D quadratic constraint problem or other simple constrained optimization problem ) showing the equivalence in iterates and the damping of oscillations. With the numerical experiments results, this can significantly improve the quality of this submission.

Other minor weaknesses include limited scope for inequality constraints. The equivalence for inequalities is only local, and the intuition behind the “two projections vs. one” distinction could be expanded with geometric visualization or numerical illustration.

Also the assumptions (strict complementarity, LICQ, SOSC) are standard but a little strong. The paper could benefit from discussing whether relaxed conditions (e.g., weak constraint qualification) still preserve equivalence.

This submission didn't have discussion on Multi-step or Second-order ALM Variants The authors explicitly limit equivalence to single-step first-order methods. It is suggest that adding brief discussion about how multi-step variants (as used in practical ALM solvers) might diverge, or outline what breaks in the proof.

**Questions:**

There are no other questions.

---

> ### Author Response · Authors · 2025-11-20
>
> We thank the reviewer for their feedback, and for highlighting the significance, novelty, and elegance of our contribution.
>
> ### Weaknesses
>
> > The most significant weakness of this submission is that this paper lacks empirical verification from numerical experiments. Although the theory neatly explains prior empirical results, no new experiments are provided. The reviewer suggest adding a simple numerical example (e.g., a 2D quadratic constraint problem or other simple constrained optimization problem ) showing the equivalence in iterates and the damping of oscillations. With the numerical experiments results, this can significantly improve the quality of this submission.
>
> We agree with the reviewer that empirical validation would strengthen our paper. We have thus conducted experiments to verify the iterate equivalence of Theorem 1. See the **Response to all Reviewers** for details.
>
> > Other minor weaknesses include limited scope for inequality constraints. The equivalence for inequalities is only local, and the intuition behind the “two projections vs. one” distinction could be expanded with geometric visualization or numerical illustration.
>
> Please find a clarification on the differences between both methods under inequality constraints in the **Response to all Reviewers**.
>
>
> > Also the assumptions (strict complementarity, LICQ, SOSC) are standard but a little strong. The paper could benefit from discussing whether relaxed conditions (e.g., weak constraint qualification) still preserve equivalence.
>
> We emphasize that our main equivalence results do **not** require these strong assumptions.
> * **For equality constraints:** The equivalence is purely algebraic and holds without any regularity assumptions. It operates directly on the iterates.
> * **For inequality constraints:** The only assumption we use to establish the equivalence of stationary points is **strict complementarity**. This is needed only to ensure the Jacobians of both ALM and dual optimistic ascent are well-defined at the KKT point.
>
> The LICQ and SOSC assumptions, which the reviewer correctly notes are strong, are introduced only in Section 4. They are not part of our equivalence results; we use them exclusively to transfer the existing, standard convergence guarantees from ALM, which do rely on them.
>
> > This submission didn't have discussion on Multi-step or Second-order ALM Variants The authors explicitly limit equivalence to single-step first-order methods. It is suggest that adding brief discussion about how multi-step variants (as used in practical ALM solvers) might diverge, or outline what breaks in the proof.
>
> The equivalence indeed does **not** extend to multi-step or second-order variants of the algorithms. **This implies that dual optimistic ascent is principled when used in the single-step, first-order setting, but not beyond it.** This is a crucial takeaway of our work, and we are happy to clarify it in the camera-ready version.
>
> For completeness, we sketch the main ideas here. We focus on the equality-constrained setting for clarity, though the same issues arise for inequalities: our results are fundamentally *first-order and single-step*.
>
> ---
>
> Our paper shows that the single-step, first-order dual optimistic ascent algorithm effectively performs a descent step on an Augmented Lagrangian, which Theorem 1 establishes via a change of variables. This equivalence is the reason the method behaves so well in practice.
>
> However, this equivalence breaks in the settings the reviewer mentioned:
> - **Multi-step primal updates.** Our change of variables guarantees equivalence for one primal update, but it does not hold for a second (or subsequent) inner step. After the first update, the "effective multiplier" encoding the augmented-Lagrangian structure becomes stale. Any additional inner steps would therefore minimize the standard Lagrangian, not the augmented one. Since the standard Lagrangian may not be locally convex, this multi-step minimization procedure may even fail to converge.
> - **Second-order primal updates.** The equivalence in Theorem 1 relies on matching the gradients of the two objectives via our change of variables. This equivalence, however, does not extend to the Hessians. The Augmented Lagrangian's Hessian contains an additional curvature term (proportional to $\nabla h \nabla h^\top$) that is absent from the standard Lagrangian—this is precisely the term that ensures local convexity. The multiplier transformation does not account for this Hessian term, so second-order primal steps differ and so do primal iterates.
>
> In short, the equivalence and the transferred guarantees apply specifically to single-step, first-order dual optimistic ascent. **This highlights a key insight of our work: the commonly used dual optimistic ascent (PI control) algorithm succeeds because it is *secretly* performing an Augmented Lagrangian step. Outside this regime, we therefore recommend using the explicit Augmented Lagrangian method.**

---

### Official Review · Reviewer_Wayq · 2025-11-03

**Soundness:** 4
**Presentation:** 4
**Contribution:** 3
**Rating:** 8
**Confidence:** 3

**Summary:**

This paper considers constrained optimization problems with both equality and inequality constraints. After describing three primal-dual approaches for solving the problem (Lagrangian, augmented Lagrangian, and dual optimistic ascent), the paper then establishes theoretical equivalence results between the augmented Lagrangian method and the dual optimistic ascent approach, hence allowing the desirable properties of augmented Lagrangian to be transferred to dual optimistic ascent (also known as PI control). Discussions are also provided on the practical implications of this equivalence, including the choice of the optimism hyperparameter in dual optimistic ascent.

**Strengths:**

- The paper is very well written and provides adequate background for an audience who may not be readily working in constrained optimization/learning.
- The provided equivalence results are interesting and could lead to significant follow-up theoretical and empirical research in constrained optimization, especially in the context of modern deep learning problems.

**Weaknesses:**

- There was no mention of how the results connect to the *parameterized* setting, where the primal variables are the parameters of a non-convex deep neural network architecture. Do you anticipate the results to have parameterized analogs under assumptions (such as near-universality of the parameterization)?
- The results are limited to a single primal step per iteration (although this is, I believe, what is mainly used in practice).
- The exact equivalence of primal iterates is limited to problems with only equality constraints.
- No empirical results are provided to corroborate/complement the theoretical findings.

**Questions:**

- What is the downside of augmented Lagrangian (and dual optimistic ascent) compared to regular Lagrangian? If convergence properties are more desirable, why is the regular Lagrangian the most commonly used approach in practice?
- Could you please clarify the statement on lines 224-225 on the replacement of the first dual update in (Lag-GD-OA) in practice? Is there a way to get around evaluations of $g$ and $h$ at $x_{t-1}$?
- Is there a way to better formalize the Jacobian "becoming real" in Corollary 5 (e.g., via characterizing the rate of decline of the imaginary eigenvalues)?
- I understand that this paper's contribution is primarily on the theoretical side. However, I believe some simple experiments on the differences between augmented Lagrangian and dual optimistic ascent in problems with inequality constraints, as well as dynamic tuning of the optimism hyperparameter by mirroring augmented Lagrangian penalty scheduling methods (as mentioned in lines 460-461), would be interesting and helpful additions to the paper.
- I would suggest swapping the discussion on OGDA and ALA in Section 2 (i.e., discussing OGDA first and then ALA) for a better flow.

---

> ### Author Response · Authors · 2025-11-20
> **Response to Reviewer Wayq**
>
> We thank the reviewer for their feedback and for highlighting the significance and potential impact of our work.
>
> ### Weaknesses
>
> > There was no mention of how the results connect to the parameterized setting, where the primal variables are the parameters of a non-convex deep neural network architecture. Do you anticipate the results to have parameterized analogs under assumptions (such as near-universality of the parameterization)?
>
> Yes, our results apply directly to the non-convex, parameterized setting.
>
> Our main results establish an equivalence between the two optimization algorithms that does not rely on convexity or on any particular parameterization; in particular, the identity holds when the primal variables are the parameters of a deep neural network.
>
> In addition, the theoretical guarantees we transfer from the ALM include convergence to local solutions (locally stable stationary points), which are notions of optimality relevant in non-convex deep learning.
>
> We hope this clarifies the connection the reviewer had in mind, and we would be happy to provide further detail if we have misunderstood their concern.
>
> > The results are limited to a single primal step per iteration (although this is, I believe, what is mainly used in practice).
>
> Indeed, as the reviewer notes, the single–primal-step setting is the standard approach in constrained deep learning. Our stated equivalence holds only under this assumption; beyond the single–step regime, the two methods are no longer equivalent.
>
> We also refer the reviewer to our response to the final weakness raised by **Reviewer GKqh**, where we explain in detail why this assumption is necessary for the equivalence and why the ALM enjoys stronger guarantees in settings that go beyond single–primal-step updates.
>
>
> > The exact equivalence of primal iterates is limited to problems with only equality constraints.
>
> This is correct. As we discuss in the paper, in the presence of inequality constraints the equivalence no longer holds at the level of primal iterates. Instead, the equivalence applies to the *locally stable stationary points*—i.e., the local solutions of the constrained problem—which we believe is the more relevant practical takeaway, as it highlights the effective interchangeability of both methods for general constrained optimization problems.
>
>
> > No empirical results are provided to corroborate/complement the theoretical findings.
>
> As detailed in the **Response to all Reviewers**, we have addressed this during the discussion period.

---

> > ### Author Response · Authors · 2025-11-20
> >
> > ### Questions
> >
> > > What is the downside of augmented Lagrangian (and dual optimistic ascent) compared to regular Lagrangian? If convergence properties are more desirable, why is the regular Lagrangian the most commonly used approach in practice?
> >
> >
> > In our experience, this largely comes down to two factors:
> > - The standard Lagrangian approach is **simpler**, avoiding multiplier extrapolations (as in ALM) or storing and using previous gradients (as in dual PI/optimistic methods). These mechanisms, while powerful, are more convoluted than the plain rule “increase the multiplier when infeasible, decrease it when (strictly) feasible.” Moreover, the plain Lagrangian **does not introduce an additional hyperparameter**, which further simplifies its practical use.
> > - In many settings, including several referenced in our submission, the standard Lagrangian already works well in practice. Even papers that employ ALM or dual PI controllers often do so to obtain *more stable or controlled convergence* (e.g., fewer oscillations near the feasibility boundary, reduced overshoot into infeasible regions), not because ALM is *required* to reach a good local solution. Somewhat surprisingly, the plain Lagrangian often succeeds in non-convex deep learning despite lacking theoretical guarantees—an underexplored but interesting phenomenon.
> >
> > As a concrete example: in our own workflows, we typically begin with the standard Lagrangian method and introduce a nonzero augmented-Lagrangian penalty $c$ only if the baseline method shows instability or fails to converge satisfactorily.
> >
> >
> > > Could you please clarify the statement on lines 224-225 on the replacement of the first dual update in (Lag-GD-OA) in practice? Is there a way to get around evaluations of $g$ and $h$ at $x_{-1}$?
> >
> > We are happy to clarify this, and we will include a brief note on it in the camera-ready version.
> >
> > The first dual update in dual optimistic ascent is
> > $$ \mu_1 = \mu_0 + \eta  h(x_0) + \omega \big[h(x_0) + h(x_{-1})\big], $$
> > which requires evaluating $h(x_{-1})$. In practice, there are two standard ways to handle this:
> >
> > 1. Set $h(x_{-1}) = 0$, which yields $\mu_1 = \mu_0 + (\eta + \omega) h(x_0)$, i.e., a gradient-ascent update with effective step size $(\eta + \omega)$.
> > 2. Set $h(x_{-1}) = h(x_0)$, which gives $\mu_1 = \mu_0 + \eta h(x_0)$, yielding a standard GA update with step-size $\eta$. This corresponds to initializing the “pre-step” violation to match the violation at initialization. In our experience, this choice is often preferable in practice, as it avoids a disproportionately aggressive first dual update.
> >
> > In either case, both options amount to gradient ascent, and their difference can be absorbed into the choice of initialization $\mu_0$. Thus, this detail does **not** affect any of our theoretical claims.
> >
> > This is directly analogous to how momentum buffers $v_0$ are typically initialized in momentum methods—either to zero or to the first gradient—both conventions are valid.
> >
> >
> > > I understand that this paper's contribution is primarily on the theoretical side. However, I believe some simple experiments on the differences between augmented Lagrangian and dual optimistic ascent in problems with inequality constraints, as well as dynamic tuning of the optimism hyperparameter by mirroring augmented Lagrangian penalty scheduling methods (as mentioned in lines 460-461), would be interesting and helpful additions to the paper.
> >
> > We have added experiments highlighting the dynamic tuning of the optimism hyper-parameter following ALM schemes. See the **Response to all Reviewers** for details. We have also added a discussion on the differences between both methods for inequality-constrained problems (see also the **Response to all Reviewers**).
> >
> > > I would suggest swapping the discussion on OGDA and ALA in Section 2 (i.e., discussing OGDA first and then ALA) for a better flow.
> >
> > That's a very good suggestion from the reviewer. We have made this change.

---

> ### Author Response · Authors · 2025-11-20
>
> > Is there a way to better formalize the Jacobian "becoming real" in Corollary 5 (e.g., via characterizing the rate of decline of the imaginary eigenvalues)?
>
> We thank the reviewer for their suggestion. We can definitely formalize this in the camera-ready version of the paper. For the sake of the discussion period, here are the broad strokes of what can be said.
>
> The Jacobian of dual optimistic ascent, $J_{\text{OGA}}$, has the same set of non-trivial eigenvalues as the ALM Jacobian, $J_{\text{ALM}}$, meaning we can analyze the spectrum of $J_{\text{ALM}}$. For the active subsystem (which excludes the trivial and real eigenvalues), $J_{\text{ALM}} = I - \eta M$, where $M$ is:
> $$
> M = \begin{pmatrix} A + (\omega+\eta)B^\top B & B^\top \\\\ -B & 0 \end{pmatrix}.
> $$
>
> The eigenvalues of $J_{\text{ALM}}$ are real if and only if the eigenvalues $\sigma$ of $M$ are real.
>
> As shown by Benzi and Simoncini (2006), the eigenvalues $\sigma$ of $M$ are the roots of the quadratic equation
> $$
> \sigma^{2}(u^{\ast}u) - \sigma \, u^{\top}(A+(\omega+\eta) B^{\top}B)u + u^{\ast}(B^{\top}B)u = 0.
> $$
>
> These eigenvalues $\sigma$ are complex only if the discriminant
> $$
> \Delta(\omega) = \big[u^{\ast}(A+(\omega+\eta) B^{\top}B)u\big]^{2} - 4(u^{\ast}u)(u^{\ast}B^{\top}B u)
> $$
> is negative.
>
> This discriminant is a convex quadratic in $\omega$ with positive leading coefficient whenever $u^{\ast} B^{\top} B u \neq 0$; otherwise, the roots are guaranteed to be real. In particular, for each such eigendirection there exists a **finite** threshold $\omega^{\ast}$ beyond which $\Delta(\omega) \ge 0$, so the corresponding eigenvalues of $M$ (and hence of $J_{\text{ALM}}$) are real. For $\omega < \omega^\ast$ with $\Delta(\omega) < 0$, the imaginary part of $\sigma$ is proportional to $\sqrt{-\Delta(\omega)}$. Under a generic non-degeneracy assumption (simple root of $\Delta$ at $\omega^\ast$), the discriminant crosses zero linearly, $\Delta(\omega) \approx K(\omega^{\ast} - \omega)$, so near $\omega^{\ast}$ the decay rate of the imaginary part is
> $$
> |\operatorname{Im}\sigma(\omega)| = \mathcal{O}\big(\sqrt{\omega^{*} - \omega}\big).
> $$
> We will formalize these conditions and bounds in the camera-ready version.

---

> > ### Comment · Reviewer_Wayq · 2025-11-25
> >
> > Thank you for your response and the added numerical experiments. I maintain my favorable rating of the paper and believe it makes a valuable contribution to the constrained optimization and learning literature.

---

### Author Response · Authors · 2025-11-20
**Response to all Reviewers**

We thank all three reviewers for their valuable feedback. We are encouraged that the reviewers described our contribution as “novel and elegant”, “interesting”, and as providing a “sound theoretical foundation” for a practical method.

We address here two concerns shared by **Reviewers Wayq** and **GKqh**.

### Lack of Empirical Results

We thank the reviewers for this crucial feedback. **Reviewer GKqh** suggested including an empirical verification to corroborate our theory, while **Reviewer Wayq** suggested substantiating our claim that the equivalence provides practical, algorithmic benefits (e.g., reusing ALM tuning schemes for dual optimistic ascent).

We have addressed both points. For the sake of the discussion period, we have added a supplementary PDF with new experiments. **We invite the reviewers to view this file, which is attached under “Supplementary Material” above.**

These new experiments demonstrate:
1. The exact primal-iterate equivalence predicted by Theorem 1.
2. The practical utility of our connection, showing that an ALM-style adaptive scheduler can be directly applied to the $\omega$ parameter in dual optimistic ascent to improve convergence.

We believe these additions make the paper significantly stronger. Should the paper be accepted, we will polish these experiments and integrate them into the camera-ready version.


### Differences between ALM and dual optimistic ascent for inequality constraints

Here, we characterize the differences between the two methods for inequality constraints. Crucially, **the effective multipliers of both methods coincide for a significant portion of the optimization trajectory.**

Recall the updates for both methods:
$$
\lambda_{t+1}^{\text{OGA}} = [\lambda_{t}^{\text{OGA}} + \eta g(x_{t}) + c (g(x_{t}) - g(x_{t-1})) ]\_+
$$
and
$$
\lambda_{t+1}^{\text{ALM}} = \left(1 - \frac{\eta}{c} \right)\lambda_{t}^{\text{ALM}} + \frac{\eta}{c} [\lambda_{t}^{\text{ALM}} + c g(x_{t+1})]_+
$$

**1. The Common Regime (Active Constraints)**.
Consider the common case where the constraint is violated ($g(x_t) > 0$) for a sustained period and multipliers become large. Here, the projection arguments remain strictly positive, so the operator $[\cdot]\_+$ acts as the identity.
Consequently, the updates simplify to:
$$
\lambda_{t+1}^{\text{OGA}} = \lambda_{t}^{\text{OGA}} + \eta g(x_{t}) + c (g(x_{t}) - g(x_{t-1}))
$$
and
$$
\lambda_{t+1}^{\text{ALM}} = \lambda_{t}^{\text{ALM}} + \eta g(x_{t+1})
$$
These recover the exact dynamics of the equality-constrained setting, where—under the initialization of Theorem 1—we know the primal iterates match exactly. This scenario is fairly common in practice, particularly during phases where the constraint is consistently violated.

**2. When Discrepancies Occur**.
The projection logic is triggered differently in the two methods whenever the **difference term** $c(g(x_{t}) - g(x_{t-1}))$ in OGA contradicts the current feasibility measurement, leading to two scenarios:
* **Premature Drop:** $g(x_{t-1}) \gg g(x_t) > 0$ (violation), but decreasing so fast that the negative trend dominates. OGA projects to zero ("anticipating" feasibility), while the ALM multiplier stays active.
* **Pre-activation:** $g(x_{t-1}) \ll g(x_t) \le 0$ (feasible), but increasing so fast that the positive trend dominates. OGA activates ("anticipating" violation), while ALM stays inactive.

Mathematically, for this disagreement to occur, the single-step change must dominate the current constraint magnitude:
$$
|g(x_t) - g(x_{t-1})| \gtrsim |g(x_t)|
$$
This condition is rare in smooth optimization with small step sizes, occurring primarily during rapid changes across the boundary.

We highlight that, near the boundary ($g(x_t) \approx 0$), the differences between both methods become **negligible**. By continuity, the multipliers here are small; thus, even if the methods disagree (one clamps to zero, the other retains a small value), the resulting primal gradient contribution $\lambda \nabla g(x)$ is still very small. For trajectories far from the boundary (such as the active case above, or a significantly inactive constraint), the projection logic is typically stable and identical.


In summary, OGA is slightly more **anticipatory** (reacting to trends), while ALM is more **robust** (reacting to states). In any case, we re-iterate the most important practical implication of our work: regardless of these transient differences, **both methods share the same set of locally stable stationary points.** Locally, both algorithms target the same solutions in a principled way.

---

### Author Response · Authors · 2025-11-30
**Closing Remarks for the AC**

We thank the AC for handling our submission.

All reviewers recognize the value of the paper:
* **Wayq** (8, accept) champions the paper; we have addressed all of their comments.
* **2hch** (6, borderline accept) finds the derivations sound and notes that we “establish a theoretical foundation” for dual optimistic ascent in constrained deep learning.
* **GKqh** (4, borderline reject) describes our main result as a “novel and elegant theoretical contribution” with “clear theoretical and practical significance.”

During the discussion period, we added numerical experiments to the supplementary PDF—where none existed before—to (i) verify the iterate-level equivalence proved in the paper and (ii) demonstrate the practical benefits of importing ALM tuning schemes into dual optimistic ascent. This directly addresses the concerns raised by **Wayq** and **GKqh**.

**Crucially, GKqh’s most significant reason for not recommending acceptance was the lack of empirical validation.** Given their initial praise for the work’s “novel and elegant” nature and “clear significance,” and the fact that we have now provided the requested experiments, we respectfully believe their concerns are fully resolved.

Scientifically, our paper offers strong value: it unifies two seemingly distinct methods (dual optimistic ascent / PI control and the Augmented Lagrangian method) and transfers ALM’s robust guarantees to a method already widely used in constrained deep learning and RL, finally placing dual optimistic ascent on firm theoretical ground.

---

### Meta-Review · Area_Chair_dSKx · 2026-01-11

**Summary:**

This paper meets three main concerns:
1. The theoretical results are limited to a single primal step per iteration;
2. No empirical results are provided to corroborate/complement the theoretical findings;
3. The exact equivalence of primal iterates is limited to problems with only equality constraints.

**Reviewer Concerns:**

1. The authors gives an explaination of the theoretical limits, and provides  detail why this assumption is necessary for the equivalence;
2. the authors have added a supplementary PDF with new experiments, which mostly address the reviewers' concerns;

3. the authors characterize the differences between the two methods for inequality constraints.

**Reviewer Scores:**

Reviewer GKqh could raise the score to 5 or 6.

---

### Decision · Program_Chairs · 2026-01-26

Accept (Poster)